# On Efficient Scaling of GNNs via IO-Aware Layers Implementations

**Daria Fomina** [* 1 2] **Daniil Krasylnikov** [* 1 2 †] **Alexey Boykov** [2 1] **Andrey Dolgovyazov** [3] **Vyacheslav Zhdanovskiy** [1]
**Fedor Velikonivtsev** [1 2]

## Abstract

Graph Neural Networks (GNNs) are bottlenecked by sparse, irregular memory access. Popular frameworks such as DGL and PyTorch Geometric support general message passing, but complex layers often materialize edge-wise intermediates, increasing memory traffic and limiting scalability on large graphs. We take an I/O- and arithmetic-intensity–centric view and show that widely used layers fall into three kernel families: SpMM-based convolutions, reduction-based aggregations, and attention-based layers (GATv2/Graph Transformer). For each family, we develop GPU kernels that reduce data movement, improve locality, and remain robust across realistic graphs. We also study graph reordering and find that its impact depends on the kernel mapping: it benefits neighbor-parallel (gather-dominated) kernels more consistently than feature-parallel designs. Empirically, our fused attention kernels reach up to $3.9\times$ speedup for Graph Transformer (median $1.6\times$), with Tensor Core (block-sparse) variants up to $7.3\times$ on locally dense graphs; for GATv2 we reach up to $8.5\times$ speedup (median $2.0\times$) while reducing peak memory by up to $76\times$ (median $6\times$). Our degree-aware reduction kernels achieve up to $10\times$ speedup (median $2.6\times$). For SpMM-based layers, properly cached cuSPARSE achieves up to $8\times$ speedup over DGL and outperforms evaluated custom baselines in the majority of evaluations. We release our implementations as drop-in replacements in our [GitHub repository](#) to support reproducible, hardware-aware GNN acceleration.

## 1. Introduction

Graph Neural Networks (GNNs) are a core tool for learning over relational and unstructured data, with applications in recommendation and fraud detection, spatiotemporal forecasting, and scientific simulation and more (Lam et al., 2023; Shi et al., 2023; Pfaff et al., 2021; Fey et al., 2023; Sanchez-Gonzalez et al., 2020). Despite rapid progress in model design, training and inference performance of GNNs on modern GPUs often lags behind dense deep learning workloads. A key reason is that dominant GNN operators are sparse and irregular: neighborhood aggregation and attention traverse graph edges with data-dependent, permutation-invariant access patterns, leading to low arithmetic intensity and limited locality (Zhang et al., 2021; Peng et al., 2024).

This gap is amplified by hardware trends. Modern GPUs exhibit a widening separation between peak compute throughput and HBM bandwidth, pushing more kernels into the memory-bound regime unless they perform substantial computational work per byte moved. In roofline terms, the ridge point (arithmetic intensity required to become compute-bound) shifts to higher values[1]. As a result, reducing FLOPs alone often does not translate into wall-clock speedups (Zhang et al., 2025). This mismatch is well documented in transformer attention, where IO-aware algorithms that explicitly reduce HBM↔SRAM traffic deliver large practical gains (Dao et al., 2022; Dao, 2023; Shah et al., 2024; Ivanov et al., 2021). We argue that a similar missing principle hinders scalable GNN acceleration: many implementations prioritize operator generality, while under-optimizing data movement and intermediate materialization, making performance increasingly dominated by memory traffic (Zhang et al., 2021).

Existing GNN software stacks typically follow one of two approaches. Frameworks such as Deep Graph Library (DGL) and PyTorch Geometric provide a general message-passing interface and optimized sparse primitives (Wang et al., 2019; Fey et al., 2025), but typical layers may still materialize large edge-wise intermediates, increasing both

---

[*]Equal contribution .[†]Work done during employment at Yandex. [1]Yandex [2]HSE University [3]ITMO University. Correspondence to: Fedor Velikonivtsev <fedorvelikonivtsev@gmail.com>.

*Proceedings of the $43^{rd}$ International Conference on Machine Learning*, Seoul, South Korea. PMLR 306, 2026. Copyright 2026 by the author(s).

---

[1]For reference, A100 SXM-4 has $\approx$312 TFLOPs (FP16/BF16 Tensor Cores) and $\approx$2 TB/s HBM bandwidth ($\approx$156 FLOPs/byte); H100 SXM-5 has $\approx$1980 TFLOPs and $\approx$3.35 TB/s ($\approx$592 FLOPs/byte).

memory traffic and peak activation footprint. At the other extreme, specialized operator implementations can require substantial per-layer engineering and may not consistently benefit from new hardware features across diverse graphs and feature sizes (Liu et al., 2024; Xiang et al., 2025).

Recent work addresses these issues via adaptive runtimes and scheduling (Wang et al., 2021), more efficient implementations for specific layers (Peng et al., 2024; Zhang et al., 2021), and structured sparsity formats that map sparse workloads onto Tensor Cores (Wang et al., 2023; Xiang et al., 2025; Li & Chandramowlishwaran, 2025). However, these approaches often target a subset of operators, require nontrivial preprocessing/custom formats, and can be sensitive to graph structure and degree distribution.

In this work, we take an I/O- and arithmetic-intensity–centric view of GNN computation and show that widely used layers cluster into three kernel families with distinct hardware behavior: (i) sparse–dense matrix multiplications-based (SpMM-based) convolutions (e.g., GCN/GraphConv), (ii) reduction-based aggregations (e.g., $\min/\max$ and related segment reductions), and (iii) complex workflows like attention-based layers (e.g., GATv2 and Graph Transformers). For each family, we analyze memory behavior of common implementations and validate our designs by profiling across realistic graphs with varying density and degree distributions.

This yields three actionable outcomes. First, *our experiments show* that for SpMM-based convolutions vendor primitives remain highly competitive: `cuSPARSE` often matches or outperforms framework baselines, and caching graph preprocessing and a transposed adjacency improves end-to-end latency for the backward pass. Second, *we empirically find* that for reduction-based layers degree-aware tiling is crucial under heavy-tailed degree distributions, improving throughput on high-degree nodes without increasing memory footprint, while a simple coalesced layout suffices for low-degree nodes. Third, for attention-based layers, we contribute FlashAttention-inspired IO-aware kernels that fuse neighbor-wise computation, avoid materializing edge-wise intermediates, and optionally leverage Tensor Cores via block-sparse layouts.

We also revisit graph reordering and show that its effectiveness depends on the kernel parallelization strategy and degree distribution: neighbor-parallel designs benefit more on high-degree graphs, whereas on low-degree graphs (i.e. road networks) the impact is limited and performance is dominated by per-node overheads. Finally, we release an open-source implementation with minimal dependencies (CUDA, PyTorch, `cuSPARSE`) and a PyTorch interface as drop-in replacements for widely used GNN layers.

## 2. Background

### 2.1. Notation and message passing

We consider a homogeneous graph $G = (V, E)$ with $|V| = N$ nodes and $|E| = M$ directed edges. Node features at layer $\ell$ are $H^{(\ell)} \in \mathbb{R}^{N \times D_\ell}$, with $H^{(0)} = X$. We primarily use Compressed Sparse Row (CSR) format for graph storage. In the remainder of the paper, we often omit the layer index $\ell$ when it is clear from the context, since we analyze the performance of a single representative layer and the same kernel-level arguments apply across layers.

Most GNN layers can be expressed through the Message Passing Neural Network (MPNN) template:

$$m_i^{(\ell)} = \text{AGG}^{(\ell)}(\{ \text{MSG}^{(\ell)}(h_i^{(\ell)}, h_j^{(\ell)}, e_{ij}) : j \in \mathcal{N}(i) \}),$$
$$h_i^{(\ell+1)} = \text{UPDATE}^{(\ell)}(h_i^{(\ell)}, m_i^{(\ell)}),$$

where AGG can be any permutation-invariant function, such as min/max/average/etc, and MSG and UPDATE can be arbitrary learnable functions (Gilmer et al., 2017; Kimura et al., 2024). From a hardware perspective, the dominant cost typically lies in the neighborhood aggregation over edges, which induces sparse and irregular memory access patterns (Peng et al., 2024; Zhang et al., 2021).

### 2.2. Three operator families of GNN layers

We focus on three operator families that cover widely used GNN layers and expose distinct hardware bottlenecks.

**(i) SpMM-based convolutions.** These layers can be expressed as SpMM between a sparse (typically normalized) adjacency matrix $\tilde{A}$ and a dense matrix $C$, where $C = H^{(\ell)} W^{(\ell)}$ [2]:

$$H^{(\ell+1)} = \sigma(\tilde{A} C) = \sigma\left( \tilde{A} H^{(\ell)} W^{(\ell)} \right).$$

For instance, in case of Graph Convolutional Network from Kipf & Welling (2017), $\tilde{a}_{ij} = 1/\sqrt{\deg(i) \deg(j)}$.

**(ii) Reduction-based aggregations.** Some layers (pooling, "segment" reductions) compute per-edge values followed by a neighborhood-wise reduction [3]:

$$h_i^{(\ell+1)} = \text{UPDATE}^{(\ell)} \left( h_i^{(\ell)}, \text{REDUCE}_{j \in \mathcal{N}(i)} h_j^{(\ell)} \right),$$

where REDUCE is typically max, min (Hamilton et al., 2017). Unlike SpMM-based operators, these operators are often implemented using scatter/gather primitives and custom reduction kernels. Their performance is sensitive to (1) degree distribution (due to load imbalance), (2) reduction

---

[2]Note that $C = H^{(\ell)} W^{(\ell)}$ is a conventional GEMM operation of two dense matrices and thus its computation is not subject to optimization in this paper.

[3]Note that in this group we consider operators which *cannot* be expressed as SpMM unlike edge-wise mean or sum for instance.

strategy (linear vs parallel), and (3) whether intermediate edge-wise tensors are materialized, if applicable.

**(iii) Attention-based layers.** The operators in this family utilize an attention mechanism to weigh messages from the neighbors during the update phase (for simplicity of notations, we omit head dimension from the indexing):

$$h_i^{(\ell+1)} = \sum_{j \in \mathcal{N}(i) \cup \{i\}} \alpha_{ij} \, v_j,$$

$$\alpha_{ij} = \text{softmax}_{j \in \mathcal{N}(i) \cup \{i\}}(e_{ij}), \;\; e_{ij} = \text{MSG}^{(\ell)}(i, j).$$

The family members mainly differ in how they parameterize the logits $e_{ij}$ and the value vectors $v_j$ that are aggregated after the softmax. Typical examples include GAT, GATv2, and graph transformer layers (Veličković et al., 2018; Brody et al., 2022; Shi et al., 2021). For GATv2, the logits are computed as $e_{ij} = \mathbf{a}^\top \text{LeakyReLU}(W_\ell h_i + W_r h_j)$ and the values as $v_j = W_r h_j$, where $W_\ell$, $W_r$, and $\mathbf{a}$ are learnable parameters. Graph Transformer layers (Min et al., 2022; Shi et al., 2021) use dot-product attention (Vaswani et al., 2017):

$$q_i = W_Q h_i, \; k_j = W_K h_j, \; v_j = W_V h_j, \; e_{ij} = \frac{q_i^\top k_j}{\sqrt{D}}.$$

## 2.3. GNN backends and prior work on GNN acceleration

Two of the most widely used Graph-ML frameworks for PyTorch are Deep Graph Library (DGL) and PyTorch Geometric (PyG). DGL exposes a graph-centric API and implements GNN layers via two generalized sparse primitives: generalized SpMM (gSpMM) for node-wise aggregation and generalized sampled dense–dense matrix multiplication (gSDDMM) for edge-wise computations. In its CUDA backend, DGL typically parallelizes gSpMM over destination nodes and gSDDMM over edges (Wang et al., 2019). PyG follows a PyTorch-native design and usually implements message passing with scatter/gather reductions and lightweight CUDA extensions, offering a flexible interface for general message passing layers (Matthias et al., 2025).

While DGL provides efficient, operator-level kernels for these primitives (e.g., via `dgl.ops`), more complex layers are often expressed as a composition of gSDDMM and gSpMM, which results in materializing intermediate edge messages and subsequently reducing them during AGG and complex MSG in both frameworks, increasing memory traffic and peak activation footprint. In practice, DGL mitigates part of this overhead with optimized primitive-level kernels, but materialization remains common for composite layers. Moreover, fusing such pipelines via automatic compilation is non-trivial, since the computation alternates between edge-parallel and node-parallel phases, making it difficult to preserve both locality and load balance across the entire layer (Zhang et al., 2021).

Prior work on accelerating GNNs on GPUs spans vendor libraries, sparse-kernel primitives, fusion-oriented systems, and graph/format transformations. On the vendor side, NVIDIA cuSPARSE provides highly optimized sparse linear algebra (notably SpMM) and serves as a strong baseline for SpMM-based GNN layers. However, reduction and attention operators are not covered by generic SpMM and their performance depends strongly on parallelization choices, intermediate materialization, and memory locality. cuGraph similarly exposes GPU graph primitives and GNN-oriented components within the RAPIDS ecosystem.

Beyond vendor libraries, GE-SpMM proposes CSR-native SpMM-like kernels tailored to GNN usage and introduces coalesced row caching and warp-level techniques to improve memory behavior (Huang et al., 2020). Adaptive sparse tiling improves locality for SpMM/SDDMM within CSR via intra-row reordering to enable tiling and data reuse (Hong et al., 2019). Classical analyses of sparse GPU kernels (e.g., CSR SpMV on cache-based GPUs) emphasize the role of locality, parameterized mappings, and (auto-)tuning, which remain relevant when interpreting cache effects and graph reordering in modern GNN workloads (Reguly & Giles, 2012).

A complementary line targets end-to-end execution graphs and fusion. FuseGNN, Fused3S, and related pipelines reduce kernel-launch overheads and intermediate tensors by fusing common operator sequences (e.g., SDDMM–softmax–SpMM for sparse attention) (Li & Chandramowlishwaran, 2025; Chen et al., 2020). DF-GNN adapts fusion and thread scheduling to operation structure and degree skew, including super-nodes (Liu et al., 2024). GNNAdvisor studies computational graphs through a coordinated computation/IO/memory lens and proposes operator reorganization, unified thread mapping for fusion across vertex/edge phases, and recomputation to reduce activation footprint (Wang et al., 2021). Cached Operator Reordering explores alternative execution orders with caching, though public implementations are not always available (Bazinska et al., 2023). MaxK-GNN represents algorithm–system co-design that modifies representations to reduce memory traffic and enable tailored kernels (Peng et al., 2024).

Finally, several works leverage specialized GPU units via format/graph transformations. TC-GNN and cuTeSpMM explore dense Tensor Cores for SpMM through structured blocking and introduce structure-dependent criteria for when Tensor-Core acceleration is beneficial (Wang et al., 2023; Xiang et al., 2025). RT-GNN targets Sparse Tensor Cores by reordering graphs to better match structured sparsity constraints; its practical impact depends on reordering cost, compatibility constraints, and how closely evaluated settings match real end-to-end pipelines (Yan et al., 2025).

## 2.4. Graph reordering for locality

Graph reordering applies a permutation $\pi$ to node indices to improve cache-locality by placing frequently co-accessed nodes closer in memory. Classic methods include METIS-style partitioning and fill-reducing orderings (Karypis & Kumar, 1997). In GNNs, reordering can reduce memory traffic and improve end-to-end runtime in general frameworks (Merkel et al., 2024). In this paper, we show that its benefit is not universal: the effect depends on the kernel's memory access pattern (feature-parallel vs. neighbor-parallel) and graph structure. In particular, for low-degree nodes, the per-node working set is small and the potential locality improvement is limited.

## 2.5. Hardware characteristics and performance considerations

We focus on GPUs, as they are the dominant platform for training and inference of modern GNNs. The qualitative performance considerations discussed below extend to other throughput-oriented accelerators.

**CUDA Execution model and GPU memory hierarchy.** GPU programs (kernels) are executed by massive number of threads grouped into warps and thread blocks scheduled on streaming multiprocessors (SMs).

Modern GPUs expose a deep memory hierarchy that trades capacity for bandwidth and latency. On-chip resources (registers, shared memory, and L1 cache) are shared within an SM and are orders of magnitude faster than global High-Bandwidth Memory (HBM), yet has much smaller capacity (NVIDIA). As peak compute throughput has grown faster than HBM bandwidth, many kernels are increasingly limited by memory traffic rather than arithmetic throughput, making effective reuse in on-chip memory critical. Achieving high performance typically requires (i) coalesced global memory accesses, (ii) sufficient parallelism to hide memory latency, and (iii) careful use of on-chip storage for partial reductions and reuse of frequently-accessed variables.

**Roofline and arithmetic intensity.** A convenient way to reason about GPU performance is the Roofline model (Williams et al., 2009), which relates attainable throughput to *arithmetic intensity* (FLOPs per byte moved from HBM). As GPU compute throughput grows faster than HBM bandwidth, the arithmetic intensity required to become compute-bound increases, pushing more sparse and irregular operators into the memory-bound regime. High-intensity kernels (e.g., dense GEMM) can be compute-bound, whereas low-intensity kernels are typically bandwidth-bound. Most sparse and reduction-heavy primitives used in GNNs exhibit low arithmetic intensity and are therefore often memory-bound, moreover, intermediate materialization further increases data movement.

**Kernel fusion and intermediate materialization.** A standard way to accelerate bandwidth-bound workloads is kernel fusion: composing multiple operations so that inputs are loaded once from HBM and reused in registers/SRAM (Wei et al., 2021; Chen et al., 2018).

**Challenges with GNNs.** GNN workloads combine sparse, irregular graph access with dense per-node/per-edge feature computation, which introduces several GPU-specific challenges.

First, neighborhood aggregation requires gathering features from non-contiguous node indices, and even when feature loads are coalesced along the feature dimension, the sequence of neighbor base addresses can be highly irregular, reducing cache effectiveness and increasing TLB/DRAM pressure. Moreover, in message-passing formulations, intermediate edge messages (often of size $O(M \cdot D_\ell)$) may be materialized before aggregation. This further increases memory traffic and peak activation footprint, and it becomes a key bottleneck for large graphs.

Second, GNN layers naturally require an edge-wise computations followed by a node-wise reductions. While modern Deep Learning compilers like `torch.compile` are very powerful, effective fusion is complicated in this alternating workflow (Zhang et al., 2021; Li et al., 2021; Dao et al., 2022; Paszke et al., 2019).

These characteristics motivate IO-aware kernel designs that (i) reduce intermediate materialization, (ii) increase on-chip reuse via tiling and fusion where possible, and (iii) explicitly account for degree skew and layout sensitivity in the mapping of work to GPU threads.

## 3. IO-aware kernels

In this section, we provide a brief overrview of the proposed techniques aimed at improved kernel memory usage and the latency. The full description and the list of available kernels' options with the analysis are provided in Appendix D in the corresponding sections.

### 3.1. Degree-aware reduction-based convolutions aggregation

Real graphs exhibit heavy-tailed degree distributions, causing load imbalance when each node is processed uniformly (Albert & Barabási, 2002; Newman, 2003). We partition nodes into `light` and `heavy` subsets by a node degree threshold (which is typically some upper quantile from the degree distribution). `Light` nodes use a feature-parallel kernel (one block per node) with coalesced memory access pattern. For `heavy` nodes, we introduce tiling for reduction over edge chunks: each thread block computes a local reduction over its chunk. Partial results are merged via atomic operations on packed 64-bit (value, index) pairs, enabling a

*Table 1.* Results for Graph Transformer speedups on selected graphs (heads $H = 4$, head dim $D = 128$). `N/A` means DF-GNN encountered illegal memory access on this setup and failed to launch; higher is better.

| Backend | ogbn-arxiv | artnet-exp | city-roads-M | pokec-regions | tolokers-2 | city-roads-L |
|---|---|---|---|---|---|---|
| **Latency speedup** ($t_{\mathrm{DGL}}/t$), fwd/bwd | | | | | | |
| Ours (CSR fused) | 1.27×/1.24× | 1.17×/1.15× | 1.15×/1.19× | 1.25×/1.15× | 2.00×/1.10× | 1.15×/1.16× |
| DF-GNN | 1.62×/1.19× | 1.19×/1.21× | 1.12×/1.20× | N/A | N/A | 1.13×/1.18× |
| Ours (WSB/TC) | 1.56×/0.77× | 1.29×/0.78× | 1.16×/1.07× | 1.50×/0.70× | 2.69×/0.71× | 1.15×/1.03× |
| **Peak-memory ratio** ($\mathrm{mem}_{\mathrm{DGL}}/\mathrm{mem}$), fwd/bwd | | | | | | |
| Ours (CSR fused) | 1.19×/1.01× | 1.22×/1.02× | 1.17×/1.00× | 1.27×/1.04× | 1.57×/1.13× | 1.16×/1.00× |
| DF-GNN | 1.18×/1.14× | 1.19×/1.08× | 1.16×/1.08× | N/A | N/A | 1.16×/1.13× |
| Ours (WSB/TC) | 1.08×/1.21× | 1.08×/1.21× | 1.08×/1.21× | 1.10×/1.21× | 1.23×/1.20× | 1.07×/1.21× |

*Table 2.* Latency speedup ($t_{\mathrm{DGL}}/t$) and Peak-memory ratio ($\mathrm{mem}_{\mathrm{DGL}}/\mathrm{mem}$) for GATv2 on selected graphs (heads $H = 2$, head dim $D = 64$); higher is better.

| **Latency speedup** | ogbn-arxiv | twitch-views | artnet-exp | pokec-regions | tolokers-2 | city-roads-L |
|---|---|---|---|---|---|---|
| Ours | 2.82×/0.98× | 5.32×/1.30× | 5.45×/2.04× | 6.68×/2.02× | 5.59×/1.89× | 9.70×/1.71× |
| PyG | 0.95×/0.95× | 0.57×/0.67× | 0.39×/0.51× | OOM/OOM | 0.45×/0.70× | 1.54×/0.62× |
| **Peak-memory ratio** | ogbn-arxiv | twitch-views | artnet-exp | pokec-regions | tolokers-2 | city-roads-L |
| Ours | 4.67×/3.46× | 44.17×/31.71× | 6.90×/5.34× | 12.74×/8.47× | 32.59×/28.55× | 1.77×/1.51× |
| PyG | 0.58×/0.69× | 0.61×/0.68× | 0.59×/0.72× | OOM/OOM | 0.61×/0.71× | 0.54×/0.72× |

GATv2 speedups for larger hidden size (heads $H = 8$, head dim $= 128$). Graphs with stable measurements using DGL are reported.

| **Latency speedup** | ogbn-arxiv | city-reviews | artnet-exp | city-roads-M | tolokers-2 | city-roads-L |
|---|---|---|---|---|---|---|
| Ours | 3.11×/1.16× | 4.96×/1.48× | 4.73×/1.83× | 2.19×/1.38× | 9.78×/2.18× | 2.01×/1.34× |
| PyG | 0.46×/0.54× | 0.31×/0.40× | 0.36×/0.47× | 0.42×/0.59× | 0.35×/0.44× | 0.42×/0.58× |
| **Peak-memory ratio** | ogbn-arxiv | city-reviews | artnet-exp | city-roads-M | tolokers-2 | city-roads-L |
| Ours | 6.73×/4.11× | 14.09×/8.62× | 10.38×/6.37× | 2.72×/1.76× | 69.74×/44.80× | 2.37×/1.53× |
| PyG | 0.55×/0.64× | 0.58×/0.66× | 0.57×/0.66× | 0.49×/0.62× | 0.60×/0.67× | 0.48×/0.61× |

single atomic operation to update both the reduction value and the source node index for backward. The backward pass stores the source node indices and implements gradient propagation as a sparse scatter-add.

The memory pattern of $O(M \cdot D)$ bytes loaded is preserved, since these reductions have very low arithmetic intensity and are fundamentally bandwidth-bound. Our design, therefore targets the dominant source of inefficiency: high-degree (`heavy`) nodes, where a one-block-per-node mapping underutilizes the GPU. By tiling their neighborhoods into chunks, we increase memory-level parallelism and improve load balance while keeping the same asymptotic I/O. Broader description is provided in Appendix D.1.

### 3.2. Attention-based layers: fused CSR attention for Graph Transformers and GATv2

Attention-based GNN layers typically materialize edge-wise logits or messages of size $O(M \cdot H)$ or $O(M \cdot H \cdot D)$ respectively, where $H$ is the number of heads. This increases HBM traffic and activation footprint. We design fused CSR kernels that perform score computation, softmax normaliza-

tion, and value aggregation in a single streaming pass over neighbors, avoiding explicit edge tensor materialization.

The key technique is online softmax: each warp maintains running statistics in registers and rescales the output accumulator incrementally as new neighbors are processed, enabling correct normalization without storing per-edge logits.

For the backward pass, following the FlashAttention-2 strategy (Dao, 2023), we store only compact per-node softmax statistics of size $O(N \cdot H)$, rather than materializing edge-wise attention weights of size $O(M \cdot H)$. This reduces peak activation memory and enables an efficient recomputation scheme in which we re-evaluate only the logits and derive attention weights on the fly. We provide fused kernels for both dot-product attention (Graph Transformers) and GATv2; implementation details are in Appendix D.2.

### 3.3. Block-sparse layouts for Tensor Core utilization

While our fused CSR kernels eliminate edge materialization, they rely on scalar cores and irregular gather patterns. To leverage Tensor Cores, we extend the block-sparse for-

mat from Li & Chandramowlishwaran (2025) to support weighted adjacency matrices, packing local neighborhoods into fixed-size 16×16 tiles compatible with WMMA instructions. This trades preprocessing and computation on padded entries for higher arithmetic intensity and better hardware utilization. We call this format Weighted Block Sparse (WSB) as it supports edge weights. This approach is most effective when row windows exhibit sufficient local density; for very sparse graphs, masked computations can dominate and eliminate the Tensor Core benefit (Xiang et al., 2025; Li & Chandramowlishwaran, 2025). In preliminary experiments, we found that backward latency for directed graphs is inefficient for block-sparse format, as it requires forming the transpose in the block-sparse format and the use of atomics, which largely negates the forward speedups. Therefore, in our WSB setting we compute the backward pass via a SpMM primitive. Implementation details can be found in Appendix D.3.

### 3.4. SpMM-based convolutions

Given the maturity of vendor libraries, we use cuSPARSE SpMM as the primary backend and focus on minimizing auxiliary overhead. We additionally cache graph-specific metadata (cuSPARSE descriptors, precomputed normalized weights, workspace buffers) and support autotuning across algorithm variants, amortizing setup costs across training iterations; we also optionally cache precomputed transposed adjacency matrix for backward pass and evaluate how this affects the runtime latency and memory footprint. Additionally, we provide Tensor Core SpMM kernels using the same weighted block-sparse format described in Section 3.3, enabling Tensor Core acceleration.

## 4. Experiments

**Setup.** We evaluate on Graph-Land (Bazhenov et al., 2025), a benchmark suite with diverse graph structures, and on `ogbn-arxiv` and `ogbn-products` from Open Graph Benchmark (Hu et al., 2021), and on standard citation networks (Cora, Citeseer, Pubmed) for compatibility with prior work (Yang et al., 2016). All experiments use full-batch configuration on a single NVIDIA A100 80GB SXM4. We benchmark GCN for SpMM-based layers, min-aggregation for reduction-based layers; for Attention-based layers, we focus on Graph Transformer and GATv2, which has been shown to be more expressive than the original GAT, while our optimization techniques remain broadly applicable to the latter architecture as well.

For SpMM-based and reduction-based convolutions, we sweep $D \in \{64, 128, 256, 512\}$. For GATv2, we sweep the number of heads $H \in \{2, 4, 8\}$ and the per-head dimension $D \in \{32, 64, 128, 256\}$ (so the total hidden size is $H \cdot D$). For Graph Transformer, we keep the total hidden

size fixed and vary $(H, D)$ pairs accordingly: for total dim 256 we use $(H, D) \in \{(2, 128), (4, 64), (8, 32)\}$, and for total dim 512 we use $(H, D) \in \{(2, 256), (4, 128), (8, 64)\}$. We evaluate METIS reordering with partition sizes $k \in \{128, 512, \ldots, 65536\}$ versus original ordering. Kernel parameters are autotuned per configuration. We measure the latency of forward/backward passes and their peak GPU memory footprint.

**Baseline backends.** We use DGL as our primary baseline, all speedups are reported relative to its behavior. For SpMM-based convolutions, we additionally compare against cuSPARSE SpMM with CSR format, TC-GNN[4], FuseGNN, PyG, and vanilla PyTorch sparse operations. For reduction-based convolutions, we additionally benchmark `min`-aggregation against cuGraph. For Graph Transformer layer, we compare against DF-GNN when available (We use DF-GNN-hyper variant as it's the only option with available backward implementation). We also evaluate our own Tensor Core kernels for SpMM and Graph Transformer layers (we label it `WSB/TC` backend).

Not all third-party systems could be evaluated on every dataset due to additional assumptions and engineering constraints. In particular, DF-GNN successfully executed only on 8 graphs in our benchmark suite. Fused3S could not build its required block-sparse representation for several large `web-topics` graph, and even when construction succeeded we observed illegal memory access leading to runtime failures. Moreover, the released Fused3S code does not provide a backward implementation for training, so we argue that it is not suitable for end-to-end alternative for training. Finally, cuGraph's attention operators require a bipartite graph to support layers with distinct left/right node features (as in GATv2) and to match the Graph Transformer interface; since our attention benchmarks use non-bipartite graphs, cuGraph attention baselines are not reported. We provide additional experiments in Appendix B.

### 4.1. Results

In this section, we summarize the main experimental findings and highlight the most representative performance results. Complete tables with per-dataset and per-configuration speedups, as well as full results regarding graph reordering, are provided in Appendix E.

**CuSparse is still hard to beat on SpMM-based convolutions.** Table 3 shows that cuSPARSE remains a strong baseline for SpMM-based layers. For instance, with cached descriptors, cuSPARSE achieves 2.30× speedup on forward pass `ogbn-arxiv` and 3.24×/1.51× on `city-roads-L`

---

[4]We observe that TC-GNN has incorrect backward pass for undirected graphs in GCN and it requires transposition of adjacency matrix in real-time during backward pass; we add this fix.

(forward/backward), while caching a precomputed $\tilde{A}^\top$ for backward increases backward speedups to $1.64\times$ and $1.74\times$ on the same graphs (and to $1.60\times$ on `web-traffic` and `web-topics`), at the cost of extra memory. WSB layout can be competitive in some regimes (e.g., $2.59\times$ forward on `tolokers-2` and $2.35\times$ on `city-roads-L`), but it is structure-dependent and incurs format overhead, so we treat it as an optional optimization rather than a universal replacement for CSR+cuSPARSE. Several specialized baselines are less robust at this scale: TC-GNN achieves $4.99\times/1.94\times$ on `city-roads-L` but shows near-zero throughput on `web-traffic` and runs out of memory on `ogbn-arxiv`.

**Reduction-based convolutions: degree-aware partitioning large forward gains.** As shown in Table 4, for the setup with hidden dim $D = 512$, our partitioned min-aggregation substantially accelerates forward pass compared to DGL across all graphs, reaching up to $8.31\times$ on `web-traffic` and $8.24\times$ on `web-topics`, while maintaining consistent backward improvements, e.g., $2.09\times$ on `ogbn-arxiv`). In addition, our implementation reduces peak memory compared to the DGL baseline for both forward and backward, with typical ratios in the $1.43$–$1.97\times$ range on the main graphs. The speedup is most noticable for graphs with high average degree, and is less pronounced on the sparse graphs with low average node degree like `city-roads-L`, where average degree is 1.96.

**Attention-based layers: fused CSR kernels improve latency and substantially reduce peak memory.** As shown in Table 2, for GATv2 our fused CSR kernels provide large forward speedups on all main graphs at the base configuration (e.g., $9.70 \times /1.71\times$ on `city-roads-L`, $6.68 \times /2.02\times$ on `pokec-regions`, and $5.59\times /1.89\times$ on `tolokers-2`), while backward is more mixed due to atomic-heavy gradient accumulation. Crucially, the fused formulation avoids materializing edge-wise messages and yields substantial peak-memory reductions (e.g., $32.59 \times /28.55\times$ on `tolokers-2` and $44.17 \times /31.71\times$ on `twitch-views`). For a larger hidden size (heads $H = 8$, head dim $= 128$) DGL failed to scale and we report other datasets. Our implementation remain stable, we observe even larger forward gains (up to $9.78\times$ on `tolokers-2`) while maintaining backward speedups of $1.16$–$2.18\times$; memory savings remain substantial (e.g., $69.74 \times /44.80\times$ on `tolokers-2` and $14.09 \times /8.62\times$ on `city-reviews`).

For Graph Transformer (Table 1), our fused CSR kernel provides consistent forward and backward gains on the shown setup (e.g., $1.27\times/1.24\times$ on `ogbn-arxiv` and $2.00\times/1.10\times$ on `tolokers-2`), while DF-GNN fails to run on some graphs in this configuration due to runtime errors (`N/A`). Our Tensor Core WSB layout kernel can further

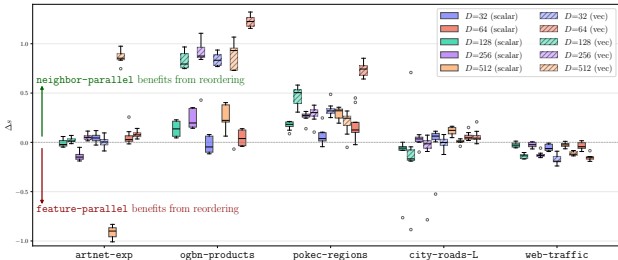

*Figure 1.* Reordering gap $\Delta s$ on selected datasets (`dot-aggr`). $\Delta s > 0$ means reordering helps `neighbor-parallel` (gather) more. We compare scalar vs. 16B vectorized loads (vec); boxes aggregate over partition sizes.

improve forward throughput on some graphs (e.g., $2.69\times$ on `tolokers-2` and $1.50\times$ on `pokec-regions`) but is not universally faster in backward due to scatter/atomic accumulation across tiles (e.g., $0.77\times$ on `ogbn-arxiv` and $0.70\times$ on `pokec-regions`).

**Graph reordering: benefits depend on the kernel access pattern and cache regime.** In our end-to-end benchmarks, graph reordering provides limited and non-universal gains. To isolate locality effects, we use a cache-sensitive microbenchmark (`dot-aggr`) with two warp-per-node implementations that have identical arithmetic but different memory access patterns: (i) `feature-parallel` (coalesced loads over features, neighbors scanned sequentially) and (ii) `neighbor-parallel` (gather-like loads over neighbors, each lane processes different neighbors). We use a second-pass to expose cache reuse and sweep hidden dimension $D$ and consider two access regimes: scalar loads vs. 16B vectorized loads. We measure the *reordering gap* $\Delta s = s_{\text{neighbor}} - s_{\text{feature}}$, where $s = t_{\text{orig}}/t_{\text{reordered}}$ is the speedup from reordering (higher is better). Positive $\Delta s$ means that reordering improves the gather-sensitive `neighbor-parallel` mapping more than `feature-parallel` one.

Figure 1 illustrates that this gap is strongly dataset- and regime-dependent (scalar vs. 16B vectorized loads). On the densest/high-degree graph in this subset, `ogbn-products` (avg. degree $\approx 50$), $\Delta s$ is consistently positive especially with vectorized loads, indicating that locality improvements primarily benefit the gather-dominated access pattern. For `pokec-regions` (avg. degree $\approx 19$), $\Delta s$ remains moderately positive across feature sizes, suggesting a smaller but still reliable advantage of reordering for gather-heavy mappings. In contrast, on the sparse road network `city-roads-L` (avg. degree $\approx 2$), $\Delta s$ stays close to zero (with occasional outliers), implying that the per-node working set is too small for reordering to create consistent cache reuse; performance is dominated by fixed per-node overheads rather than neighbor locality. A similar "low-degree" behavior is observed for `web-traffic` (avg. degree $\approx 4$), where $\Delta s$ is near-zero and slightly

*Table 3.* Results for SpMM with GCN as the operator (hidden dim $D = 512$); cuSPARSE reuses normalized values, CSR descriptor, workspace, and "+ $\tilde{A}^{\top}$" additionally caches precomputed $\tilde{A}^{\top}$ for backward.

| Backend | ogbn-arxiv | web-traffic | twitch-views | pokec-regions | tolokers-2 | city-roads-L |
|---|---|---|---|---|---|---|
| **Latency speedup** ($t_{\mathrm{DGL}}/t$), fwd/bwd | | | | | | |
| cuSPARSE | 2.30×/0.97× | 2.44×/0.94× | 1.26×/0.61× | 1.34×/0.66× | 2.33×/0.65× | 3.24×/1.51× |
| cuSPARSE + $\tilde{A}^{\top}$ | 2.27×/1.64× | 2.44×/1.60× | 1.26×/1.18× | 1.34×/1.28× | 2.46×/1.32× | 3.27×/1.74× |
| Ours (WSB/TC) | 1.56×/1.41× | 0.87×/1.35× | 1.80×/1.07× | 1.99×/1.20× | 2.59×/0.79× | 2.35×/1.55× |
| TC-GNN | 2.63×/OOM | <0.01×/<0.01× | 0.08×/0.09× | 1.76×/2.05× | 0.71×/0.50× | 4.99×/1.94× |
| FuseGNN | 0.27×/1.44× | 0.19×/0.09× | 0.32×/0.37× | 0.92×/1.20× | 0.35×/0.58× | 0.53×/1.80× |
| PyG | 0.20×/0.18× | OOM/OOM | 0.06×/OOM | OOM/OOM | 0.11×/0.07× | 0.36×/0.36× |
| **Peak-memory ratio** ($\mathrm{mem}_{\mathrm{DGL}}/\mathrm{mem}$), fwd/bwd | | | | | | |
| cuSPARSE | 0.49×/0.54× | 1.47×/1.14× | 0.64×/0.66× | 1.92×/1.23× | 0.06×/0.08× | 1.82×/1.19× |
| cuSPARSE + $\tilde{A}^{\top}$ | 0.23×/0.29× | 1.42×/1.12× | 0.30×/0.35× | 1.50×/1.09× | 0.02×/0.03× | 1.80×/1.18× |
| Ours (WSB/TC) | 1.49×/0.99× | 1.31×/1.00× | 1.20×/0.96× | 1.44×/0.99× | 1.24×/0.98× | 1.51×/1.00× |
| TC-GNN | 1.93×/OOM | 1.48×/1.15× | 1.36×/0.84× | 1.78×/1.45× | 1.92×/1.17× | 2.38×/1.48× |
| FuseGNN | 0.21×/0.20× | 0.11×/0.14× | 0.27×/0.23× | 0.22×/0.22× | 0.22×/0.22× | 0.22×/0.21× |
| PyG | 0.01×/0.01× | OOM/OOM | 0.03×/OOM | OOM/OOM | 0.01×/0.01× | 0.12×/0.12× |

*Table 4.* Results for reduction-based convolutions (with min-aggregation as the operator) on selected graphs (hidden dim $D = 512$).

| Backend | ogbn-arxiv | web-traffic | web-topics | pokec-regions | tolokers-2 | city-roads-L |
|---|---|---|---|---|---|---|
| **Latency speedup** ($t_{\mathrm{DGL}}/t$), fwd/bwd | | | | | | |
| Ours | 4.84×/2.09× | 8.31×/1.52× | 8.24×/1.50× | 2.59×/1.41× | 2.54×/1.32× | 4.14×/1.63× |
| cuGraph | 2.87×/1.45× | 0.06×/2.17× | 0.06×/2.11× | 1.42×/1.14× | 0.44×/0.87× | 3.50×/1.38× |
| **Peak-memory ratio** ($\mathrm{mem}_{\mathrm{DGL}}/\mathrm{mem}$), fwd/bwd | | | | | | |
| Ours | 1.71×/1.37× | 1.43×/1.28× | 1.57×/1.33× | 1.77×/1.40× | 1.97×/1.51× | 1.66×/1.35× |
| cuGraph | 1.19×/1.14× | 1.19×/1.13× | 1.24×/1.15× | 1.27×/1.16× | 1.19×/1.12× | 1.28×/1.17× |

negative in many settings, meaning that reordering is not a universally positive optimization and can even favor the feature-parallel/coalesced mapping. Finally, artnet-exp (avg. degree $\approx 11$) highlights sensitivity to the execution regime: the sign/magnitude of $\Delta s$ can flip between scalar and vectorized loads, consistent with the idea that reducing instruction overhead (vectorization) makes memory locality a more prominent bottleneck and thereby changes which mapping benefits more from reordering.

## 5. Conclusion and Future Work

We study GNN performance from an IO- and arithmetic-intensity–centric perspective and show that widely used layers cluster into three kernel families with distinct bottlenecks: SpMM-based convolutions, reduction-based aggregations, and attention-based layers. Building on this taxonomy, we develop GPU implementations that reduce redundant HBM traffic, improve locality, and remain robust on realistic graphs. In particular, our fused attention kernels avoid materializing edge-wise intermediates and substantially reduce peak activation memory in both forward and backward, while degree-aware tiling improves throughput for reduction operators under heavy-tailed degree distributions. For SpMM-based layers, we highlight that vendor primitives remain a strong baseline: with graph-aware

caching and a cached transpose for backward, cuSPARSE can match or outperform specialized systems in our setting. We also explore a block-sparse Tensor-Core path that can improve forward throughput on locally dense graphs; however, for directed graphs its training-time backward can be less efficient due to scatter/atomic accumulation, which can dominate the gains. Finally, we revisit graph reordering and show that its impact is *not* universal: it depends on the kernel's parallelization strategy, memory access pattern and effective working set, with gather-sensitive neighbor-parallel mappings benefiting more consistently than feature-parallel ones.

As a next step, an adaptive runtime/autotuner that selects between backends (e.g., cuSPARSE vs. custom kernels, CSR-fused vs. block-sparse Tensor-Core paths) based on graph statistics (degree skew, local density, feature size) would make performance more predictable and reduce manual tuning. Another promising direction is improving the training-time efficiency of Tensor-Core variants by reducing scatter/atomic pressure in backward, especially for directed graphs. Finally, we release our implementations as drop-in replacements with a lightweight dependency stack (CUDA/PyTorch and optional cuSPARSE), enabling reproducible, hardware-aware GNN acceleration in practice.

## Impact Statement

This paper presents work whose goal is to advance the field of Machine Learning. There are many potential societal consequences of our work, none which we feel must be specifically highlighted here.

## Acknowledgments

We thank Liudmila Prokhorenkova and Oleg Platonov for early discussions and feedback on the research direction. We thank Dmitry Eremeev for help with testing an early version of the library. We also thank Roman Garipov, Ilya Drobyshevskiy, Ilia Sudakov, Timofei Senin, and Aleksandr Matosyan for valuable feedback on the paper and helpful discussions on the code implementation and future research directions.

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

# A. Code availability and Reproducibility

Experimental details are described in Section 4. We use 10 warmup iterations and 30 measurement iterations and then average the obtained metrics. We autotune the kernels on every setup and report the metrics obtained from the best performing config.

We release the source code in our repository: github.com/yandex-research/On-Efficient-Scaling-Of-GNNs.

# B. Additional Experiments

### B.1. Degree-Aware Warp Allocation for Attention Kernels

The degree-aware partitioning introduced in Section 3.1 applies naturally to attention kernels. When intra-graph degree variance is high, thread blocks assigned to high-degree nodes run substantially longer than those for low-degree nodes, creating tail latency and SM underutilization. We test this extension on our fused attention kernels: destination nodes are partitioned into light and heavy buckets, and each bucket launches with an independently configured warp count. Table 5 reports the speedup of this variant over the original degree-agnostic attention kernel for Graph Transformer and GATv2 (head dim $= 512$, $H = 4$ heads). The speedup correlates with degree skewness: `web-traffic` (skewness 738) and `ogbn-arxiv` (skewness 111) yield the highest gains, while road-network graphs with near-uniform degree distributions show minimal improvement. GT backward benefits less, as `atomicAdd` on $dK$ introduces serialization regardless of warp count.

*Table 5.* Speedup of degree-aware warp allocation over the degree-agnostic attention kernel (GT and GATv2, head dim $= 512$, $H = 4$), fwd/bwd.

| Dataset | Skewness | GT fwd | GT bwd | GATv2 fwd | GATv2 bwd |
|---|---|---|---|---|---|
| web-traffic | 738.1 | 5.83× | 6.34× | 5.45× | 7.34× |
| ogbn-arxiv | 110.8 | 3.72× | 1.09× | 3.91× | 2.49× |
| pokec-regions | 71.8 | 1.07× | 1.02× | 1.13× | 1.10× |
| twitch-views | 42.4 | 2.10× | 1.45× | 2.23× | 2.08× |
| city-reviews | 35.0 | 2.07× | 1.40× | 2.24× | 1.78× |
| ogbn-products | 17.5 | 1.07× | 1.02× | 1.19× | 1.11× |
| avazu-ctr | 9.4 | 2.48× | 1.37× | 2.30× | 2.04× |
| hm-categories | 7.3 | 1.79× | 1.26× | 1.62× | 1.65× |

### B.2. Direction-Aware Backward Kernels for Undirected Graphs

For undirected graphs, the backward pass can traverse the same CSR structure used in the forward pass, eliminating the scatter/atomic operations over a transposed adjacency that dominate the directed backward for WSB. We develop dedicated direction-aware backward kernels for both the CSR-fused and WSB backends that exploit undirectedness when available. Table 6 reports the speedup over the directed backward baseline (head dim $= 128$, $H = 4$ heads). CSR-based GT backward gains 5–21% from this optimization, as eliminating the transposed traversal improves locality. WSB achieves a median $7.4\times$ speedup, since removing scatter/atomic contention across row windows fully recovers the Tensor Core advantage in backward that is lost for directed graphs. For GATv2 on undirected graphs, we additionally fuse the gradient computation with respect to source and destination features into a single kernel pass, eliminating one full graph traversal. All backends produce numerically identical gradients to the directed implementation.

### B.3. WSB Preprocessing Cost and Amortization

WSB format construction incurs a one-time preprocessing cost per graph structure that must be amortized across training iterations. This cost can also be eliminated entirely by serializing the WSB representation to disk for reuse across runs. Table 7 reports WSB construction time, per-epoch training latency relative to DGL, and the resulting break-even epoch count (GT, 4 layers, head dim $= 128$, $H = 4$ heads). On locally dense graphs such as `hm-prices` (break-even: 65 epochs) and `avazu-ctr` (break-even: 159 epochs) WSB amortizes quickly and becomes preferable for realistic training runs. On 9 of 16 graphs, however, the CSR-fused backend is faster per epoch and the break-even exceeds 1000 epochs, consistent with WSB being an optional acceleration path for locally dense graphs rather than a universal replacement.

*Table 6.* Backward speedup of direction-aware undirected kernels over the directed baseline (head dim $= 128$, $H = 4$). GT (CUDA) and GATv2 (CUDA) use CSR-fused kernels; GT (WSB) uses the block-sparse Tensor Core path. Graphs are transformed to undirected.

| Dataset | Avg. deg | GT (CUDA) | GATv2 (CUDA) | GT (WSB) |
|---|---|---|---|---|
| avazu-ctr | 289 | 1.21× | 1.12× | 8.59× |
| hm-categories | 461 | 1.19× | 1.08× | 8.16× |
| tolokers-2 | 89 | 1.01× | 1.05× | 7.57× |
| twitch-views | 81 | 1.03× | 1.01× | 9.36× |
| ogbn-products | 51 | 1.11× | 1.14× | 7.03× |
| pokec-regions | 28 | 1.11× | 0.97× | 6.76× |
| ogbn-arxiv | 14 | 1.13× | 1.09× | 7.73× |
| web-traffic | 9 | 1.06× | 1.21× | 10.74× |
| city-roads-L | 4 | 1.10× | 1.01× | 2.75× |

*Table 7.* WSB amortization (GT, 4 layers, head dim $= 128$, $H = 4$). Break-even is the number of training epochs needed to recover the one-time WSB construction cost.

| Dataset | WSB cost (s) | DGL (ms/ep) | WSB (ms/ep) | Speedup | Break-even (ep.) |
|---|---|---|---|---|---|
| hm-prices | 16.1 | 731.2 | 484.4 | 1.51× | 65 |
| avazu-ctr | 14.9 | 825.5 | 731.9 | 1.13× | 159 |
| city-roads-L | 4.4 | 296.3 | 285.5 | 1.04× | 404 |
| city-roads-M | 4.3 | 145.5 | 139.4 | 1.04× | 693 |

*Table 8.* Dataset statistics (computed on our processed graphs).

| Dataset | #nodes | #edges | density | avg. degree |
|---|---|---|---|---|
| hm-categories | 46,563 | 21,461,990 | $9.90 \times 10^{-3}$ | 460.92 |
| tolokers-2 | 11,758 | 1,038,000 | $7.51 \times 10^{-3}$ | 88.28 |
| avazu-ctr | 76,269 | 21,968,154 | $3.78 \times 10^{-3}$ | 288.04 |
| cora | 2,708 | 10,556 | $1.44 \times 10^{-3}$ | 3.90 |
| citeseer | 3,327 | 9,104 | $8.23 \times 10^{-4}$ | 2.74 |
| twitch-views | 168,114 | 13,595,114 | $4.81 \times 10^{-4}$ | 80.87 |
| pubmed | 19,717 | 88,648 | $2.28 \times 10^{-4}$ | 4.50 |
| artnet-exp | 50,405 | 560,696 | $2.21 \times 10^{-4}$ | 11.12 |
| city-reviews | 148,801 | 2,330,830 | $1.05 \times 10^{-4}$ | 15.66 |
| city-roads-M | 57,073 | 132,570 | $4.07 \times 10^{-5}$ | 2.32 |
| ogbn-arxiv | 169,343 | 1,166,243 | $4.07 \times 10^{-5}$ | 6.89 |
| ogbn-products | 2,449,029 | 123,718,280 | $2.06 \times 10^{-5}$ | 50.52 |
| city-roads-L | 142,257 | 279,061 | $1.38 \times 10^{-5}$ | 1.96 |
| pokec-regions | 1,632,803 | 30,622,564 | $1.15 \times 10^{-5}$ | 18.75 |
| web-traffic | 2,890,331 | 12,895,369 | $1.54 \times 10^{-6}$ | 4.46 |

## C. Datasets

### C.1. Overview.

Our evaluation suite combines (i) industrial, attribute-rich graphs from GraphLand (Bazhenov et al., 2025) and (ii) widely used citation-network benchmarks (Cora/Citeseer/Pubmed) from the Planetoid setup (Yang et al., 2016), as well as (iii) OGB node-property graphs (ogbn-arxiv, ogbn-products) (Hu et al., 2021). Table 8 reports basic structural statistics computed on our processed graph representations.

**GraphLand datasets (Bazhenov et al., 2025).** GraphLand is a collection of graphs from realistic industrial domains. We use the following datasets:

**web-fraud, web-topics, web-traffic.** Datasets are presented as large directed web graph (graph is shared): nodes are websites, edges represent observed user transitions between websites. We use web-traffic in our experiments.

The tasks differ: fraud detection (binary, imbalanced), topic classification (multiclass), and traffic prediction (regression).

*Table 9.* Memory footprint (MB) of graph representations required by different backends. Values are measured for the cached/preprocessed graph objects used by each backend; `N/A` denotes unavailable measurements due to Fused3S failed to build block-sparse representation.

| Dataset | DGL | PyG | cuGraph | cuSPARSE | cuSPARSE + $\bar{A}^T$ | Torch sparse | Ours (CSR fused) | TC-GNN | DF-GNN | Fused3S | Ours (WSB/TC) |
|---|---|---|---|---|---|---|---|---|---|---|---|
| artnet-exp | 24.28 | 24.28 | 59.63 | 17.48 | 20.00 | 46.62 | 21.11 | 21.57 | 26.03 | 34.97 | 49.72 |
| avazu-ctr | 413.17 | 413.17 | 1681.86 | 252.57 | 336.95 | 1681.86 | 245.86 | 329.11 | 497.38 | 1343.80 | 1681.86 |
| citeseer | 47.22 | 47.22 | 47.96 | 47.09 | 47.15 | 47.30 | 47.17 | 47.15 | 47.23 | 47.09 | 47.82 |
| city-reviews | 155.40 | 155.56 | 298.91 | 127.59 | 137.62 | 189.67 | 138.18 | 144.84 | 163.16 | 143.78 | 266.75 |
| city-roads-L | 120.84 | 120.84 | 147.46 | 116.56 | 118.71 | 124.24 | 119.25 | 118.18 | 120.82 | 116.12 | 134.01 |
| city-roads-M | 18.30 | 18.30 | 29.59 | 16.35 | 17.29 | 19.46 | 17.51 | 17.15 | 8.37 | 16.12 | 23.78 |
| cora | 15.04 | 15.04 | 15.82 | 14.89 | 14.96 | 15.11 | 14.97 | 14.97 | 15.06 | 14.90 | 15.63 |
| hm-categories | 350.69 | 350.68 | 1640.98 | 246.32 | 328.55 | 1640.98 | 187.12 | 268.68 | 432.84 | 1311.87 | 1640.98 |
| ogbn-arxiv | 124.16 | 125.45 | 224.84 | 184.36 | 203.72 | 224.84 | 197.98 | 147.29 | 208.97 | 156.08 | 224.84 |
| ogbn-products | 4791.77 | 4811.92 | 13996.19 | 10582.25 | 12072.83 | 13996.19 | 11582.20 | 7599.92 | 13771.59 | 8511.87 | 13996.19 |
| pokec-regions | 858.82 | 858.38 | 2717.03 | 496.16 | 626.39 | 2461.85 | 632.62 | 723.00 | 962.47 | 1871.63 | 2461.85 |
| pubmed | 39.86 | 39.86 | 46.22 | 38.70 | 39.19 | 40.43 | 39.26 | 39.30 | 40.05 | 38.77 | 45.03 |
| tolokers-2 | 17.00 | 16.99 | 80.09 | 12.06 | 16.11 | 80.09 | 9.12 | 12.90 | 21.07 | 63.92 | 80.09 |
| twitch-views | 229.77 | 228.48 | 1053.04 | 159.14 | 212.29 | 1053.04 | 124.91 | 174.45 | 279.05 | 831.86 | 1053.04 |
| web-traffic | 13281.56 | 13281.56 | 14206.88 | 13112.91 | 13184.15 | 13363.83 | 13197.13 | 13202.41 | 13314.69 | N/A | 13860.13 |

**artnet-views, artnet-exp.** Both share the same (friendship) social graph of art creators: nodes are users, edges connect friends. Tasks: predict user views (regression) and predict explicit-content creators (binary classification). We use `artnet-views` in our experiments.

**city-roads-M, city-roads-L.** Directed road-network graphs: nodes are road segments, edges indicate feasible transitions between segments under traffic rules. The task is to predict average travel speed for each segment at a given time (regression).

**city-reviews.** User graph from a review service: nodes are users, edges connect users who often review the same organizations. Task: detect fraudulent reviewers (binary classification).

**avazu-ctr.** Graph built from Avazu CTR competition data: nodes are devices, undirected edges connect devices that often visit the same websites. Task: predict device CTR (regression).

**hm-categories, hm-prices.** Derived from shared H&M co-purchasing graph: nodes are products, undirected edges connect products frequently bought together. Tasks: category prediction (multiclass) and price prediction (regression). We use `hm-categories` in our experiments.

**pokec-regions.** Pokec online social network: nodes are users, directed edges correspond to "friend" markings. Task: region prediction (extreme multiclass) from user-profile features.

**twitch-views.** Twitch social network: nodes are users, undirected edges connect mutually following users. Task: predict user views (regression).

**tolokers-2.** Worker network from Toloka: nodes are workers (tolokers), undirected edges connect workers who worked on the same task. Task: fraud detection (banned vs not banned) with an extended feature set.

**OGB datasets (Hu et al., 2021).** `ogbn-arxiv` is a directed citation graph of arXiv papers with a node-property prediction task over paper categories/fields (node classification). `ogbn-products` is an Amazon product co-purchasing graph with a node-property prediction task over product categories (node classification).

**Citation-networks (Yang et al., 2016).** `cora`, `citeseer`, and `pubmed` are classic citation networks used in semi-supervised node classification. In the Planetoid setup, nodes are documents with bag-of-words features and edges represent citation links (symmetrized), and the goal is to classify documents into classes. We treat them as small-scale sanity checks rather than representative scaling benchmarks.

**Graph representations and preprocessing overhead.**

**Graph formats and representation overhead.** Different backends require different graph encodings and preprocessing, which affects the GPU memory footprint even before accounting for node features and intermediate activations. Table 9 reports the memory (MB) of the cached/preprocessed graph objects used by each backend. CSR-based backends (DGL, PyG, cuSPARSE, and our CSR-fused kernels) store row offsets and column indices, optionally with edge values, and therefore have comparable representation sizes across most datasets. Caching a transposed adjacency for backward in cuSPARSE ("cuSPARSE + $\tilde{A}^\top$") increases representation memory as expected, e.g., from $184.36$MB to $203.72$MB on `ogbn-arxiv`, and from $496.16$MB to $626.39$MB on `pokec-regions`.

Our Tensor-Core path uses a weighted block-sparse (WSB) layout, which trades extra structure memory (due to blocking/padding and tile metadata) for higher arithmetic intensity and Tensor Core utilization. Consequently, WSB can be substantially more expensive than CSR on dense or locally dense graphs, e.g., `hm-prices` ($246.32$MB in cuSPARSE vs. $1640.98$MB in Ours (WSB/TC)) and `tolokers-2` ($12.06$MB vs. $80.09$MB). On very large sparse graphs, the overhead is more moderate relative to the already large CSR footprint, e.g., `web-traffic` ($13112.91$MB in cuSPARSE vs. $13860.13$MB in Ours (WSB/TC)). Finally, for some datasets Fused3S fails to build its required block-sparse representation (marked as `N/A` in Table 9), highlighting that format requirements can affect not only memory but also the robustness of the evaluation pipeline.

# D. IO aware kernels (extended)

## D.1. Degree-aware reduction-based convolutions aggregation

On GPUs, this operator is typically bandwidth-bound: the dominant cost is streaming neighbor features from HBM, while the per-element compute is limited to comparisons and updates. A major practical challenge is the heavy-tailed degree distribution of real graphs: a small fraction of high-degree (`heavy`) nodes can dominate the runtime if each node is processed with a uniform mapping such as one block (or warp) per destination node (Newman, 2003; Albert & Barabási, 2002).

**Degree-adaptive partitioning.** To address degree skew and improve load balance, we partition destination nodes into `light` and `heavy` subsets using a graph-specific degree threshold $\gamma$; we describe the values and the partitioning mechanism later. Light nodes are processed with a standard feature-parallel kernel, while heavy nodes are handled by a 2D tiling scheme that parallelizes across edge chunks and merges partial reductions.

**Light nodes: feature-parallel reduction.** For light nodes, we adopt a simple and efficient mapping where each CUDA block is assigned to one destination node and threads iterate over the feature dimension. Concretely, each thread processes a strided subset of features, and for each feature performs a sequential scan over the neighbor list to update the running minimum and the corresponding source index. This mapping keeps per-thread state in registers and avoids inter-thread synchronization, which is effective when degrees are moderate and the neighbor scan does not dominate end-to-end runtime.

**Heavy nodes: 2D edge tiling and parallel reduction.** For heavy nodes, the neighbor scan becomes the bottleneck, and the one-block-per-node strategy underutilizes the GPU due to insufficient parallelism and poor tail latency. We therefore introduce a 2D grid decomposition: `blockIdx.x` selects a heavy destination node, and `blockIdx.y` selects an edge chunk within its adjacency list. Each block processes a contiguous chunk of `EDGES_PER_BLOCK` neighbors, computes a *local* minimum (and argmin) for each feature, and then merges these partial results into a single per-node output. The chunk size `EDGES_PER_BLOCK` and the number of warps per block are exposed as tuning knobs, trading off additional merge overhead against increased parallelism for high-degree nodes.

**Merging partial minima with atomic (value, index) reduction.** To combine partial results across edge chunks without block-level synchronization across `blockIdx.y`, we use an atomic reduction over packed `(value, index)` pairs. We map the floating-point value to an order-preserving integer key and pack it together with the argmin index into a 64-bit word, enabling a single `atomicMin` to update both the minimum value and its source node. This atomic merge produces the same result as a full reduction over all neighbors while allowing independent edge chunks to execute in parallel.

**Decoupled materialization via unpacking.** For heavy nodes, partial reductions are accumulated into an intermediate packed buffer of shape $|\mathcal{V}_H| \times D$. A dedicated unpack kernel then converts the packed representation back into the output

feature tensor and the argmin tensor. While this introduces an extra linear pass over memory, it is fully coalesced and keeps the main reduction kernel simple and highly parallel, avoiding costly global synchronization.

**Backward pass: sparse scatter using saved argmin.** In the backward pass, the gradient of a `min`/`max` reduction is non-zero only along the selected argmin/argmax source. We therefore save the argmin indices during forward and implement backward as a sparse scatter-add: for each $(v, f)$ we add $\nabla \text{out}[v, f]$ to $\nabla X[\text{argmin}[v, f], f]$ using atomic adds. Storing argmin avoids recomputing the reduction structure in backward and reduces redundant work, analogous in spirit to caching normalization statistics in fused attention kernels to minimize extra operations during backpropagation.

**IO and compute characteristics.** Both the vanilla and our degree-aware implementations must read neighbor features from HBM. However, the vanilla mapping suffers from poor load balance, as long neighbor scans for a few nodes serialize execution. For a graph with $M$ edges and feature dimension $D$, this corresponds to reading approximately $M \cdot D \cdot sizeof(\text{dtype})$ bytes from HBM in the forward pass, independent of the execution mapping. The degree-aware partitioning does not change this bound. It parallelizes the neighbor scans of high-degree nodes across multiple thread blocks, improving concurrency and reducing tail latency. The additional merge and unpack steps introduce extra memory accesses, but these are fully coalesced and linear in memory layout, resulting in negligible overhead compared to the neighbor feature reads.

**Degree threshold and autotuning parameters.** Destination nodes are classified as `heavy` if their degree exceeds a graph-specific defined as a quantile of the degree distribution. In our experiments, we consider $\gamma =$`huge_degree_threshold_quantile` $\in [0.99, 0.999]$, corresponding to the top $1\%$ and $0.1\%$ of nodes by degree, respectively. We additionally include a baseline setting with no partitioning, in which all nodes are marked as `light`. Kernel launch parameters are selected via an autotuner that grid-searches a bounded parameters space. We search over `warps_per_block` $\in [1, 4, 16, 32]$ and `edges_per_block_heavy_nodes` $\in [32, 64, 128, 512]$ for heavy nodes. These parameters trade off feature-level parallelism, chunk-level parallelism, and the degree of serialization during the merge step. For each graph and feature dimension, the autotuner selects the configuration with the lowest measured latency.

**Order-preserving packing for atomic reductions.** To enable a correct `atomicMin` reduction over floating-point values, we transform IEEE-754 floats into an order-preserving unsigned integer representation. The bit pattern of each value is remapped such that the standard unsigned integer comparison reflects the total ordering of floating-point numbers: negative values are bitwise inverted, while non-negative values have their sign bit set. This transformation ensures that a smaller floating-point value always corresponds to a smaller integer key under unsigned comparison. The resulting 32-bit key is packed together with the source node index into a single 64-bit word, with the value key occupying the most significant bits. Applying `atomicMin` to this packed representation therefore performs a lexicographic reduction: values are compared first, and source indices are used to break ties deterministically. This enables a correct global `min` reduction across edge chunks without cross-block synchronization.

## D.2. Attention-based layers: fused CSR attention for Graph Transformers and GATv2

For Graph Transformers, our measurements include the full attention block comprising layer normalization, QKV projection, and the attention mechanism itself.

Throughout this section, $N$ denotes the number of nodes, $M$ the number of edges, $H$ the number of heads, and $D$ the per-head feature dimension.

Attention-based GNN layers (Graph Transformers and GAT-family) are particularly challenging on GPUs because a straightforward message-passing implementation typically materializes edge-wise attention logits and/or messages of size $O(M \cdot H)$ or $O(M \cdot H \cdot D)$ before reducing them at destination nodes. This intermediate materialization increases HBM traffic and peak activation footprint, and it amplifies kernel-launch overhead due to alternating edge-parallel (SDDMM-like) and node-parallel (SpMM-like) phases. To address this, we design fused, IO-aware CUDA kernels that stream over neighbors in CSR format and perform *score computation, softmax normalization, and value aggregation* within a single kernel, avoiding explicit edge tensor materialization.

**Fused forward in CSR (Graph Transformer).** We implement a fused Graph Transformer kernel over CSR neighborhoods, parameterized by node $i$ and head $h$. Each CUDA block is assigned to one pair $(i, h)$ (2D grid over nodes and heads), and

internally uses a fixed number of warps (`kWarpsPerBlock=4`) to parallelize over the neighbor list. At the start of the kernel, we cooperatively stage the key vector associated with the destination node into shared memory (e.g., `k_shared`), enabling reuse across all neighbor interactions. Each warp iterates over a disjoint strided subset of neighbors and computes the attention score $e_{ij} = \langle K_i, Q_j \rangle / \sqrt{D}$ using warp-striped loads over the feature dimension and warp-level reductions. Partial results produced by different warps are merged via a cross-warp reduction to obtain the final normalized output. As in FlashAttention-style kernels, we additionally store `logsumexp[i,h]` for an efficient backward pass.

**Fused forward in CSR (GATv2).** For GATv2, the attention logit is computed as $e_{ij} = \mathbf{a}^\top \sigma(\ell_i + r_j)$, where $\sigma$ is LeakyReLU and $\ell_i, r_j \in \mathbb{R}^D$ are precomputed per-node features. We follow the same IO-aware pattern as in the Graph Transformer kernel: each block handles one $(i, h)$ pair, stages $\ell_i$ into shared memory, and streams over neighbors in CSR order. Within the neighbor loop, threads cooperatively compute $e_{ij}$ using vectorized `float4` loads and warp-level reductions, and aggregate contributions from neighbors in a single streaming pass. We store `logsumexp[i,h]` to avoid materializing attention weights and to support an efficient backward pass.

**Online softmax and fused aggregation.** To avoid storing logits or attention weights, the kernel maintains per-warp online softmax state in registers (`max_val`, `sum_exp`) and updates it incrementally for each processed neighbor. When the running maximum changes, previously accumulated quantities are rescaled by a correction factor, and the weighted value accumulator is updated in the same streaming pass. Concretely, each warp keeps an output accumulator in registers (warp-striped scalars over the feature dimension) and performs a fused update of the form $\mathbf{o} \leftarrow \mathbf{o} \cdot \mathrm{corr} + \exp(e_{ij} - m)\,\mathbf{v}_j$, where $\mathrm{corr} = \exp(m_{\mathrm{old}} - m_{\mathrm{new}})$ implements the online softmax rescaling. This design fuses (i) score computation, (ii) softmax normalization, and (iii) value aggregation, thus eliminating $O(M \cdot H)$ intermediate logits and $O(M \cdot H \cdot D)$ edge messages from HBM.

**Cross-warp merge and normalization (Graph Transformer).** In the Graph Transformer forward kernel, different warps process disjoint neighbor subsets, and each warp produces a partial online-softmax state and a partial accumulator. We store these per-warp partial results in shared memory (`warp_out`, `warp_max`, `warp_sum`), and perform a cross-warp merge in warp 0. The merge first computes the global maximum across warps and then the globally normalized denominator by reweighting each warp's local sum via $\exp(m_w - m_{\mathrm{global}})$. The same scale factors are reused to combine partial output accumulators before applying the final normalization. The kernel writes the final output $O[i, h, :]$ and additionally stores the per-node log-sum-exp statistic `logsumexp[i,h]` for use in the backward pass. We also provide a set of specialized kernels for common head dimensions (`D=32,64,128,256`) via compile-time specialization, which enables static unrolling and reduces control-flow overhead, while falling back to a runtime-$D$ variant when needed.

**Backward pass: caching statistics to reduce redundant work (Graph Transformer).** In the backward pass, a naive implementation would either store all per-edge attention weights (prohibitively large) or recompute normalization terms multiple times. Following the same principle used in fused attention kernels, we cache compact per-node statistics during forward to reduce backward overhead. Specifically, we reuse the saved `logsumexp` and compute an additional scalar $\Delta[i, h] = \langle O[i, h, :], dO[i, h, :] \rangle$ in a dedicated kernel using vectorized loads. The main backward kernel iterates over the transposed graph ($\mathrm{CSR}^\top$) to process incoming edges for each source node $j$ and accumulates gradients by recomputing $\alpha_{ij} = \exp(e_{ij} - L_i)$ from the saved `logsumexp` $L_i$. Gradients for $Q$ and $V$ at node $j$ are accumulated locally, while $dK$ at node $i$ is updated via atomic adds along incoming edges, matching the $\mathrm{CSR}^\top$ traversal.

**Backward pass: fused two-kernel structure (GATv2).** For GATv2, we similarly reuse the saved `logsumexp` to recompute $\alpha_{ij}$ on demand. We structure backward into two fused kernels: an `AL` kernel that computes gradients w.r.t. the attention parameters and left features and also computes an auxiliary scalar statistic $G_{i,h} = \sum_j \alpha_{ij} \langle dh_i, r_j \rangle$, and an `R` kernel that processes $\mathrm{CSR}^\top$ to compute gradients w.r.t. the right features. Both kernels use shared-memory staging of per-node feature blocks (and `float4` vectorization) to increase reuse within a warp while streaming over edges. Finally, since the attention vector gradient is accumulated per node as $[N, H, D]$, we perform a separate reduction over the node dimension to obtain the final $[H, D]$ gradient; this reduction is implemented as a **tiled** kernel that processes node chunks and feature tiles to balance parallelism and memory efficiency.

**Attention vector gradient reduction (GATv2).** The gradient w.r.t. the attention vector $\mathbf{a}$, accumulated per-node in the AL kernel as a $[N, H, D]$ tensor, must be reduced over the node dimension to obtain the final $[H, D]$ gradient. We implement this as a tiled 2D reduction kernel where each thread block processes a chunk of nodes (controlled by

`grad_A_reduce_row_chunk_size`, typically 512–2048) and a feature tile of size 32 (one warp) for a specific head $h$. Threads cooperatively load tiles of grad_a$[n, h, d]$ into shared memory in a transposed layout, perform warp-level reductions over the node dimension, and accumulate partial sums across multiple node chunks in registers. Final values are written via atomic adds to handle contributions from different blocks processing the same $(h, d)$ pair, trading off reduction parallelism against atomic contention.

**Parameters swept in experiments (GATv2 and Graph Transformer).** We evaluate performance across model and kernel parameters that directly affect compute intensity and memory behavior. On the model side, we sweep $D \in [256, 512]$ and $H \in [2, 4, 8]$, which control the per-head and total feature width and thus scale both arithmetic work and feature traffic. For GATv2, we additionally sweep the reduction chunk size used to aggregate the attention-vector gradient over nodes, `grad_A_reduce_row_chunk_size`$\in [16, 32, 64, 128, 256, 512, 1024, 2048]$, which trades off reduction parallelism, shared-memory reuse, and atomic update pressure in the $[N, H, D] \to [H, D]$ stage. We keep graph reordering disabled in this evaluation (`graph_reordering_partition_size=−1`) to isolate kernel-level behavior. All experiments are run over multiple real-world graphs (GraphLand (Bazhenov et al., 2025) and standard benchmarks) to cover diverse degree distributions that interact with warp-level CSR streaming.

**IO and arithmetic-intensity analysis (vanilla vs. fused).** We quantify the impact of fusion using an IO-centric view similar in spirit to roofline analysis. Let $N$ be the number of nodes, $M$ the number of edges, $H$ the number of heads, and $D$ the head dimension. We count HBM traffic (global memory reads+writes) and use $b$ bytes per scalar (for our current fp32 implementation $b=4$). We omit the $O(M)$ index traffic of `row_ptr`/`col_idx` for clarity, as it is typically dominated by feature reads when $D$ is moderate or large.

**Vanilla decompositions.** A common baseline decomposes attention into separate phases: (i) edge-wise score computation (SDDMM-like), (ii) row-wise softmax over each destination neighborhood, and (iii) weighted aggregation (SpMM-like). This requires materializing at least the edge attention weights $\alpha \in \mathbb{R}^{M \times H}$ (and often also logits), incurring additional HBM traffic beyond the unavoidable feature reads. Ignoring constant factors, the forward-pass HBM bytes for such a decomposition scale as

$$B_{\text{vanilla,fwd}} \approx \underbrace{3MHDb}_{\text{read } Q/K/V \text{ (GT) or } \ell/r/\mathbf{a} \text{ (GATv2)}} + \underbrace{O(MHb)}_{\text{materialize logits}/\alpha} + \underbrace{NHDb}_{\text{write } O} + \underbrace{NHb}_{\text{optional } L}, \tag{1}$$

where the $O(EHb)$ term accounts for writing and later re-reading edge logits and/or normalized weights. In a more generic message-passing implementation, intermediate edge messages of size $M \in \mathbb{R}^{M \times H \times D}$ may be materialized before reduction, which further increases both traffic and activation footprint to $O(EHD)$.

**Fused CSR streaming (ours).** Our kernels fuse score computation, softmax normalization, and value aggregation into a single CSR traversal. In forward, each block processes one $(i, h)$ pair and streams over neighbors without writing edge tensors. The dominant HBM traffic is then due to reading the participating node features and writing only the final outputs and compact per-node statistics:

$$B_{\text{fused,fwd}} \approx \underbrace{2MHDb}_{\text{read } (Q,V) \text{ or } (\ell,r) \text{ per edge}} + \underbrace{NHDb}_{\text{read cached } K \text{ or } \ell} + \underbrace{NHDb}_{\text{write } O} + \underbrace{NHb}_{\text{write } L}, \tag{2}$$

where $L \in \mathbb{R}^{N \times H}$ denotes `logsumexp`. The key difference from Eq. (1) is the absence of $O(EH)$ or $O(EHD)$ edge materialization terms. For GATv2, we additionally read the attention vector $\mathbf{a} \in \mathbb{R}^{H \times D}$ once per destination node (negligible $NHDb$ compared to edge reads when graphs are non-trivial).

**FLOPs and arithmetic intensity.** Both vanilla and fused variants perform the same core math (score, softmax, and weighted sum), so speedups are primarily IO-driven. For Graph Transformer, the dominant forward FLOPs per edge are the dot product ($\sim 2D$ FLOPs) plus the fused update of the output accumulator ($\sim 3D$ FLOPs), yielding

$$F_{\text{fwd,GT}} \approx M \cdot H \cdot (5D + c_{\text{exp}}), \tag{3}$$

where $c_{\text{exp}}$ captures constant-cost non-linear operations (exp/max/log). For GATv2, the score computation requires evaluating $e_{ij} = \mathbf{a}^\top \sigma(\ell_i + r_j)$, which adds element-wise addition ($D$ ops), LeakyReLU ($D$ ops), and a dot product with $\mathbf{a}$ ($2D$ ops), alongside the weighted aggregation ($3D$ ops), giving

$$F_{\text{fwd,GATv2}} \approx M \cdot H \cdot (7D + c_{\text{exp}} + c_{\text{LReLU}}), \tag{4}$$

*Table 10.* Saved activations for attention layers during training. Our fused kernels avoid storing edge-wise softmax outputs and instead cache compact per-node statistics.

| Method | Edge tensors saved | Node tensors saved | Peak saved size |
|---|---|---|---|
| Generic MP (messages) | $M \in \mathbb{R}^{M \times H \times D}$ | – | $O(MHD)$ |
| Vanilla (edge weights) | $\alpha \in \mathbb{R}^{M \times H}$ | $L \in \mathbb{R}^{N \times H}$ (opt.) | $O(MH)$ |
| Ours (fused, Graph Transformer) | – | $L, \Delta \in \mathbb{R}^{N \times H}$ | $O(NH)$ |
| Ours (fused, GATv2) | – | $L, G \in \mathbb{R}^{N \times H}$ | $O(NH)$ |

where $c_{\text{LReLU}}$ accounts for the LeakyReLU activation. Eliminating edge materialization increases the effective arithmetic intensity $\text{AI} = F/B$ by reducing $B$ in both cases.

**Backward: saving compact statistics instead of edge weights.** A vanilla backward implementation often stores edge-wise attention weights $\alpha \in \mathbb{R}^{M \times H}$ (and sometimes logits), resulting in an $O(MH)$ activation footprint and additional HBM traffic. Our backward kernels avoid storing edge tensors by caching only compact per-node statistics during forward and recomputing $\alpha_{ij} = \exp(e_{ij} - L_i)$ on demand. For the Graph Transformer, we additionally compute $\Delta_{i,h} = \langle O_{i,h,:}, dO_{i,h,:} \rangle$ and use $(L, \Delta)$ to form the softmax gradient without materializing $\alpha$. For GATv2, we similarly cache $L$ and compute a per-node scalar $G_{i,h} = \sum_j \alpha_{ij} \langle dh_i, r_j \rangle$ to evaluate $\nabla e_{ij} = \alpha_{ij}(p_{ij} - G_{i,h})$. As a result, the saved activations for attention scale as $O(NH)$ rather than $O(MH)$ (or $O(MHD)$ in generic message passing).

### D.3. Block-sparse layouts for Tensor Core utilization

While the fused CSR kernels in §3.2 remove edge-tensor materialization, they still operate on irregular gather patterns and primarily rely on scalar cores. To additionally leverage Tensor Cores, we introduce a *weighted block-sparse* representation that packs local neighborhoods into small fixed-size tiles compatible with $16 \times 16$ WMMA-style matrix multiplication. The key idea is to trade preprocessing and (sometimes) extra computation on padded entries for substantially higher arithmetic intensity and better utilization of matrix-multiply hardware.

**WSB format and TCB construction.** We partition destination nodes into row windows of size $R = \text{ROW\_WINDOW\_SIZE} = 16$. For each row window, we collect all edges $(i, j)$ whose destination $i$ lies in the window and build a local list of unique source columns $\{j\}$. These columns are sorted and split into *Tensor Core Blocks* (TCBs) of width $W = \text{TCB\_WIDTH} = 8$. Each TCB stores: (i) `col_idx` - the 8 global column indices for this block (padded with zeros if fewer than 8), (ii) a compact `bitmap` (two 64-bit words) encoding which of the $16 \times 8$ edges exist, and (iii) (for SpMM) a dense $16 \times 8$ weight tile stored row-major (`weights`, padded with zeros for absent edges). We also store `tcb_row_offset`, an index array that maps each row window to its contiguous range of TCBs. This preprocessing is performed once per graph and amortized across training iterations.

**Tensor Core SpMM on WSB.** For SpMM-based convolutions with weighted adjacency, we compute $Y = AX$ using a Tensor Core kernel over WSB. The kernel assigns one program instance to each row window (16 rows) and accumulates a dense output block $Y_{\text{win}} \in \mathbb{R}^{16 \times F}$ in fp32. To form a $16 \times 16$ WMMA-compatible tile, we pair two consecutive TCBs (each $16 \times 8$) into a $16 \times 16$ weight matrix $W_{\text{full}}$. For each pair, we gather the corresponding 16 source columns (8 from each TCB), load the feature block $X_{\text{tile}} \in \mathbb{R}^{16 \times F}$ in fp16, and perform a Tensor Core matmul update:

$$Y_{\text{win}} \mathrel{+}= W_{\text{full}} \cdot X_{\text{tile}}.$$

This design exposes high compute density inside the tile (matrix multiply) while keeping the sparse structure encoded compactly via `col_idx` and the per-row-window TCB list. We find that the expensive component remains the gather of $X_{\text{tile}}$ (irregular column indices), but the subsequent WMMA update amortizes this cost when $F$ is sufficiently large or when windows exhibit reusable column structure.

**Tensor Core attention (Graph Transformer) on WSB.** We apply the same packing to Graph Transformer. For each row window and head, the kernel loads a query block $Q_{\text{win}} \in \mathbb{R}^{16 \times D}$ (fp16) and iterates over pairs of TCBs. Each TCB pair provides up to 16 key/value columns, producing $K_{\text{blk}}, V_{\text{blk}} \in \mathbb{R}^{16 \times D}$. We compute logits via a Tensor Core matmul:

$$S = Q_{\text{win}} K_{\text{blk}}^{\top} \in \mathbb{R}^{16 \times 16},$$

then apply the per-TCB bitmap mask to set invalid edges to $-\infty$. We perform a FlashAttention-style online softmax over columns for each of the 16 rows, maintaining per-row running statistics $(m_i, \ell_i)$ and a value accumulator $\text{acc} \in \mathbb{R}^{16 \times D}$:

$$\text{acc} \leftarrow \text{acc} \cdot \exp(m_{\text{old}} - m_{\text{new}}) + \exp(S - m_{\text{new}}) V_{\text{blk}}.$$

Finally, we normalize $\text{acc}$ by $\ell_i$ to produce $O_{\text{win}}$ and store `logsumexp` for backward. Compared to CSR streaming, WSB increases data reuse within the tile: each loaded key vector participates in dot products with all 16 queries in the window, and each query participates in dot products with 16 keys at once, yielding substantially higher arithmetic intensity inside each TCB pair.

**Backward and practical trade-offs.** The WSB backward kernel follows the same cached-statistics principle as our CSR fused backward: it reuses saved `logsumexp` to reconstruct $P = \exp(S - L)$, computes $dV = P^\top dO$, $dP = dOV^\top$, $dS = P \odot (dP - \Delta)$, and then $dQ = dSK$, $dK = dS^\top Q$ via Tensor Core matmuls. However, because the same source columns can appear in multiple row windows, updates to $dK$ and $dV$ require atomic accumulation. In our SpMM implementation, we observed that a Tensor-Core backward for $dX = A^\top G$ is often dominated by irregular scatters and atomics; in practice, using a high-performance library backend (e.g., cuSPARSE SpMM with a precomputed transposed adjacency) can be faster and more robust.

**When does WSB help?** WSB is most effective when row windows exhibit sufficient local density and/or reuse so that the cost of gathering 16 source columns is amortized by the dense $16 \times 16$ compute. If windows are extremely sparse (few true edges per $16 \times 16$ block), the approach may waste compute on masked entries and increase preprocessing and padding overhead, reducing or eliminating the Tensor Core benefit. We therefore treat WSB as an optional acceleration path and evaluate its sensitivity to graph structure, degree skew, and ordering.

**IO/FLOPs perspective for WSB.** In the CSR fused attention kernel, each edge contributes an independent dot product between a destination vector and a neighbor vector. This exposes limited reuse of $K/Q$ across edges, and the computation is largely memory-bound due to irregular gathers. In contrast, WSB packs neighborhoods into $16 \times 16$ tiles, enabling Tensor Core matmuls that reuse data across the tile.

**Compute.** For Graph Transformer, a single WSB tile pair computes $S = Q_{\text{win}} K_{\text{blk}}^\top$ with $16 \cdot 16 \cdot D$ multiply-adds per head (plus the value update), regardless of how many edges are actually present (masked entries are later set to $-\infty$). Let $\rho \in [0, 1]$ denote the average edge density within the $16 \times 16$ masked tile. Compared to CSR, which performs work proportional to the number of true edges, WSB can increase compute by roughly a factor of $1/\rho$ due to masked computations, while benefiting from Tensor Core throughput and data reuse.

**HBM traffic.** Both CSR and WSB must ultimately load the participating node features. WSB additionally stores compact structural metadata (column lists and bitmaps) and, for weighted SpMM, dense $16 \times 8$ weight tiles. WSB reduces the need to materialize per-edge tensors and can reduce index overhead per potential edge (bitmap vs explicit edge list), but it may increase feature traffic when padded columns are loaded. Overall, WSB should be viewed primarily as an arithmetic-intensity optimization rather than a pure IO-reduction technique.

**Storage overhead.** For attention, each TCB stores 8 column indices (32 bytes) and a 128-bit bitmap (16 bytes), i.e., 48 bytes of metadata per $16 \times 8$ block (covering 128 potential edges). For weighted SpMM, each TCB additionally stores a dense $16 \times 8$ weight tile (128 scalars). This overhead is amortized when row windows are sufficiently dense and reused across iterations; otherwise, padding can increase memory footprint relative to CSR.

**Limitations and when Tensor Cores do not help.** The WSB/TCB path trades irregular sparse compute for dense $16 \times 16$ tile matmuls, and its benefit depends critically on the effective density of each tile. When row windows are very sparse (small $\rho$), the kernel still performs dense $QK^\top$ and subsequent matmul updates on many masked entries, increasing wasted FLOPs and often making performance memory- or overhead-bound again. WSB also introduces preprocessing and padding overhead: constructing per-window unique column lists, bitmaps, and (for weighted SpMM) dense $16 \times 8$ weight tiles increases both conversion time and storage, which must be amortized over repeated training/inference steps. Finally, the backward pass can be less favorable than forward: since the same source columns can appear in multiple row windows, updates to $dK/dV$ (and, for SpMM, $dX$) often require atomic accumulation and irregular scatters, which can dominate runtime and limit the attainable speedup despite using Tensor Core matmuls in the inner loop.

**FLOPs and arithmetic intensity.** We consider FLOPs and arithmetic intensity in GraphTransformer, which performs computations on graphs in CSR and WSB formats. Let's denote $N$ as number of vertices in graph, nnz as number of non-zero elements in adjency matrix, $D$ as embedding dimensionality.

**CSR.** We perform SpDDMM with query and key in bfloat and perform SpMM in shared memory. For every non-zero element we calculate dot product ($2D$ FLOPs per non-zero), calculate row-size softmax (2 FLOPs per non-zero), and perform weighted SpMM ($2D$ FLOPs per non-zero). So, total FLOPs can be approximately written as

$$\text{FLOPs}_{\text{CSR}} \approx \text{nnz} \cdot (4D + 2) \tag{5}$$

During kernel execution, we load every element of key, query and value matrix once ($6ND$ bytes), and two INT32 (8 bytes) indices for every non-zero. This simplified memory-traffic model yields the following:

$$\text{Bytes}_{\text{CSR}} \approx 8\text{nnz} + 6ND, \tag{6}$$

leading to the arithmetic intensity

$$\text{AI}_{\text{CSR}} \approx \frac{\text{nnz} \cdot (4D + 2)}{8\text{nnz} + 6ND}. \tag{7}$$

CSR kernels are typically executed on scalar CUDA cores. So, peak performance is bounded by performance of CUDA threads, so actual flops

$$P_{CSR} = \min(P_{CUDA}, \text{AI}_{CSR} \cdot B_{mem}) \tag{8}$$

where $B_{mem}$ is memory bandwidth.

**WSB.** Now consider same calculations, performed on graph, stored in our WSB format. Graph divded into $16 \times 16$ blocks, and only non-zero blocks are stored. Let's denote $\mu$ as a mean number of non-zero elements in one WSB block ($\mu < 256$). Also, let's denote $n_b = \frac{\text{nnz}}{\mu}$ as number of blocks. For every block in SpDDMM we perform $\frac{256D}{16} = 16D$ floating point operations. In online softmax we perform additional 512 (2 FLOPs per element in block) operations, and after that perform $24D$ amount of operations in weighted SpMM

The executed number of floating-point operations is

$$\text{FLOPs}_{\text{WBS}} \approx n_b(40D + 512) \tag{9}$$

Following the same logic, as in CSR-based GraphTransformer, we load each element of QKV-projections into memory once and for every block, we load 256 BF16 elements and one 128-bit word (16 bytes), and store output once. So, the result is the following:

$$\text{Bytes}_{\text{BS}} \approx 512\,n_b + 4\,n_b + 12ND + 2ND \tag{10}$$

leading to the arithmetic intensity

$$\text{AI}_{\text{BS}} \approx \frac{n_b(40D + 512)}{512\,n_b + 4\,n_b + 12ND + 2ND}. \tag{11}$$

WSB kernels can leverage Tensor Cores; therefore, peak performance is bounded by Tensor Core throughput, and the attainable performance is

$$P_{\text{BS}} = \min(P_{\text{TC}},\ \text{AI}_{\text{BS}} \cdot B_{\text{mem}}), \tag{12}$$

where $B_{\text{mem}}$ is the memory bandwidth.

Note that $P_{\text{TC}} \gg P_{\text{CUDA}}$, so the upper bound on performance of WSB-format-based kernels is much larger than for CSR-based ones.

**Closed-form intensity ratio and limiting regimes.** To make the arithmetic-intensity advantage of WSB precise, we derive a closed-form expression for the ratio $\text{AI}_{\text{BS}}/\text{AI}_{\text{CSR}}$. Substituting $n_b = \text{nnz}/\mu$ into $\text{AI}_{\text{BS}}$ and using $(40D + 512) = 8(5D + 64)$ and $(4D + 2) = 2(2D + 1)$, we obtain

$$\frac{\text{AI}_{\text{BS}}}{\text{AI}_{\text{CSR}}} = \frac{(40D + 512)(8\,\text{nnz} + 6NHD)}{(4D + 2)(516\,n_b + 14NHD)}. \tag{13}$$

Since $\mu \leq 256$, we have $n_b \geq \mathrm{nnz}/256$, so the denominator satisfies $516\, n_b + 14NHD \leq (516/256)\,\mathrm{nnz} + 14NHD \leq 2.02\,\mathrm{nnz} + 14NHD$. A straightforward comparison of numerator and denominator then shows that the ratio is always $\geq 1$: WSB has arithmetic intensity at least as high as CSR regardless of graph size, density, or feature dimension.

The two limiting cases are informative. When edges dominate ($\mathrm{nnz} \gg NHD$), the node-feature terms vanish and the ratio approaches

$$\frac{\mathrm{AI_{BS}}}{\mathrm{AI_{CSR}}} \xrightarrow{\ \mathrm{nnz} \gg NHD\ } \frac{40D + 512}{2(4D + 2)} \cdot \frac{8}{516/\mu}, \tag{14}$$

which for $\mu = 256$ (fully dense tiles) and large $D$ approaches $\approx 5\times$. When node features dominate ($NHD \gg \mathrm{nnz}$), the edge-index terms vanish and the ratio approaches

$$\frac{\mathrm{AI_{BS}}}{\mathrm{AI_{CSR}}} \xrightarrow{\ NHD \gg \mathrm{nnz}\ } \frac{(40D + 512) \cdot 6}{(4D + 2) \cdot 14} \cdot \frac{1}{\rho}, \tag{15}$$

where $\rho = \mu/256 \in (0, 1]$ is the mean tile fill fraction. For large $D$ this simplifies to $\approx (6/14) \cdot (40/4)/\rho \approx 4.3/\rho$, confirming that sparser tiles reduce the intensity advantage proportionally.

**Roofline performance bounds.**   Combining the intensity analysis with the hardware compute ceilings, the attainable throughput for each format is bounded by

$$P_{\mathrm{CSR}} = \min(P_{\mathrm{CUDA}},\ \mathrm{AI_{CSR}} \cdot B_{\mathrm{mem}}), \tag{16}$$

$$P_{\mathrm{BS}} = \min(P_{\mathrm{TC}},\ \mathrm{AI_{BS}} \cdot B_{\mathrm{mem}}), \tag{17}$$

where $B_{\mathrm{mem}}$ is the HBM bandwidth, $P_{\mathrm{CUDA}}$ is the peak scalar CUDA-core throughput, and $P_{\mathrm{TC}}$ is the peak Tensor Core throughput. For typical GNN graphs and feature sizes, $\mathrm{AI_{CSR}}$ is small and the CSR kernel is memory-bound, so $P_{\mathrm{CSR}} \approx \mathrm{AI_{CSR}} \cdot B_{\mathrm{mem}}$. WSB's higher arithmetic intensity moves the kernel closer to the compute-bound regime, and its higher ceiling $P_{\mathrm{TC}} \gg P_{\mathrm{CUDA}}$ (e.g., $\approx 312$ vs. $\approx 19.5$ TFLOPs on A100 for BF16 vs. FP32 scalar) means that even a partially memory-bound WSB kernel can substantially outperform a fully memory-bound CSR kernel. The Tensor Core advantage pays off as long as $P_{\mathrm{TC}}/P_{\mathrm{CUDA}} \gg 1/\rho$, which holds for moderately dense row windows on graphs with locally clustered structure.

### D.4. SpMM-based convolutions

Many widely used GNN layers (e.g., GCN- and SAGE-style aggregations) can be expressed as sparse–dense matrix multiplication (SpMM), $Y = AX$, where $A$ is a sparse adjacency (optionally weighted and normalized) and $X \in \mathbb{R}^{N \times F}$ is a dense feature matrix. Given the maturity and strong performance of vendor sparse linear algebra, we treat NVIDIA cuSPARSE SpMM as a primary backend for this operator family and focus on making it practical and efficient inside a training pipeline.

**cuSPARSE integration.**   We represent $A$ in CSR with 32-bit indices and create cuSPARSE descriptors for the sparse matrix $A$ and dense matrices $B = X$ and $C = Y$. cuSPARSE exposes multiple SpMM algorithm variants for CSR (e.g., `CUSPARSE_SPMM_CSR_ALG1/2/3`), whose performance can vary across graphs and feature dimensions. We therefore support selecting the algorithm explicitly or via a lightweight autotuning routine that benchmarks the available variants and records the best choice for a given graph.

**Normalization as a pre-processing step.**   To support common GNN normalizations, we implement four modes: `none`, `left`, `right`, and `both`. When normalization is enabled and/or edge weights are provided, we precompute node degrees and construct a normalized edge-value array aligned with the CSR `indices`. This produces a float32 edge-value vector that is used as the CSR value buffer in the cuSPARSE sparse-matrix descriptor. When caching is enabled, both degree vectors and normalized edge values are stored and reused across subsequent calls.

**Graph-structure cache for repeated training steps.**   In typical GNN training, the graph structure is fixed while SpMM is invoked many times per epoch (and across epochs), making one-time preprocessing and descriptor creation worth amortizing. We implement a graph cache keyed by the CSR pointers (`indptr`, `indices`) together with the normalization mode, the presence of edge weights, and whether the sparse matrix is used transposed. For each cached graph we store: (i) a cuSPARSE

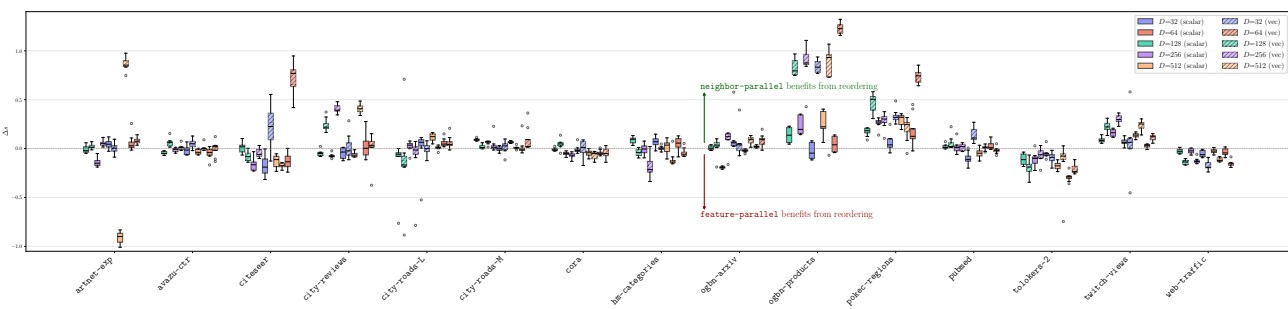

*Figure 2.* Reordering gap $\Delta s = s_{\text{neighbor}} - s_{\text{feature}}$ on all datasets (`dot-aggr`). $\Delta s > 0$ means reordering helps `neighbor-parallel` (gather) more; $\Delta s < 0$ favors `feature-parallel` (coalesced). We sweep $D \in \{32, 64, 128, 256, 512\}$ with scalar vs. 16B vectorized loads (vec); boxes aggregate over partition sizes.

CSR matrix descriptor (`cusparseSpMatDescr_t`) that references the CSR structure and the (possibly normalized) values, (ii) precomputed degree vectors and normalized edge values, (iii) the selected cuSPARSE SpMM algorithm, and (iv) the cuSPARSE workspace buffer and its size. This avoids repeatedly recreating descriptors and reallocating workspace, and it enables reusing the same cuSPARSE handle and graph-specific SpMM configuration across forward and backward passes.

**Forward and backward reuse.** For forward, we invoke `cusparseSpMM` on either $A$ or $A^\top$ (controlled by `do_transpose_a`) depending on the layer formulation. For backward, the same cached graph metadata can be reused: gradients with respect to inputs correspond to multiplying by the transposed adjacency, i.e., $dX = A^\top\, dY$, which is efficiently handled by cuSPARSE using the same CSR structure and cached workspace. By creating and caching the cuSPARSE descriptors and workspace during forward, we reduce overhead in backward and ensure stable performance when the same graph is reused across many iterations.

**Practical considerations.** Our implementation uses 32-bit CSR indices and assumes contiguous dense feature matrices. We expose (i) the SpMM algorithm choice (fixed or autotuned), (ii) the normalization mode, (iii) whether $A$ is transposed, and (iv) the CUDA block size used in degree and normalization preprocessing kernels. We also provide a cache reset utility to control memory usage when benchmarking multiple graphs.

**Overheads and why caching matters.** In practice, for moderate feature sizes or smaller graphs, end-to-end runtime is often not dominated by the SpMM kernel itself but by auxiliary overheads: degree computation and weight normalization, repeated descriptor construction, querying workspace size, and workspace allocation/free. By caching graph-specific preprocessing results (degrees and normalized values) together with the cuSPARSE CSR descriptor, selected algorithm, and workspace buffer, we amortize these costs across repeated training iterations. This turns SpMM invocation into a near-pure library call in both forward and backward, making performance more stable and improving throughput in realistic training loops.

## E. Extended Results

In this section, we provide the extended measurements for all setups evaluated in the paper.

On several large-graph setups (primarily `web-traffic` graph at higher model sizes), the DGL baselines fail during backward due to out-of-memory errors in dense projection layers: the output projection in GATv2 and the QKV projection in Graph Transformer. These failures occur before (or independently of) the custom attention kernels and prevent end-to-end latency/memory measurements for the baseline at those configurations. We mark such entries as `OOM` and omit them from direct comparisons.

### E.1. Graph reordering

Figure 2 demonstrates the reordering effect on `dot-aggr` kernel on all datasets being used.

| Dataset | Fwd time | | | Bwd time | | | Fwd mem | | | Bwd mem | | |
|---|---|---|---|---|---|---|---|---|---|---|---|---|
| | DF-GNN | Ours | Ours (WSB/TC) | DF-GNN | Ours | Ours (WSB/TC) | DF-GNN | Ours | Ours (WSB/TC) | DF-GNN | Ours | Ours (WSB/TC) |
| artnet-exp | 1.26 | 1.37 | 1.39 | 1.21 | 1.14 | 0.67 | 1.20 | 1.24 | 1.07 | 1.03 | 1.04 | 1.19 |
| avazu-ctr | N/A | 2.68 | 5.59 | N/A | 0.86 | 0.91 | N/A | 2.31 | 1.25 | N/A | 2.00 | 1.55 |
| citeseer | 3.02 | 4.58 | 2.56 | 2.20 | 2.57 | 1.91 | 1.04 | 1.05 | 1.03 | 1.12 | 1.01 | 1.08 |
| city-reviews | N/A | 1.48 | 2.02 | N/A | 0.92 | 0.73 | N/A | 1.23 | 1.07 | N/A | 1.05 | 1.20 |
| city-roads-L | 1.04 | 1.13 | 1.18 | 1.17 | 1.24 | 1.00 | 1.15 | 1.15 | 1.06 | 1.09 | 1.01 | 1.20 |
| city-roads-M | 1.17 | 1.22 | 1.27 | 1.23 | 1.24 | 1.04 | 1.16 | 1.17 | 1.07 | 1.01 | 1.01 | 1.21 |
| cora | 2.97 | 4.53 | 2.61 | 2.11 | 2.50 | 1.95 | 1.06 | 1.07 | 1.03 | 1.16 | 1.01 | 1.09 |
| hm-categories | N/A | 3.66 | 6.91 | N/A | 0.95 | 0.92 | N/A | 2.96 | 1.34 | N/A | 2.62 | 1.73 |
| ogbn-arxiv | 2.50 | 1.53 | 2.16 | 1.21 | 1.60 | 0.62 | 1.17 | 1.19 | 1.06 | 1.11 | 1.02 | 1.20 |
| ogbn-products | N/A | 1.51 | 2.73 | N/A | 1.35 | 0.74 | N/A | 1.43 | 1.05 | N/A | 1.16 | 1.20 |
| pokec-regions | N/A | 1.48 | 1.86 | N/A | 1.23 | 0.63 | N/A | 1.27 | 1.04 | N/A | 1.06 | 1.19 |
| pubmed | 1.80 | 1.67 | 2.07 | 0.94 | 1.32 | 1.02 | 1.09 | 1.13 | 1.04 | 1.20 | 1.01 | 1.14 |
| tolokers-2 | N/A | 2.34 | 3.01 | N/A | 1.04 | 0.62 | N/A | 1.54 | 1.09 | N/A | 1.21 | 1.18 |
| twitch-views | N/A | 2.15 | 3.29 | N/A | 0.98 | 0.55 | N/A | 1.60 | 1.00 | N/A | 1.26 | 1.16 |
| web-traffic | N/A | 1.71 | 2.49 | N/A | 0.78 | 1.73 | N/A | 1.11 | 1.04 | N/A | 1.01 | 1.15 |

*Table 11.* Forward/Backward latency speedup ($t_{DGL}/t_{backend}$) and Forward/Backward memory footprint improvement ratio (memory$_{DGL}$/memory$_{backend}$) on Graph Transformer on all datasets (heads $H$=2, head dim $D$=128); **higher is better**. N/A means DF-GNN encountered illegal memory access on this setup and failed to launch.

| Dataset | Fwd time | | | Bwd time | | | Fwd mem | | | Bwd mem | | |
|---|---|---|---|---|---|---|---|---|---|---|---|---|
| | DF-GNN | Ours | Ours (WSB/TC) | DF-GNN | Ours | Ours (WSB/TC) | DF-GNN | Ours | Ours (WSB/TC) | DF-GNN | Ours | Ours (WSB/TC) |
| artnet-exp | 1.15 | 1.16 | 1.29 | 1.17 | 1.09 | 0.62 | 1.19 | 1.20 | 1.07 | 1.08 | 1.02 | 1.20 |
| avazu-ctr | N/A | 2.16 | 4.90 | N/A | 0.73 | 0.69 | N/A | 2.14 | 1.38 | N/A | 1.43 | 1.32 |
| citeseer | 2.37 | 2.45 | 2.99 | 1.70 | 1.66 | 1.34 | 1.06 | 1.07 | 1.03 | 1.18 | 1.01 | 1.10 |
| city-reviews | N/A | 1.20 | 1.54 | N/A | 0.89 | 0.69 | N/A | 1.22 | 1.09 | N/A | 1.03 | 1.21 |
| city-roads-L | 1.13 | 1.16 | 1.15 | 1.18 | 1.15 | 1.01 | 1.16 | 1.16 | 1.07 | 1.13 | 1.00 | 1.21 |
| city-roads-M | 1.12 | 1.17 | 1.16 | 1.15 | 1.19 | 1.05 | 1.16 | 1.17 | 1.08 | 1.08 | 1.00 | 1.21 |
| cora | 2.14 | 2.51 | 2.75 | 1.65 | 1.64 | 1.23 | 1.09 | 1.09 | 1.04 | 1.21 | 1.00 | 1.11 |
| hm-categories | N/A | 2.70 | 6.13 | N/A | 0.77 | 0.67 | N/A | 2.65 | 1.46 | N/A | 1.80 | 1.47 |
| ogbn-arxiv | 1.57 | 1.15 | 1.50 | 1.18 | 1.20 | 0.61 | 1.18 | 1.19 | 1.07 | 1.14 | 1.01 | 1.21 |
| ogbn-products | N/A | 1.25 | 2.17 | N/A | 1.17 | 0.55 | N/A | 1.37 | 1.11 | N/A | 1.08 | 1.21 |
| pokec-regions | N/A | 1.20 | 1.51 | N/A | 1.10 | 0.53 | N/A | 1.24 | 1.08 | N/A | 1.03 | 1.20 |
| pubmed | 1.20 | 1.18 | 1.28 | 1.18 | 1.20 | 0.89 | 1.14 | 1.15 | 1.06 | 1.06 | 1.01 | 1.18 |
| tolokers-2 | N/A | 1.75 | 2.50 | N/A | 0.91 | 0.47 | N/A | 1.47 | 1.15 | N/A | 1.11 | 1.18 |
| twitch-views | N/A | 1.57 | 2.57 | N/A | 0.82 | 0.37 | N/A | 1.50 | 1.10 | N/A | 1.13 | 1.19 |
| web-traffic | N/A | 1.24 | 1.60 | N/A | OOM | OOM | N/A | 1.13 | 1.06 | N/A | OOM | OOM |

*Table 12.* Forward/Backward latency speedup ($t_{DGL}/t_{backend}$) and Forward/Backward memory footprint improvement ratio (memory$_{DGL}$/memory$_{backend}$) on Graph Transformer on all datasets (heads $H$=2, head dim $D$=256); **higher is better**. OOM indicates that the we couldn't measure the metric for the layer. N/A means DF-GNN encountered illegal memory access on this setup and failed to launch.

| Dataset | Fwd time | | | Bwd time | | | Fwd mem | | | Bwd mem | | |
|---|---|---|---|---|---|---|---|---|---|---|---|---|
| | DF-GNN | Ours | Ours (WSB/TC) | DF-GNN | Ours | Ours (WSB/TC) | DF-GNN | Ours | Ours (WSB/TC) | DF-GNN | Ours | Ours (WSB/TC) |
| artnet-exp | 1.24 | 1.23 | 1.45 | 1.23 | 1.21 | 0.77 | 1.20 | 1.26 | 1.10 | 1.03 | 1.04 | 1.20 |
| avazu-ctr | N/A | 2.76 | 5.64 | N/A | 0.92 | 1.21 | N/A | 3.49 | 1.90 | N/A | 2.64 | 2.05 |
| citeseer | 3.79 | 4.57 | 2.71 | 2.20 | 2.48 | 2.09 | 1.04 | 1.05 | 1.03 | 1.12 | 1.01 | 1.08 |
| city-reviews | N/A | 1.52 | 2.13 | N/A | 0.93 | 0.89 | N/A | 1.30 | 1.13 | N/A | 1.06 | 1.21 |
| city-roads-L | 1.05 | 1.11 | 1.29 | 1.15 | 1.22 | 0.88 | 1.16 | 1.16 | 1.07 | 1.09 | 1.01 | 1.20 |
| city-roads-M | 1.15 | 1.06 | 1.25 | 1.21 | 1.29 | 0.93 | 1.17 | 1.18 | 1.08 | 1.01 | 1.01 | 1.21 |
| cora | 3.83 | 4.60 | 2.75 | 2.27 | 2.57 | 2.08 | 1.07 | 1.07 | 1.04 | 1.16 | 1.01 | 1.09 |
| hm-categories | N/A | 3.28 | 6.36 | N/A | 1.01 | 1.18 | N/A | 4.68 | 2.11 | N/A | 3.57 | 2.35 |
| ogbn-arxiv | 2.62 | 1.62 | 2.23 | 1.21 | 1.55 | 0.79 | 1.19 | 1.22 | 1.09 | 1.11 | 1.02 | 1.20 |
| ogbn-products | N/A | 1.46 | 2.48 | N/A | 1.33 | 0.83 | N/A | 1.66 | 1.22 | N/A | 1.19 | 1.23 |
| pokec-regions | N/A | 1.34 | 1.69 | N/A | 1.18 | 0.68 | N/A | 1.36 | 1.11 | N/A | 1.07 | 1.20 |
| pubmed | 1.38 | 1.45 | 1.71 | 0.94 | 1.28 | 1.02 | 1.10 | 1.15 | 1.06 | 1.19 | 1.01 | 1.14 |
| tolokers-2 | N/A | 2.39 | 3.25 | N/A | 1.14 | 0.90 | N/A | 1.88 | 1.33 | N/A | 1.24 | 1.22 |
| twitch-views | N/A | 2.29 | 3.00 | N/A | 0.98 | 0.67 | N/A | 1.97 | 1.23 | N/A | 1.30 | 1.21 |
| web-traffic | N/A | 1.84 | 2.47 | N/A | 0.74 | 2.08 | N/A | 1.12 | 1.05 | N/A | 1.01 | 1.16 |

*Table 13.* Forward/Backward latency speedup ($t_{DGL}/t_{backend}$) and Forward/Backward memory footprint improvement ratio (memory$_{DGL}$/memory$_{backend}$) on Graph Transformer on all datasets (heads $H$=4, head dim $D$=64); **higher is better**. N/A means DF-GNN encountered illegal memory access on this setup and failed to launch.

| Dataset | Fwd time | | | Bwd time | | | Fwd mem | | | Bwd mem | | |
|---|---|---|---|---|---|---|---|---|---|---|---|---|
| | DF-GNN | Ours | Ours (WSB/TC) | DF-GNN | Ours | Ours (WSB/TC) | DF-GNN | Ours | Ours (WSB/TC) | DF-GNN | Ours | Ours (WSB/TC) |
| artnet-exp | 1.19 | 1.17 | 1.29 | 1.21 | 1.15 | 0.78 | 1.19 | 1.22 | 1.08 | 1.08 | 1.02 | 1.21 |
| avazu-ctr | N/A | 2.46 | 5.03 | N/A | 0.91 | 0.97 | N/A | 2.47 | 1.59 | N/A | 1.75 | 1.62 |
| citeseer | 2.22 | 2.41 | 2.73 | 1.84 | 1.85 | 1.52 | 1.06 | 1.07 | 1.03 | 1.18 | 1.01 | 1.10 |
| city-reviews | N/A | 1.30 | 1.61 | N/A | 1.06 | 0.85 | N/A | 1.24 | 1.10 | N/A | 1.03 | 1.21 |
| city-roads-L | 1.13 | 1.15 | 1.15 | 1.18 | 1.16 | 1.03 | 1.16 | 1.16 | 1.07 | 1.13 | 1.00 | 1.21 |
| city-roads-M | 1.12 | 1.15 | 1.16 | 1.20 | 1.19 | 1.07 | 1.16 | 1.17 | 1.08 | 1.08 | 1.00 | 1.21 |
| cora | 2.22 | 2.57 | 2.72 | 1.83 | 1.85 | 1.55 | 1.09 | 1.09 | 1.04 | 1.21 | 1.00 | 1.11 |
| hm-categories | N/A | 2.91 | 5.88 | N/A | 0.98 | 0.95 | N/A | 3.28 | 1.81 | N/A | 2.35 | 1.92 |
| ogbn-arxiv | 1.62 | 1.27 | 1.56 | 1.19 | 1.24 | 0.77 | 1.18 | 1.19 | 1.08 | 1.14 | 1.01 | 1.21 |
| ogbn-products | N/A | 1.37 | 2.16 | N/A | OOM | OOM | N/A | 1.43 | 1.16 | N/A | OOM | OOM |
| pokec-regions | N/A | 1.25 | 1.50 | N/A | 1.15 | 0.70 | N/A | 1.27 | 1.10 | N/A | 1.04 | 1.21 |
| pubmed | 1.22 | 1.22 | 1.30 | 1.14 | 1.18 | 1.00 | 1.15 | 1.15 | 1.07 | 1.06 | 1.01 | 1.18 |
| tolokers-2 | N/A | 2.00 | 2.69 | N/A | 1.10 | 0.71 | N/A | 1.57 | 1.23 | N/A | 1.13 | 1.20 |
| twitch-views | N/A | 1.87 | 2.54 | N/A | 1.02 | 0.54 | N/A | 1.60 | 1.17 | N/A | 1.15 | 1.21 |
| web-traffic | N/A | 1.35 | 1.66 | N/A | OOM | OOM | N/A | 1.14 | 1.06 | N/A | OOM | OOM |

*Table 14.* Forward/Backward latency speedup ($t_{\text{DGL}}/t_{\text{backend}}$) and Forward/Backward memory footprint improvement ratio (memory$_{\text{DGL}}$/memory$_{\text{backend}}$) on Graph Transformer on all datasets (heads $H$=4, head dim $D$=128); **higher is better**. OOM indicates that the we couldn't measure the metric for the layer. N/A means DF-GNN encountered illegal memory access on this setup and failed to launch.

| Dataset | Fwd time | | | Bwd time | | | Fwd mem | | | Bwd mem | | |
|---|---|---|---|---|---|---|---|---|---|---|---|---|
| | DF-GNN | Ours | Ours (WSB/TC) | DF-GNN | Ours | Ours (WSB/TC) | DF-GNN | Ours | Ours (WSB/TC) | DF-GNN | Ours | Ours (WSB/TC) |
| artnet-exp | 1.23 | 1.21 | 1.52 | 1.17 | 1.07 | 0.89 | 1.22 | 1.31 | 1.14 | 1.03 | 1.05 | 1.21 |
| avazu-ctr | N/A | 3.50 | 5.63 | N/A | 0.89 | 1.72 | N/A | 4.90 | 2.66 | N/A | 3.92 | 3.04 |
| citeseer | 3.78 | 4.48 | 2.72 | 2.25 | 2.33 | 2.20 | 1.04 | 1.05 | 1.03 | 1.12 | 1.01 | 1.08 |
| city-reviews | N/A | 1.63 | 2.24 | N/A | 0.88 | 1.08 | N/A | 1.36 | 1.19 | N/A | 1.07 | 1.23 |
| city-roads-L | 1.02 | 0.93 | 1.30 | 1.07 | 1.09 | 0.86 | 1.16 | 1.17 | 1.08 | 1.09 | 1.01 | 1.20 |
| city-roads-M | 1.10 | 0.93 | 1.32 | 1.13 | 1.11 | 0.93 | 1.17 | 1.18 | 1.09 | 1.01 | 1.01 | 1.21 |
| cora | 3.79 | 4.61 | 2.76 | 2.26 | 2.41 | 2.22 | 1.07 | 1.08 | 1.04 | 1.16 | 1.01 | 1.09 |
| hm-categories | N/A | 3.62 | 6.01 | N/A | 0.95 | 1.69 | N/A | 6.72 | 3.03 | N/A | 5.46 | 3.60 |
| ogbn-arxiv | 2.64 | 1.97 | 2.39 | 1.23 | 1.45 | 0.98 | 1.20 | 1.25 | 1.11 | 1.11 | 1.03 | 1.21 |
| ogbn-products | N/A | 1.75 | 2.51 | N/A | 1.10 | 1.05 | N/A | 1.89 | 1.39 | N/A | 1.33 | 1.38 |
| pokec-regions | N/A | 1.38 | 1.67 | N/A | 0.99 | 0.81 | N/A | 1.45 | 1.19 | N/A | 1.09 | 1.22 |
| pubmed | 1.24 | 1.17 | 1.53 | 0.89 | 1.20 | 1.05 | 1.11 | 1.16 | 1.07 | 1.19 | 1.01 | 1.14 |
| tolokers-2 | N/A | 2.99 | 3.83 | N/A | 1.15 | 1.32 | N/A | 2.31 | 1.64 | N/A | 1.40 | 1.38 |
| twitch-views | N/A | 2.92 | 2.72 | N/A | 0.88 | 0.89 | N/A | 2.42 | 1.51 | N/A | 1.73 | 1.60 |
| web-traffic | N/A | 1.88 | 2.37 | N/A | 0.70 | 2.45 | N/A | 1.13 | 1.07 | N/A | 1.02 | 1.16 |

*Table 15.* Forward/Backward latency speedup ($t_{\text{DGL}}/t_{\text{backend}}$) and Forward/Backward memory footprint improvement ratio (memory$_{\text{DGL}}$/memory$_{\text{backend}}$) on Graph Transformer on all datasets (heads $H$=8, head dim $D$=32); **higher is better**. N/A means DF-GNN encountered illegal memory access on this setup and failed to launch.

| Dataset | Fwd time | | | Bwd time | | | Fwd mem | | | Bwd mem | | |
|---|---|---|---|---|---|---|---|---|---|---|---|---|
| | DF-GNN | Ours | Ours (WSB/TC) | DF-GNN | Ours | Ours (WSB/TC) | DF-GNN | Ours | Ours (WSB/TC) | DF-GNN | Ours | Ours (WSB/TC) |
| artnet-exp | 1.21 | 1.11 | 1.27 | 1.21 | 1.11 | 0.82 | 1.20 | 1.24 | 1.11 | 1.08 | 1.03 | 1.21 |
| avazu-ctr | N/A | 2.43 | 4.77 | N/A | 0.93 | 1.23 | N/A | 3.33 | 2.15 | N/A | 2.46 | 2.28 |
| citeseer | 2.19 | 2.28 | 2.70 | 1.74 | 1.71 | 1.52 | 1.06 | 1.07 | 1.03 | 1.18 | 1.01 | 1.10 |
| city-reviews | N/A | 1.27 | 1.63 | N/A | 1.02 | 0.93 | N/A | 1.27 | 1.14 | N/A | 1.04 | 1.22 |
| city-roads-L | 1.11 | 1.07 | 1.15 | 1.15 | 1.12 | 0.91 | 1.16 | 1.17 | 1.08 | 1.13 | 1.00 | 1.21 |
| city-roads-M | 1.11 | 1.06 | 1.16 | 1.19 | 1.16 | 0.97 | 1.17 | 1.18 | 1.08 | 1.08 | 1.01 | 1.22 |
| cora | 2.24 | 2.50 | 2.76 | 1.79 | 1.76 | 1.58 | 1.09 | 1.10 | 1.04 | 1.21 | 1.01 | 1.11 |
| hm-categories | N/A | 2.66 | 5.31 | N/A | 1.03 | 1.21 | N/A | 4.58 | 2.54 | N/A | 3.45 | 2.82 |
| ogbn-arxiv | 1.70 | 1.32 | 1.62 | 1.21 | 1.21 | 0.87 | 1.19 | 1.21 | 1.10 | 1.14 | 1.02 | 1.21 |
| ogbn-products | N/A | 1.36 | 2.08 | N/A | OOM | OOM | N/A | 1.55 | 1.26 | N/A | OOM | OOM |
| pokec-regions | N/A | 1.19 | 1.43 | N/A | 1.13 | 0.75 | N/A | 1.31 | 1.14 | N/A | 1.05 | 1.22 |
| pubmed | 1.14 | 1.04 | 1.23 | 1.13 | 1.14 | 0.99 | 1.15 | 1.16 | 1.07 | 1.06 | 1.01 | 1.18 |
| tolokers-2 | N/A | 2.06 | 2.81 | N/A | 1.17 | 0.92 | N/A | 1.76 | 1.39 | N/A | 1.17 | 1.24 |
| twitch-views | N/A | 1.86 | 2.31 | N/A | 1.01 | 0.64 | N/A | 1.79 | 1.32 | N/A | 1.20 | 1.26 |
| web-traffic | N/A | 1.33 | 1.61 | N/A | OOM | OOM | N/A | 1.15 | 1.07 | N/A | OOM | OOM |

*Table 16.* Forward/Backward latency speedup ($t_{\text{DGL}}/t_{\text{backend}}$) and Forward/Backward memory footprint improvement ratio (memory$_{\text{DGL}}$/memory$_{\text{backend}}$) on Graph Transformer on all datasets (heads $H$=8, head dim $D$=64); **higher is better**. OOM indicates that the we couldn't measure the metric for the layer. N/A means DF-GNN encountered illegal memory access on this setup and failed to launch.

| Dataset | Fwd time | | Bwd time | | Fwd mem | | Bwd mem | |
|---|---|---|---|---|---|---|---|---|
| | Ours | PyG | Ours | PyG | Ours | PyG | Ours | PyG |
| artnet-exp | 6.61 | 0.72 | 1.81 | 0.64 | 6.27 | 0.60 | 4.73 | 0.70 |
| avazu-ctr | 4.91 | 0.68 | 1.43 | 0.85 | 50.54 | 0.61 | 53.35 | 0.68 |
| citeseer | 6.15 | 1.20 | 2.43 | 1.61 | 1.09 | 0.89 | 1.10 | 0.92 |
| city-reviews | 3.22 | 0.69 | 1.11 | 0.94 | 6.74 | 0.61 | 5.74 | 0.69 |
| city-roads-L | 3.33 | 0.60 | 1.29 | 0.59 | 1.57 | 0.58 | 1.39 | 0.71 |
| city-roads-M | 5.47 | 0.46 | 1.76 | 0.90 | 1.90 | 0.56 | 1.61 | 0.71 |
| cora | 8.52 | 1.72 | 2.26 | 1.68 | 1.22 | 0.84 | 1.30 | 0.91 |
| hm-categories | 7.43 | 0.66 | 1.79 | 0.70 | 66.40 | 0.61 | 72.70 | 0.68 |
| ogbn-arxiv | 2.69 | 1.37 | 1.01 | 1.32 | 3.96 | 0.59 | 3.18 | 0.68 |
| pokec-regions | 5.84 | 0.44 | 1.72 | 0.49 | 11.13 | 0.60 | 8.08 | 0.67 |
| pubmed | 6.03 | 1.25 | 1.81 | 1.41 | 1.80 | 0.67 | 1.84 | 0.78 |
| tolokers-2 | 4.76 | 0.60 | 1.65 | 0.91 | 21.76 | 0.61 | 21.61 | 0.70 |
| twitch-views | 4.65 | 0.77 | 1.29 | 0.93 | 34.56 | 0.61 | 27.95 | 0.68 |
| web-traffic | 2.43 | 2.36 | 0.88 | 6.57 | 1.55 | 0.72 | 1.54 | 0.77 |

*Table 17.* Forward/Backward latency speedup ($t_{\text{DGL}}/t_{\text{backend}}$) and Forward/Backward memory footprint improvement ratio (memory$_{\text{DGL}}$/memory$_{\text{backend}}$) on Gatv2 results on all datasets (heads $H$=2, head dim $D$=32); **higher is better**.

| Dataset | Fwd time | | Bwd time | | Fwd mem | | Bwd mem | |
|---|---|---|---|---|---|---|---|---|
| | Ours | PyG | Ours | PyG | Ours | PyG | Ours | PyG |
| artnet-exp | 5.45 | 0.39 | 2.04 | 0.51 | 6.90 | 0.59 | 5.34 | 0.72 |
| avazu-ctr | 5.87 | 0.43 | 1.47 | 0.51 | 81.35 | 0.61 | 74.38 | 0.68 |
| citeseer | 3.60 | 1.25 | 1.94 | 1.57 | 1.15 | 0.83 | 1.26 | 0.88 |
| city-reviews | 3.81 | 0.54 | 1.19 | 0.67 | 8.64 | 0.60 | 6.59 | 0.69 |
| city-roads-L | 9.70 | 1.54 | 1.71 | 0.62 | 1.77 | 0.54 | 1.51 | 0.72 |
| city-roads-M | 4.67 | 0.62 | 1.73 | 0.65 | 2.10 | 0.54 | 1.92 | 0.76 |
| cora | 6.06 | 1.24 | 2.05 | 1.66 | 1.41 | 0.76 | 1.61 | 0.84 |
| hm-categories | 8.66 | 0.41 | 1.96 | 0.44 | 110.29 | 0.61 | 105.89 | 0.68 |
| ogbn-arxiv | 2.82 | 0.95 | 0.98 | 0.95 | 4.67 | 0.58 | 3.46 | 0.69 |
| pokec-regions | 6.68 | OOM | 2.02 | OOM | 12.74 | OOM | 8.47 | OOM |
| pubmed | 5.75 | 1.01 | 1.71 | 0.87 | 2.26 | 0.62 | 2.56 | 0.79 |
| tolokers-2 | 5.59 | 0.45 | 1.89 | 0.70 | 32.59 | 0.61 | 28.55 | 0.71 |
| twitch-views | 5.32 | 0.57 | 1.30 | 0.67 | 44.17 | 0.61 | 31.71 | 0.68 |
| web-traffic | 2.44 | 1.48 | 0.83 | 3.98 | 1.92 | 0.65 | 1.79 | 0.72 |

*Table 18.* Forward/Backward latency speedup ($t_{\text{DGL}}/t_{\text{backend}}$) and Forward/Backward memory footprint improvement ratio (memory$_{\text{DGL}}$/memory$_{\text{backend}}$) on Gatv2 results on all datasets (heads $H$=2, head dim $D$=64); **higher is better**. OOM indicates that the we couldn't measure the metric for the layer.

| Dataset | Fwd time | | Bwd time | | Fwd mem | | Bwd mem | |
|---|---|---|---|---|---|---|---|---|
| | Ours | PyG | Ours | PyG | Ours | PyG | Ours | PyG |
| artnet-exp | 4.37 | 0.36 | 1.89 | 0.51 | 8.00 | 0.58 | 5.40 | 0.69 |
| citeseer | 6.26 | 1.32 | 1.91 | 0.84 | 1.28 | 0.74 | 1.44 | 0.83 |
| city-reviews | 3.55 | 0.41 | 1.06 | 0.54 | 10.07 | 0.58 | 7.00 | 0.67 |
| city-roads-L | 2.07 | 0.41 | 1.44 | 0.61 | 1.89 | 0.51 | 1.47 | 0.67 |
| city-roads-M | 7.44 | 1.35 | 1.60 | 0.42 | 2.20 | 0.51 | 1.76 | 0.71 |
| cora | 11.12 | 2.26 | 2.77 | 2.70 | 1.65 | 0.67 | 1.94 | 0.81 |
| ogbn-arxiv | 2.38 | 0.65 | 0.78 | 0.69 | 5.13 | 0.56 | 3.55 | 0.67 |
| pubmed | 4.16 | 0.66 | 1.39 | 0.67 | 2.74 | 0.58 | 2.61 | 0.79 |
| tolokers-2 | 6.65 | 0.40 | 1.56 | 0.52 | 42.98 | 0.60 | 32.84 | 0.69 |
| twitch-views | 5.50 | 0.35 | 1.12 | OOM | 51.03 | 0.60 | 33.94 | OOM |
| web-traffic | 2.14 | OOM | OOM | OOM | 2.38 | OOM | OOM | OOM |

*Table 19.* Forward/Backward latency speedup ($t_{\text{DGL}}/t_{\text{backend}}$) and Forward/Backward memory footprint improvement ratio (memory$_{\text{DGL}}$/memory$_{\text{backend}}$) on Gatv2 results on all datasets (heads $H$=2, head dim $D$=128); **higher is better**. OOM indicates that the we couldn't measure the metric for the layer.

| Dataset | Fwd time | | Bwd time | | Fwd mem | | Bwd mem | |
|---|---|---|---|---|---|---|---|---|
| | Ours | PyG | Ours | PyG | Ours | PyG | Ours | PyG |
| artnet-exp | 3.23 | 0.37 | 1.63 | 0.50 | 8.42 | 0.57 | 5.49 | 0.67 |
| citeseer | 5.50 | 1.27 | 1.56 | 1.24 | 1.49 | 0.67 | 1.66 | 0.79 |
| city-reviews | 2.86 | 0.38 | 1.13 | 0.51 | 11.08 | 0.58 | 7.24 | 0.66 |
| city-roads-L | 1.61 | 0.50 | 1.24 | 0.67 | 1.98 | 0.50 | 1.45 | 0.65 |
| city-roads-M | 1.73 | 0.49 | 1.32 | 0.67 | 2.28 | 0.50 | 1.68 | 0.67 |
| cora | 5.31 | 0.92 | 1.38 | 0.73 | 2.02 | 0.62 | 2.27 | 0.78 |
| ogbn-arxiv | 1.73 | 0.54 | 0.80 | 0.63 | 5.43 | 0.56 | 3.58 | 0.66 |
| pubmed | 2.15 | 0.46 | 1.29 | 0.62 | 3.20 | 0.56 | 2.61 | 0.73 |
| tolokers-2 | 5.77 | 0.35 | 1.79 | 0.45 | 52.21 | 0.60 | 36.43 | 0.69 |

*Table 20.* Forward/Backward latency speedup ($t_{\text{DGL}}/t_{\text{backend}}$) and Forward/Backward memory footprint improvement ratio (memory$_{\text{DGL}}$/memory$_{\text{backend}}$) on Gatv2 results on all datasets (heads $H$=2, head dim $D$=256); **higher is better**.

| Dataset | Fwd time | | Bwd time | | Fwd mem | | Bwd mem | |
|---|---|---|---|---|---|---|---|---|
| | Ours | PyG | Ours | PyG | Ours | PyG | Ours | PyG |
| artnet-exp | 4.81 | 0.46 | 1.77 | 0.52 | 7.54 | 0.59 | 5.82 | 0.72 |
| avazu-ctr | 5.61 | 0.46 | 1.51 | 0.52 | 85.40 | 0.61 | 79.71 | 0.67 |
| citeseer | 6.15 | 1.26 | 2.69 | 1.85 | 1.16 | 0.83 | 1.27 | 0.88 |
| city-reviews | 3.81 | 0.57 | 1.20 | 0.68 | 9.38 | 0.60 | 7.18 | 0.68 |
| city-roads-L | 2.44 | 0.41 | 1.03 | 0.54 | 1.88 | 0.54 | 1.56 | 0.70 |
| city-roads-M | 3.82 | 0.64 | 1.52 | 0.59 | 2.26 | 0.53 | 2.03 | 0.74 |
| cora | 6.10 | 1.23 | 2.46 | 1.78 | 1.43 | 0.76 | 1.63 | 0.84 |
| hm-categories | 7.19 | 0.43 | 1.81 | 0.46 | 115.38 | 0.61 | 113.01 | 0.67 |
| ogbn-arxiv | 3.29 | 1.10 | 1.09 | 0.92 | 5.12 | 0.57 | 3.75 | 0.68 |
| pokec-regions | 5.59 | OOM | 1.68 | OOM | 14.17 | OOM | 9.37 | OOM |
| pubmed | 5.39 | 0.92 | 1.47 | 0.82 | 2.36 | 0.62 | 2.67 | 0.78 |
| tolokers-2 | 6.02 | 0.56 | 1.84 | 0.66 | 34.30 | 0.61 | 30.93 | 0.71 |
| twitch-views | 5.12 | 0.59 | 1.36 | 0.66 | 48.69 | 0.61 | 35.20 | 0.67 |
| web-traffic | 2.61 | 1.48 | 0.92 | 3.99 | 1.98 | 0.65 | 1.84 | 0.71 |

*Table 21.* Forward/Backward latency speedup ($t_{\text{DGL}}/t_{\text{backend}}$) and Forward/Backward memory footprint improvement ratio (memory$_{\text{DGL}}$/memory$_{\text{backend}}$) on Gatv2 results on all datasets (heads $H$=4, head dim $D$=32); **higher is better**. OOM indicates that the we couldn't measure the metric for the layer.

| Dataset | Fwd time | | Bwd time | | Fwd mem | | Bwd mem | |
|---|---|---|---|---|---|---|---|---|
| | Ours | PyG | Ours | PyG | Ours | PyG | Ours | PyG |
| artnet-exp | 5.18 | 0.37 | 1.91 | 0.45 | 8.62 | 0.58 | 5.91 | 0.68 |
| avazu-ctr | 6.65 | OOM | OOM | OOM | 125.85 | OOM | OOM | OOM |
| citeseer | 6.14 | 1.16 | 2.16 | 1.61 | 1.31 | 0.74 | 1.46 | 0.82 |
| city-reviews | 4.36 | 0.44 | 1.32 | 0.52 | 11.17 | 0.59 | 7.67 | 0.67 |
| city-roads-L | 2.38 | 0.38 | 1.34 | 0.54 | 2.05 | 0.51 | 1.53 | 0.65 |
| city-roads-M | 2.91 | 0.45 | 1.48 | 0.56 | 2.42 | 0.51 | 1.86 | 0.70 |
| cora | 6.14 | 1.24 | 2.01 | 1.67 | 1.68 | 0.67 | 1.95 | 0.80 |
| hm-categories | 8.95 | OOM | OOM | OOM | 176.25 | OOM | OOM | OOM |
| ogbn-arxiv | 2.97 | 0.69 | 0.97 | 0.67 | 5.70 | 0.56 | 3.85 | 0.66 |
| pubmed | 6.25 | 0.88 | 2.01 | 0.65 | 2.93 | 0.58 | 2.77 | 0.78 |
| tolokers-2 | 7.16 | 0.45 | 2.05 | 0.52 | 47.46 | 0.60 | 35.85 | 0.69 |
| twitch-views | 6.06 | 0.38 | 1.49 | OOM | 57.04 | 0.60 | 37.71 | OOM |
| web-traffic | 2.61 | OOM | OOM | OOM | 2.51 | OOM | OOM | OOM |

*Table 22.* Forward/Backward latency speedup ($t_{\text{DGL}}/t_{\text{backend}}$) and Forward/Backward memory footprint improvement ratio (memory$_{\text{DGL}}$/memory$_{\text{backend}}$) on Gatv2 results on all datasets (heads $H$=4, head dim $D$=64); **higher is better**. OOM indicates that the we couldn't measure the metric for the layer.

| Dataset | Fwd time | | Bwd time | | Fwd mem | | Bwd mem | |
|---|---|---|---|---|---|---|---|---|
| | Ours | PyG | Ours | PyG | Ours | PyG | Ours | PyG |
| artnet-exp | 4.61 | 0.36 | 1.75 | 0.47 | 9.33 | 0.57 | 6.03 | 0.67 |
| citeseer | 6.06 | 0.70 | 1.72 | 0.42 | 1.52 | 0.66 | 1.70 | 0.78 |
| city-reviews | 4.48 | 0.37 | 1.27 | 0.49 | 12.42 | 0.58 | 7.98 | 0.66 |
| city-roads-L | 2.05 | 0.42 | 1.30 | 0.57 | 2.17 | 0.49 | 1.51 | 0.63 |
| city-roads-M | 2.17 | 0.41 | 1.39 | 0.59 | 2.51 | 0.50 | 1.76 | 0.66 |
| cora | 10.93 | 2.23 | 1.69 | 1.31 | 2.10 | 0.61 | 2.38 | 0.77 |
| ogbn-arxiv | 2.66 | 0.52 | 0.95 | 0.59 | 6.09 | 0.56 | 3.90 | 0.65 |
| pubmed | 2.93 | 0.42 | 1.39 | 0.57 | 3.47 | 0.55 | 2.77 | 0.72 |
| tolokers-2 | 8.59 | 0.37 | 1.90 | 0.47 | 57.97 | 0.60 | 40.47 | 0.69 |

*Table 23.* Forward/Backward latency speedup ($t_{\text{DGL}}/t_{\text{backend}}$) and Forward/Backward memory footprint improvement ratio (memory$_{\text{DGL}}$/memory$_{\text{backend}}$) on Gatv2 results on all datasets (heads $H$=4, head dim $D$=128); **higher is better**.

| Dataset | Fwd time | | Bwd time | | Fwd mem | | Bwd mem | |
|---|---|---|---|---|---|---|---|---|
| | Ours | PyG | Ours | PyG | Ours | PyG | Ours | PyG |
| artnet-exp | 3.30 | 0.38 | 1.55 | 0.49 | 9.71 | 0.57 | 6.03 | 0.66 |
| citeseer | 3.79 | 1.05 | 1.26 | 0.87 | 1.82 | 0.60 | 1.94 | 0.75 |
| city-reviews | 3.36 | 0.32 | 1.34 | 0.42 | 13.18 | 0.58 | 8.15 | 0.66 |
| city-roads-L | 1.60 | 0.49 | 1.22 | 0.68 | 2.24 | 0.49 | 1.50 | 0.62 |
| city-roads-M | 1.71 | 0.49 | 1.21 | 0.66 | 2.57 | 0.49 | 1.71 | 0.63 |
| cora | 3.80 | 0.70 | 1.15 | 0.88 | 2.57 | 0.58 | 2.70 | 0.76 |
| ogbn-arxiv | 1.99 | 0.47 | 0.99 | 0.58 | 6.30 | 0.55 | 3.92 | 0.65 |
| pubmed | 2.15 | 0.45 | 1.27 | 0.60 | 3.89 | 0.54 | 2.76 | 0.68 |
| tolokers-2 | 7.19 | 0.34 | 2.01 | 0.42 | 65.28 | 0.60 | 42.01 | 0.68 |

*Table 24.* Forward/Backward latency speedup ($t_{\text{DGL}}/t_{\text{backend}}$) and Forward/Backward memory footprint improvement ratio (memory$_{\text{DGL}}$/memory$_{\text{backend}}$) on Gatv2 results on all datasets (heads $H$=4, head dim $D$=256); **higher is better**.

| Dataset | Fwd time | | Bwd time | | Fwd mem | | Bwd mem | |
|---|---|---|---|---|---|---|---|---|
| | Ours | PyG | Ours | PyG | Ours | PyG | Ours | PyG |
| artnet-exp | 4.79 | 0.42 | 1.64 | 0.45 | 9.13 | 0.58 | 6.22 | 0.68 |
| avazu-ctr | 6.14 | OOM | OOM | OOM | 130.84 | OOM | OOM | OOM |
| citeseer | 6.08 | 1.25 | 2.23 | 1.69 | 1.32 | 0.74 | 1.47 | 0.82 |
| city-reviews | 4.21 | 0.47 | 1.30 | 0.54 | 11.80 | 0.59 | 8.11 | 0.66 |
| city-roads-L | 2.22 | 0.37 | 1.74 | 0.76 | 2.14 | 0.50 | 1.56 | 0.64 |
| city-roads-M | 2.70 | 0.40 | 1.24 | 0.48 | 2.55 | 0.51 | 1.92 | 0.69 |
| cora | 5.89 | 1.23 | 1.90 | 1.66 | 1.71 | 0.67 | 1.99 | 0.80 |
| hm-categories | 7.34 | OOM | OOM | OOM | 183.08 | OOM | OOM | OOM |
| ogbn-arxiv | 3.29 | 0.74 | 1.11 | 0.69 | 6.05 | 0.56 | 4.05 | 0.66 |
| pubmed | 3.88 | 0.61 | 1.47 | 0.61 | 3.03 | 0.58 | 2.86 | 0.77 |
| tolokers-2 | 6.47 | 0.48 | 1.94 | 0.54 | 49.03 | 0.60 | 38.02 | 0.69 |
| twitch-views | 5.57 | 0.41 | 1.46 | OOM | 60.66 | 0.60 | 40.19 | OOM |
| web-traffic | 2.57 | OOM | OOM | OOM | 2.58 | OOM | OOM | OOM |

*Table 25.* Forward/Backward latency speedup ($t_{\text{DGL}}/t_{\text{backend}}$) and Forward/Backward memory footprint improvement ratio (memory$_{\text{DGL}}$/memory$_{\text{backend}}$) on Gatv2 results on all datasets (heads $H$=8, head dim $D$=32); **higher is better**. OOM indicates that the we couldn't measure the metric for the layer.

| Dataset | Fwd time | | Bwd time | | Fwd mem | | Bwd mem | |
|---|---|---|---|---|---|---|---|---|
| | Ours | PyG | Ours | PyG | Ours | PyG | Ours | PyG |
| artnet-exp | 5.23 | 0.37 | 1.98 | 0.45 | 9.94 | 0.57 | 6.37 | 0.67 |
| citeseer | 6.11 | 1.25 | 1.97 | 1.47 | 1.55 | 0.66 | 1.72 | 0.78 |
| city-reviews | 5.11 | 0.40 | 1.59 | 0.47 | 13.22 | 0.58 | 8.44 | 0.66 |
| city-roads-L | 2.40 | 0.39 | 1.32 | 0.53 | 2.29 | 0.49 | 1.54 | 0.62 |
| city-roads-M | 2.54 | 0.39 | 1.42 | 0.53 | 2.66 | 0.50 | 1.81 | 0.65 |
| cora | 6.23 | 1.19 | 1.85 | 1.42 | 2.17 | 0.61 | 2.42 | 0.77 |
| ogbn-arxiv | 3.39 | 0.55 | 1.17 | 0.57 | 6.47 | 0.56 | 4.09 | 0.65 |
| pubmed | 5.38 | 0.61 | 1.68 | 0.54 | 3.64 | 0.55 | 2.89 | 0.72 |
| tolokers-2 | 8.23 | 0.40 | 2.37 | 0.48 | 61.85 | 0.60 | 43.06 | 0.69 |

*Table 26.* Forward/Backward latency speedup ($t_{\text{DGL}}/t_{\text{backend}}$) and Forward/Backward memory footprint improvement ratio (memory$_{\text{DGL}}$/memory$_{\text{backend}}$) on Gatv2 results on all datasets (heads $H$=8, head dim $D$=64); **higher is better**.

| Dataset | Fwd time | | Bwd time | | Fwd mem | | Bwd mem | |
|---|---|---|---|---|---|---|---|---|
| | Ours | PyG | Ours | PyG | Ours | PyG | Ours | PyG |
| artnet-exp | 4.73 | 0.36 | 1.83 | 0.47 | 10.38 | 0.57 | 6.37 | 0.66 |
| citeseer | 5.43 | 0.97 | 1.38 | 0.83 | 1.87 | 0.60 | 2.00 | 0.75 |
| city-reviews | 4.96 | 0.31 | 1.48 | 0.40 | 14.09 | 0.58 | 8.62 | 0.66 |
| city-roads-L | 2.01 | 0.42 | 1.34 | 0.58 | 2.37 | 0.48 | 1.53 | 0.61 |
| city-roads-M | 2.19 | 0.42 | 1.38 | 0.59 | 2.72 | 0.49 | 1.76 | 0.62 |
| cora | 5.62 | 1.10 | 1.38 | 0.88 | 2.67 | 0.58 | 2.80 | 0.76 |
| ogbn-arxiv | 3.11 | 0.46 | 1.16 | 0.54 | 6.73 | 0.55 | 4.11 | 0.64 |
| pubmed | 2.91 | 0.40 | 1.47 | 0.55 | 4.10 | 0.54 | 2.85 | 0.67 |
| tolokers-2 | 9.78 | 0.35 | 2.18 | 0.44 | 69.74 | 0.60 | 44.80 | 0.67 |

*Table 27.* Forward/Backward latency speedup ($t_{\text{DGL}}/t_{\text{backend}}$) and Forward/Backward memory footprint improvement ratio (memory$_{\text{DGL}}$/memory$_{\text{backend}}$) on Gatv2 results on all datasets (heads $H$=8, head dim $D$=128); **higher is better**.

| Dataset | Fwd time | | Bwd time | | Fwd mem | | Bwd mem | |
|---|---|---|---|---|---|---|---|---|
| | Ours | PyG | Ours | PyG | Ours | PyG | Ours | PyG |
| artnet-exp | 3.37 | 0.38 | 1.63 | 0.49 | 10.62 | 0.57 | 6.37 | 0.65 |
| citeseer | 2.34 | 0.65 | 1.26 | 0.74 | 2.21 | 0.56 | 2.07 | 0.76 |
| city-reviews | 3.57 | OOM | OOM | OOM | 14.56 | OOM | OOM | OOM |
| city-roads-L | 1.59 | 0.49 | 1.25 | 0.68 | 2.41 | 0.48 | 1.52 | 0.60 |
| city-roads-M | 1.70 | 0.49 | 1.28 | 0.67 | 2.75 | 0.49 | 1.74 | 0.61 |
| cora | 2.46 | 0.63 | 1.20 | 0.70 | 3.12 | 0.55 | 2.76 | 0.77 |
| ogbn-arxiv | 2.23 | 0.37 | 1.19 | 0.50 | 6.86 | 0.55 | 4.13 | 0.64 |
| pubmed | 2.18 | 0.45 | 1.36 | 0.61 | 4.40 | 0.53 | 2.86 | 0.65 |
| tolokers-2 | 8.51 | 0.28 | 2.28 | 0.34 | 74.59 | 0.60 | 45.63 | 0.67 |

*Table 28.* Forward/Backward latency speedup ($t_{\text{DGL}}/t_{\text{backend}}$) and Forward/Backward memory footprint improvement ratio (memory$_{\text{DGL}}$/memory$_{\text{backend}}$) on Gatv2 results on all datasets (heads $H$=8, head dim $D$=256); **higher is better**. OOM indicates that the we couldn't measure the metric for the layer.

| Dataset | Fwd time | | | | | | | Bwd time | | | | | | |
|---|---|---|---|---|---|---|---|---|---|---|---|---|---|---|
| | cuSPARSE | cuSPARSE+ $\bar{A}^\top$ | FuseGNN | TC-GNN | Ours (WSB/TC) | PyG | Pytorch CSR-SpMM | cuSPARSE | cuSPARSE+ $\bar{A}^\top$ | FuseGNN | TC-GNN | Ours (WSB/TC) | PyG | Pytorch CSR-SpMM |
| artnet-exp | 7.26 | 11.19 | 0.31 | 2.45 | 5.05 | 0.76 | 10.15 | 1.77 | 1.64 | 1.44 | 0.62 | 1.55 | 0.38 | 0.47 |
| avazu-ctr | 1.45 | 1.52 | 0.18 | 0.01 | 1.57 | 0.10 | 1.29 | 0.74 | 1.24 | 0.25 | 0.01 | 1.01 | 0.09 | 0.18 |
| citeseer | 9.95 | 12.02 | 0.55 | 12.76 | 5.42 | 1.46 | 14.89 | 1.93 | 2.04 | 2.03 | 0.84 | 1.79 | 1.98 | 0.57 |
| city-reviews | 2.61 | 2.67 | 0.25 | 0.26 | 2.35 | 0.31 | 2.42 | 1.38 | 1.36 | 0.49 | 0.19 | 0.52 | 0.22 | 0.32 |
| city-roads-L | 5.89 | 10.14 | 0.41 | 5.42 | 4.51 | 0.89 | 9.73 | 1.61 | 1.76 | 1.57 | 1.11 | 1.59 | 0.62 | 0.70 |
| city-roads-M | 8.22 | 10.73 | 0.27 | 10.14 | 4.65 | 1.26 | 11.41 | 1.74 | 2.09 | 1.97 | 0.97 | 1.63 | 0.91 | 0.64 |
| cora | 10.92 | 12.07 | 0.55 | 11.24 | 4.64 | 1.45 | 12.96 | 1.93 | 2.24 | 2.33 | 0.97 | 2.03 | 2.20 | 0.63 |
| hm-categories | 1.44 | 1.54 | 0.26 | 0.00 | 2.04 | 0.10 | 1.32 | 0.87 | 1.32 | 0.46 | 0.00 | 0.98 | 0.09 | 0.15 |
| ogbn-arxiv | 3.57 | 6.19 | 0.25 | 1.92 | 1.80 | 0.48 | 5.76 | 1.65 | 2.45 | 1.54 | OOM | 1.71 | 0.37 | 0.56 |
| ogbn-products | 1.21 | 1.17 | 0.34 | 0.34 | 2.22 | 0.11 | 1.11 | 0.73 | 1.18 | 0.89 | 0.49 | 1.07 | OOM | 0.28 |
| pokec-regions | 1.36 | 1.36 | 0.40 | 0.51 | 2.44 | 0.15 | 1.21 | 0.92 | 1.26 | 0.99 | 0.76 | 1.16 | 0.17 | 0.34 |
| pubmed | 11.78 | 14.60 | 0.41 | 7.62 | 5.75 | 1.53 | 14.68 | 2.15 | 2.13 | 3.17 | 1.09 | 2.31 | 1.80 | 0.78 |
| tolokers-2 | 6.66 | 7.92 | 0.38 | 0.93 | 1.81 | 0.54 | 6.96 | 0.99 | 1.87 | 0.99 | 0.31 | 1.31 | 0.25 | 0.31 |
| twitch-views | 1.59 | 1.62 | 0.25 | 0.02 | 1.70 | 0.16 | 1.43 | 1.13 | 1.20 | 0.34 | 0.02 | 1.09 | 0.13 | 0.23 |
| web-traffic | 2.31 | 2.52 | 0.08 | 6.55e-05 | 0.32 | 0.20 | 1.88 | 1.74 | 1.37 | 0.08 | 5.15e-04 | 1.11 | 0.55 | 0.44 |

*Table 29.* Forward/Backward latency speedup ($t_{\text{DGL}}/t_{\text{backend}}$) on SpMM (with GCN as the operator) on all datasets (hidden dim $D$=64); **higher is better**. OOM indicates that the we couldn't measure the metric for the layer.

| Dataset | Fwd time | | | | | | | Bwd time | | | | | | |
|---|---|---|---|---|---|---|---|---|---|---|---|---|---|---|
| | cuSPARSE | cuSPARSE+ $\bar{A}^\top$ | FuseGNN | TC-GNN | Ours (WSB/TC) | PyG | Pytorch CSR-SpMM | cuSPARSE | cuSPARSE+ $\bar{A}^\top$ | FuseGNN | TC-GNN | Ours (WSB/TC) | PyG | Pytorch CSR-SpMM |
| artnet-exp | 6.05 | 6.20 | 0.28 | 1.75 | 4.72 | 0.46 | 6.29 | 1.35 | 1.69 | 1.17 | 0.62 | 1.49 | 0.23 | 0.52 |
| avazu-ctr | 1.40 | 1.41 | 0.14 | 0.01 | 1.61 | 0.05 | 1.33 | 0.50 | 1.20 | 0.16 | 0.01 | 1.02 | 0.04 | 0.21 |
| citeseer | 14.93 | 15.43 | 0.56 | 14.77 | 5.67 | 1.48 | 16.31 | 2.35 | 2.50 | 2.64 | 1.03 | 1.75 | 2.11 | 0.68 |
| city-reviews | 2.56 | 2.58 | 0.25 | 0.24 | 2.12 | 0.19 | 2.26 | 0.91 | 1.30 | 0.32 | 0.23 | 1.19 | 0.14 | 0.37 |
| city-roads-L | 5.38 | 7.96 | 0.43 | 4.20 | 4.05 | 0.62 | 7.67 | 1.32 | 1.65 | 1.41 | 1.09 | 1.48 | 0.41 | 0.76 |
| city-roads-M | 7.36 | 8.45 | 0.28 | 7.85 | 5.40 | 0.94 | 8.36 | 1.28 | 1.85 | 1.74 | 0.98 | 1.63 | 0.54 | 0.65 |
| cora | 11.37 | 14.03 | 0.58 | 12.05 | 6.21 | 1.53 | 13.72 | 1.28 | 2.32 | 2.35 | 1.04 | 1.62 | 2.06 | 0.61 |
| hm-categories | 1.33 | 1.41 | 0.22 | 0.00 | 1.91 | 0.05 | 1.37 | 0.56 | 1.26 | 0.32 | 0.00 | 0.98 | 0.04 | 0.18 |
| ogbn-arxiv | 3.56 | 3.62 | 0.21 | 1.60 | 1.68 | 0.33 | 3.24 | 1.01 | 1.53 | 1.06 | OOM | 1.46 | 0.21 | 0.40 |
| ogbn-products | 1.15 | 1.15 | 0.38 | 0.35 | 1.86 | OOM | 1.06 | 0.53 | 1.12 | 0.73 | 0.57 | 1.09 | OOM | 0.38 |
| pokec-regions | 1.39 | 1.37 | 0.46 | 0.57 | 2.00 | 0.10 | 1.27 | 0.66 | 1.29 | 0.84 | 0.83 | 1.20 | 0.11 | 0.46 |
| pubmed | 11.63 | 14.57 | 0.52 | 6.64 | 5.74 | 1.22 | 12.21 | 1.64 | 2.53 | 2.64 | 1.22 | 2.37 | 1.08 | 0.78 |
| tolokers-2 | 6.33 | 6.32 | 0.36 | 0.81 | 3.38 | 0.35 | 5.83 | 0.93 | 1.75 | 0.48 | 0.26 | 0.67 | 0.13 | 0.33 |
| twitch-views | 1.60 | 1.61 | 0.20 | 0.03 | 1.68 | 0.10 | 1.25 | 0.66 | 1.30 | 0.23 | 0.03 | 1.16 | 0.09 | 0.27 |
| web-traffic | 2.54 | 2.57 | 0.07 | 1.07e-04 | 0.43 | 0.17 | 2.04 | 1.50 | 1.45 | 0.05 | 6.38e-04 | 1.17 | 0.41 | 0.51 |

*Table 30.* Forward/Backward latency speedup ($t_{\text{DGL}}/t_{\text{backend}}$) on SpMM (with GCN as the operator) on all datasets (hidden dim $D$=128); **higher is better**. OOM indicates that the we couldn't measure the metric for the layer.

| Dataset | Fwd time | | | | | | | Bwd time | | | | | | |
|---|---|---|---|---|---|---|---|---|---|---|---|---|---|---|
| | cuSPARSE | cuSPARSE+ $\bar{A}^\top$ | FuseGNN | TC-GNN | Ours (WSB/TC) | PyG | Pytorch CSR-SpMM | cuSPARSE | cuSPARSE+ $\bar{A}^\top$ | FuseGNN | TC-GNN | Ours (WSB/TC) | PyG | Pytorch CSR-SpMM |
| artnet-exp | 3.88 | 3.98 | 0.28 | 1.67 | 4.09 | 0.27 | 3.69 | 1.17 | 1.55 | 1.46 | 0.82 | 1.40 | 0.17 | 0.63 |
| avazu-ctr | 1.25 | 1.35 | 0.28 | 0.02 | 2.34 | 0.04 | 1.19 | 0.53 | 1.20 | 0.38 | 0.02 | 1.08 | 0.04 | 0.32 |
| citeseer | 13.68 | 16.03 | 0.40 | 14.99 | 6.32 | 1.47 | 16.07 | 2.34 | 2.59 | 2.06 | 1.06 | 2.04 | 2.13 | 0.68 |
| city-reviews | 1.98 | 1.96 | 0.37 | 0.37 | 2.10 | 0.14 | 1.69 | 0.91 | 1.39 | 0.58 | 0.40 | 0.67 | 0.13 | 0.53 |
| city-roads-L | 3.77 | 3.80 | 0.45 | 4.04 | 2.88 | 0.42 | 3.23 | 1.49 | 1.76 | 1.65 | 1.50 | 1.51 | 0.38 | 1.00 |
| city-roads-M | 6.01 | 6.30 | 0.29 | 6.99 | 5.04 | 0.65 | 5.89 | 1.47 | 1.75 | 1.68 | 1.17 | 1.54 | 0.42 | 0.75 |
| cora | 13.35 | 15.66 | 0.54 | 11.83 | 6.26 | 1.46 | 13.51 | 2.20 | 2.53 | 2.29 | 1.06 | 2.11 | 1.98 | 0.55 |
| hm-categories | 1.20 | 1.40 | 0.36 | 0.01 | 2.98 | 0.04 | 1.27 | 0.57 | 1.47 | 0.56 | 0.01 | 1.00 | 0.04 | 0.34 |
| ogbn-arxiv | 2.72 | 2.71 | 0.29 | 2.01 | 1.58 | 0.24 | 2.38 | 0.99 | 3.08 | 1.43 | OOM | 1.43 | 0.19 | 1.06 |
| ogbn-products | 1.13 | 1.14 | 0.63 | 0.68 | 1.93 | OOM | 1.08 | 0.53 | 1.12 | 1.05 | 1.04 | 1.08 | OOM | 0.56 |
| pokec-regions | 1.35 | 1.35 | 0.72 | 1.04 | 2.00 | 0.09 | 1.24 | 0.66 | 1.28 | 1.24 | 1.40 | 1.20 | OOM | 0.66 |
| pubmed | 9.06 | 9.21 | 0.51 | 5.81 | 5.74 | 0.91 | 9.44 | 1.68 | 2.03 | 1.90 | 0.90 | 1.87 | 0.50 | 0.65 |
| tolokers-2 | 3.62 | 3.64 | 0.36 | 0.75 | 2.97 | 0.19 | 3.20 | 0.75 | 1.45 | 0.69 | 0.35 | 0.88 | 0.09 | 0.41 |
| twitch-views | 1.33 | 1.36 | 0.32 | 0.05 | 1.82 | 0.07 | 1.21 | 0.66 | 1.27 | 0.43 | 0.05 | 1.18 | 0.07 | 0.46 |
| web-traffic | 2.49 | 2.49 | 0.18 | 2.08e-04 | 0.63 | 0.14 | 1.96 | 1.05 | 1.54 | 0.11 | 9.21e-04 | 1.27 | 0.26 | 0.64 |

*Table 31.* Forward/Backward latency speedup ($t_{\text{DGL}}/t_{\text{backend}}$) on SpMM (with GCN as the operator) on all datasets (hidden dim $D$=256); **higher is better**. OOM indicates that the we couldn't measure the metric for the layer.

| Dataset | Fwd time | | | | | | | Bwd time | | | | | | |
|---|---|---|---|---|---|---|---|---|---|---|---|---|---|---|
| | cuSPARSE | cuSPARSE+ $\bar{A}^\top$ | FuseGNN | TC-GNN | Ours (WSB/TC) | PyG | Pytorch CSR-SpMM | cuSPARSE | cuSPARSE+ $\bar{A}^\top$ | FuseGNN | TC-GNN | Ours (WSB/TC) | PyG | Pytorch CSR-SpMM |
| artnet-exp | 2.76 | 2.84 | 0.29 | 1.77 | 3.28 | 0.18 | 2.47 | 1.08 | 1.48 | 1.36 | 1.12 | 1.34 | 0.15 | 0.77 |
| avazu-ctr | 1.22 | 1.22 | 0.30 | 0.03 | 3.22 | OOM | 1.09 | 0.53 | 1.16 | 0.36 | 0.03 | 1.05 | OOM | 0.48 |
| citeseer | 13.20 | 16.50 | 0.56 | 14.45 | 6.02 | 1.48 | 15.27 | 2.38 | 2.55 | 2.59 | 1.06 | 1.81 | 1.53 | 0.69 |
| city-reviews | 1.75 | 1.73 | 0.41 | 0.54 | 2.14 | 0.12 | 1.48 | 0.89 | 1.40 | 0.55 | 0.66 | 1.11 | 0.12 | 0.74 |
| city-roads-L | 3.24 | 3.27 | 0.53 | 4.99 | 2.35 | 0.36 | 2.69 | 1.51 | 1.74 | 1.80 | 1.94 | 1.55 | 0.36 | 1.19 |
| city-roads-M | 4.05 | 4.30 | 0.28 | 6.04 | 3.15 | 0.42 | 3.65 | 1.48 | 1.69 | 1.76 | 1.54 | 1.54 | 0.37 | 0.91 |
| cora | 13.84 | 15.92 | 0.55 | 11.42 | 6.39 | 1.47 | 12.83 | 2.27 | 2.55 | 2.58 | 1.08 | 2.06 | 1.50 | 0.58 |
| hm-categories | 1.19 | 1.22 | 0.39 | 0.01 | 4.24 | OOM | 1.10 | 0.56 | 1.18 | 0.50 | 0.01 | 1.05 | OOM | 0.43 |
| ogbn-arxiv | 2.30 | 2.27 | 0.27 | 2.63 | 1.56 | 0.20 | 1.95 | 0.97 | 1.64 | 1.44 | OOM | 1.41 | 0.18 | 0.86 |
| ogbn-products | 1.14 | 1.14 | 0.78 | 1.20 | 2.02 | OOM | 1.10 | 0.52 | 1.12 | 1.03 | 1.71 | 1.08 | OOM | 0.74 |
| pokec-regions | 1.34 | 1.34 | 0.92 | 1.76 | 1.99 | OOM | 1.23 | 0.66 | 1.28 | 1.20 | 2.05 | 1.20 | OOM | 0.85 |
| pubmed | 6.35 | 6.73 | 0.50 | 5.10 | 5.28 | 0.59 | 6.37 | 1.35 | 1.71 | 1.51 | 0.97 | 1.57 | 0.33 | 0.69 |
| tolokers-2 | 2.33 | 2.46 | 0.35 | 0.71 | 2.59 | 0.11 | 2.19 | 0.65 | 1.32 | 0.58 | 0.50 | 0.79 | 0.07 | 0.50 |
| twitch-views | 1.26 | 1.26 | 0.32 | 0.08 | 1.80 | 0.06 | 1.13 | 0.61 | 1.18 | 0.37 | 0.09 | 1.07 | OOM | 0.62 |
| web-traffic | 2.44 | 2.44 | 0.19 | 4.08e-04 | 0.87 | OOM | 1.92 | 0.94 | 1.60 | 0.09 | 0.00 | 1.35 | OOM | 0.83 |

*Table 32.* Forward/Backward latency speedup ($t_{\text{DGL}}/t_{\text{backend}}$) on SpMM (with GCN as the operator) on all datasets (hidden dim $D$=512); **higher is better**. OOM indicates that the we couldn't measure the metric for the layer.

| Dataset | Fwd mem | | | | | | | Bwd mem | | | | | | |
| --- | --- | --- | --- | --- | --- | --- | --- | --- | --- | --- | --- | --- | --- | --- |
| | cuSPARSE | cuSPARSE+ $\bar{A}^{\top}$ | FuseGNN | TC-GNN | Ours (WSB/TC) | PyG | Pytorch CSR-SpMM | cuSPARSE | cuSPARSE+ $\bar{A}^{\top}$ | FuseGNN | TC-GNN | Ours (WSB/TC) | PyG | Pytorch CSR-SpMM |
| artnet-exp | 2.05 | 1.95 | 0.04 | 1.97 | 1.17 | 0.03 | 1.43 | 1.41 | 1.37 | 0.05 | 1.39 | 0.97 | 0.04 | 1.06 |
| avazu-ctr | 0.57 | 0.23 | 0.31 | 3.37 | 1.29 | 0.09 | 2.20 | 0.56 | 0.23 | 0.40 | 2.54 | 1.25 | 0.09 | 0.67 |
| citeseer | 0.02 | 0.01 | 0.02 | 1.04 | 1.02 | 0.02 | 1.02 | 0.02 | 0.01 | 0.02 | 1.02 | 1.00 | 0.02 | 1.00 |
| city-reviews | 0.18 | 0.07 | 0.14 | 1.72 | 1.08 | 0.10 | 1.37 | 0.21 | 0.08 | 0.17 | 1.36 | 0.96 | 0.12 | 1.01 |
| city-roads-L | 1.40 | 1.33 | 0.11 | 1.44 | 1.24 | 0.10 | 1.20 | 1.14 | 1.10 | 0.13 | 1.16 | 1.00 | 0.12 | 1.01 |
| city-roads-M | 0.04 | 0.02 | 0.03 | 1.75 | 1.38 | 0.03 | 1.31 | 0.06 | 0.02 | 0.04 | 1.22 | 1.01 | 0.04 | 1.02 |
| cora | 0.01 | 0.00 | 0.01 | 1.11 | 1.05 | 0.01 | 1.06 | 0.01 | 0.00 | 0.01 | 1.06 | 1.00 | 0.01 | 1.01 |
| hm-categories | 0.50 | 0.19 | 0.29 | 3.86 | 1.46 | 0.08 | 2.39 | 0.49 | 0.20 | 0.37 | 2.84 | 1.42 | 0.09 | 0.65 |
| ogbn-arxiv | 0.13 | 0.05 | 0.12 | 1.70 | 1.18 | 0.10 | 1.32 | 0.16 | 0.07 | 0.15 | OOM | 0.97 | 0.12 | 1.04 |
| ogbn-products | 2.28 | 1.36 | 0.73 | 2.54 | 0.92 | 0.13 | 1.76 | 1.72 | 1.20 | 1.02 | 1.84 | 0.90 | OOM | 0.81 |
| pokec-regions | 1.82 | 0.85 | 0.65 | 2.24 | 1.08 | 0.17 | 1.57 | 1.35 | 0.81 | 0.75 | 1.52 | 0.95 | 0.20 | 0.97 |
| pubmed | 0.03 | 0.01 | 0.03 | 1.28 | 1.12 | 0.03 | 1.15 | 0.03 | 0.01 | 0.03 | 1.13 | 1.00 | 0.03 | 1.03 |
| tolokers-2 | 0.03 | 0.01 | 0.03 | 3.31 | 0.97 | 0.02 | 2.06 | 0.03 | 0.01 | 0.03 | 2.26 | 0.96 | 0.02 | 0.73 |
| twitch-views | 0.34 | 0.14 | 0.25 | 3.19 | 0.91 | 0.09 | 2.03 | 0.37 | 0.15 | 0.31 | 2.17 | 0.89 | 0.10 | 0.74 |
| web-traffic | 1.12 | 1.05 | 0.95 | 1.13 | 1.07 | 0.66 | 1.07 | 1.06 | 1.01 | 0.93 | 1.07 | 1.00 | 0.68 | 1.01 |

*Table 33.* Forward/Backward memory footprint improvement ratio ($\text{memory}_{\text{DGL}}/\text{memory}_{\text{backend}}$) on SpMM (with GCN as the operator) on all datasets (hidden dim $D$=64); **higher is better**. OOM indicates that the we couldn't measure the metric for the layer.

| Dataset | Fwd mem | | | | | | | Bwd mem | | | | | | |
| --- | --- | --- | --- | --- | --- | --- | --- | --- | --- | --- | --- | --- | --- | --- |
| | cuSPARSE | cuSPARSE+ $\bar{A}^{\top}$ | FuseGNN | TC-GNN | Ours (WSB/TC) | PyG | Pytorch CSR-SpMM | cuSPARSE | cuSPARSE+ $\bar{A}^{\top}$ | FuseGNN | TC-GNN | Ours (WSB/TC) | PyG | Pytorch CSR-SpMM |
| artnet-exp | 1.96 | 1.90 | 0.06 | 1.98 | 1.29 | 0.05 | 1.39 | 1.32 | 1.30 | 0.08 | 1.31 | 0.98 | 0.06 | 1.03 |
| avazu-ctr | 0.57 | 0.23 | 0.33 | 3.18 | 1.28 | 0.05 | 2.09 | 0.59 | 0.25 | 0.42 | 2.32 | 1.22 | 0.06 | 0.70 |
| citeseer | 0.02 | 0.01 | 0.02 | 1.07 | 1.05 | 0.02 | 1.04 | 0.02 | 0.01 | 0.02 | 1.04 | 1.00 | 0.02 | 1.00 |
| city-reviews | 0.24 | 0.10 | 0.19 | 1.79 | 1.18 | 0.10 | 1.36 | 0.29 | 0.12 | 0.23 | 1.30 | 0.97 | 0.13 | 1.04 |
| city-roads-L | 1.55 | 1.49 | 0.16 | 1.59 | 1.35 | 0.14 | 1.24 | 1.16 | 1.14 | 0.20 | 1.17 | 1.00 | 0.18 | 1.01 |
| city-roads-M | 0.07 | 0.03 | 0.06 | 1.85 | 1.47 | 0.05 | 1.32 | 0.10 | 0.04 | 0.08 | 1.21 | 1.01 | 0.07 | 1.01 |
| cora | 0.01 | 0.00 | 0.01 | 1.17 | 1.10 | 0.01 | 1.09 | 0.01 | 0.00 | 0.01 | 1.08 | 1.00 | 0.01 | 1.01 |
| hm-categories | 0.50 | 0.20 | 0.30 | 3.71 | 1.45 | 0.05 | 2.30 | 0.51 | 0.21 | 0.38 | 2.65 | 1.39 | 0.05 | 0.67 |
| ogbn-arxiv | 0.19 | 0.08 | 0.18 | 1.79 | 1.29 | 0.12 | 1.33 | 0.24 | 0.10 | 0.22 | OOM | 0.98 | 0.16 | 1.02 |
| ogbn-products | 2.02 | 1.37 | 0.88 | 2.40 | 1.01 | OOM | 1.65 | 1.55 | 1.20 | 1.06 | 1.62 | 0.92 | OOM | 0.91 |
| pokec-regions | 1.78 | 1.02 | 0.89 | 2.16 | 1.20 | 0.14 | 1.49 | 1.29 | 0.92 | 0.87 | 1.39 | 0.97 | 0.17 | 1.06 |
| pubmed | 0.03 | 0.01 | 0.03 | 1.40 | 1.22 | 0.03 | 1.19 | 0.04 | 0.02 | 0.04 | 1.15 | 1.00 | 0.04 | 1.02 |
| tolokers-2 | 0.03 | 0.01 | 0.03 | 2.97 | 1.02 | 0.02 | 1.86 | 0.04 | 0.01 | 0.04 | 1.95 | 0.98 | 0.03 | 0.82 |
| twitch-views | 0.38 | 0.16 | 0.29 | 2.88 | 0.96 | 0.06 | 1.84 | 0.43 | 0.19 | 0.37 | 1.87 | 0.92 | 0.07 | 0.83 |
| web-traffic | 1.19 | 1.13 | 1.03 | 1.21 | 1.14 | 0.57 | 1.10 | 1.09 | 1.04 | 0.97 | 1.09 | 1.00 | 0.60 | 1.01 |

*Table 34.* Forward/Backward memory footprint improvement ratio ($\text{memory}_{\text{DGL}}/\text{memory}_{\text{backend}}$) on SpMM (with GCN as the operator) on all datasets (hidden dim $D$=128); **higher is better**. OOM indicates that the we couldn't measure the metric for the layer.

| Dataset | Fwd mem | | | | | | | Bwd mem | | | | | | |
| --- | --- | --- | --- | --- | --- | --- | --- | --- | --- | --- | --- | --- | --- | --- |
| | cuSPARSE | cuSPARSE+ $\bar{A}^{\top}$ | FuseGNN | TC-GNN | Ours (WSB/TC) | PyG | Pytorch CSR-SpMM | cuSPARSE | cuSPARSE+ $\bar{A}^{\top}$ | FuseGNN | TC-GNN | Ours (WSB/TC) | PyG | Pytorch CSR-SpMM |
| artnet-exp | 1.98 | 1.94 | 0.10 | 1.99 | 1.41 | 0.07 | 1.37 | 1.26 | 1.25 | 0.13 | 1.26 | 0.99 | 0.09 | 1.02 |
| avazu-ctr | 0.60 | 0.25 | 0.35 | 2.93 | 1.28 | 0.03 | 1.93 | 0.63 | 0.28 | 0.46 | 2.07 | 1.19 | 0.03 | 0.76 |
| citeseer | 0.03 | 0.01 | 0.03 | 1.13 | 1.09 | 0.03 | 1.06 | 0.03 | 0.01 | 0.03 | 1.06 | 1.00 | 0.03 | 1.00 |
| city-reviews | 0.36 | 0.15 | 0.29 | 1.86 | 1.31 | 0.11 | 1.35 | 0.41 | 0.19 | 0.34 | 1.26 | 0.98 | 0.14 | 1.03 |
| city-roads-L | 1.70 | 1.66 | 0.25 | 1.73 | 1.44 | 0.19 | 1.28 | 1.18 | 1.16 | 0.31 | 1.19 | 1.00 | 0.25 | 1.00 |
| city-roads-M | 0.13 | 0.05 | 0.10 | 1.91 | 1.52 | 0.09 | 1.33 | 0.17 | 0.07 | 0.14 | 1.21 | 1.00 | 0.12 | 1.01 |
| cora | 0.01 | 0.00 | 0.01 | 1.28 | 1.18 | 0.01 | 1.13 | 0.01 | 0.01 | 0.01 | 1.11 | 1.00 | 0.01 | 1.00 |
| hm-categories | 0.51 | 0.21 | 0.32 | 3.42 | 1.43 | 0.03 | 2.14 | 0.54 | 0.22 | 0.41 | 2.36 | 1.33 | 0.03 | 0.71 |
| ogbn-arxiv | 0.31 | 0.13 | 0.29 | 1.87 | 1.41 | 0.15 | 1.33 | 0.36 | 0.17 | 0.35 | OOM | 0.99 | 0.20 | 1.01 |
| ogbn-products | 2.01 | 1.51 | 1.14 | 2.27 | 1.14 | OOM | 1.54 | 1.41 | 1.20 | 1.10 | 1.45 | 0.95 | OOM | 1.05 |
| pokec-regions | 1.86 | 1.26 | 1.14 | 2.09 | 1.33 | 0.12 | 1.42 | 1.25 | 1.02 | 0.98 | 1.30 | 0.98 | OOM | 1.03 |
| pubmed | 0.05 | 0.02 | 0.05 | 1.55 | 1.31 | 0.05 | 1.23 | 0.07 | 0.03 | 0.07 | 1.17 | 1.00 | 0.06 | 1.01 |
| tolokers-2 | 0.04 | 0.02 | 0.04 | 2.63 | 1.13 | 0.02 | 1.69 | 0.05 | 0.02 | 0.05 | 1.63 | 0.98 | 0.03 | 0.94 |
| twitch-views | 0.48 | 0.21 | 0.38 | 2.59 | 1.07 | 0.04 | 1.68 | 0.53 | 0.25 | 0.46 | 1.61 | 0.94 | 0.06 | 0.95 |
| web-traffic | 1.31 | 1.25 | 1.16 | 1.33 | 1.20 | 0.48 | 1.15 | 1.12 | 1.08 | 1.02 | 1.12 | 1.00 | 0.53 | 1.01 |

*Table 35.* Forward/Backward memory footprint improvement ratio ($\text{memory}_{\text{DGL}}/\text{memory}_{\text{backend}}$) on SpMM (with GCN as the operator) on all datasets (hidden dim $D$=256); **higher is better**. OOM indicates that the we couldn't measure the metric for the layer.

| Dataset | Fwd mem | | | | | | | Bwd mem | | | | | | |
| --- | --- | --- | --- | --- | --- | --- | --- | --- | --- | --- | --- | --- | --- | --- |
| | cuSPARSE | cuSPARSE+ $\bar{A}^{\top}$ | FuseGNN | TC-GNN | Ours (WSB/TC) | PyG | Pytorch CSR-SpMM | cuSPARSE | cuSPARSE+ $\bar{A}^{\top}$ | FuseGNN | TC-GNN | Ours (WSB/TC) | PyG | Pytorch CSR-SpMM |
| artnet-exp | 1.99 | 1.97 | 0.17 | 1.99 | 1.48 | 0.09 | 1.35 | 1.23 | 1.23 | 0.22 | 1.23 | 0.99 | 0.12 | 1.01 |
| avazu-ctr | 0.67 | 0.29 | 0.41 | 2.68 | 1.31 | OOM | 1.77 | 0.70 | 0.33 | 0.53 | 1.79 | 1.14 | OOM | 0.85 |
| citeseer | 0.03 | 0.01 | 0.03 | 1.23 | 1.16 | 0.03 | 1.10 | 0.04 | 0.01 | 0.04 | 1.09 | 1.00 | 0.04 | 1.00 |
| city-reviews | 0.54 | 0.24 | 0.44 | 1.92 | 1.41 | 0.11 | 1.34 | 0.58 | 0.30 | 0.50 | 1.23 | 0.99 | 0.15 | 1.01 |
| city-roads-L | 1.82 | 1.80 | 0.41 | 1.84 | 1.51 | 0.27 | 1.30 | 1.19 | 1.18 | 0.47 | 1.19 | 1.00 | 0.35 | 1.00 |
| city-roads-M | 0.23 | 0.09 | 0.18 | 1.95 | 1.56 | 0.14 | 1.33 | 0.29 | 0.12 | 0.24 | 1.20 | 1.00 | 0.20 | 1.00 |
| cora | 0.02 | 0.01 | 0.02 | 1.43 | 1.27 | 0.02 | 1.18 | 0.02 | 0.01 | 0.02 | 1.14 | 1.00 | 0.02 | 1.00 |
| hm-categories | 0.55 | 0.23 | 0.35 | 3.04 | 1.41 | OOM | 1.93 | 0.60 | 0.26 | 0.46 | 2.04 | 1.26 | OOM | 0.78 |
| ogbn-arxiv | 0.49 | 0.23 | 0.47 | 1.93 | 1.49 | 0.18 | 1.33 | 0.54 | 0.29 | 0.52 | OOM | 0.99 | 0.24 | 1.01 |
| ogbn-products | 2.01 | 1.66 | 1.43 | 2.16 | 1.27 | OOM | 1.46 | 1.32 | 1.20 | 1.14 | 1.34 | 0.97 | OOM | 1.04 |
| pokec-regions | 1.92 | 1.50 | 1.40 | 2.05 | 1.44 | OOM | 1.38 | 1.23 | 1.09 | 1.07 | 1.25 | 0.99 | OOM | 1.02 |
| pubmed | 0.08 | 0.03 | 0.08 | 1.69 | 1.40 | 0.07 | 1.26 | 0.11 | 0.04 | 0.11 | 1.18 | 1.00 | 0.09 | 1.00 |
| tolokers-2 | 0.06 | 0.02 | 0.06 | 2.39 | 1.24 | 0.02 | 1.56 | 0.08 | 0.03 | 0.08 | 1.45 | 0.98 | 0.03 | 1.07 |
| twitch-views | 0.64 | 0.30 | 0.54 | 2.36 | 1.20 | 0.03 | 1.55 | 0.66 | 0.35 | 0.60 | 1.43 | 0.96 | OOM | 1.06 |
| web-traffic | 1.47 | 1.42 | 1.33 | 1.48 | 1.31 | OOM | 1.20 | 1.14 | 1.12 | 1.08 | 1.15 | 1.00 | OOM | 1.00 |

*Table 36.* Forward/Backward memory footprint improvement ratio ($\text{memory}_{\text{DGL}}/\text{memory}_{\text{backend}}$) on SpMM (with GCN as the operator) on all datasets (hidden dim $D$=512); **higher is better**. OOM indicates that the we couldn't measure the metric for the layer.

| Dataset | Fwd time | | Bwd time | | Fwd mem | | Bwd mem | |
|---|---|---|---|---|---|---|---|---|
| | cuGraph | Ours | cuGraph | Ours | cuGraph | Ours | cuGraph | Ours |
| artnet-exp | 0.96 | 2.56 | 0.79 | 1.64 | 1.16 | 1.67 | 1.11 | 1.41 |
| avazu-ctr | 0.54 | 7.71 | 0.84 | 1.25 | 0.93 | 2.81 | 0.94 | 2.53 |
| citeseer | 0.07 | 4.43 | 0.91 | 1.75 | 1.02 | 1.04 | 1.02 | 1.04 |
| city-reviews | 0.56 | 5.59 | 1.17 | 1.44 | 1.11 | 1.55 | 1.08 | 1.38 |
| city-roads-L | 1.43 | 2.18 | 1.15 | 1.58 | 1.16 | 1.38 | 1.12 | 1.26 |
| city-roads-M | 1.25 | 2.96 | 0.76 | 2.26 | 1.23 | 1.58 | 1.14 | 1.34 |
| cora | 1.46 | 4.20 | 0.65 | 1.61 | 1.04 | 1.10 | 1.04 | 1.09 |
| hm-categories | 0.51 | 5.07 | 0.97 | 1.38 | 0.92 | 3.32 | 0.92 | 3.00 |
| ogbn-arxiv | 4.31 | 9.35 | 1.78 | 2.60 | 1.16 | 1.55 | 1.12 | 1.35 |
| ogbn-products | 0.67 | 2.84 | 1.11 | 1.41 | 1.04 | 2.12 | 1.03 | 1.76 |
| pokec-regions | 0.80 | 3.40 | 1.14 | 1.68 | 1.13 | 1.88 | 1.09 | 1.53 |
| pubmed | 2.00 | 4.87 | 0.59 | 2.31 | 1.11 | 1.25 | 1.07 | 1.19 |
| tolokers-2 | 0.60 | 3.67 | 0.84 | 1.70 | 0.99 | 2.71 | 0.98 | 2.10 |
| twitch-views | 0.55 | 7.00 | 1.01 | 1.39 | 1.00 | 2.62 | 1.00 | 2.07 |
| web-traffic | 0.12 | 16.85 | 2.99 | 1.26 | 1.05 | 1.12 | 1.05 | 1.11 |

*Table 37.* Forward/Backward latency speedup ($t_{\text{DGL}}/t_{\text{backend}}$) and Forward/Backward memory footprint improvement ratio (memory$_{\text{DGL}}$/memory$_{\text{backend}}$) on reduction-based convolutions (with min-aggregation as the operator) on all datasets (hidden dim $D$=64); **higher is better**.

| Dataset | Fwd time | | Bwd time | | Fwd mem | | Bwd mem | |
|---|---|---|---|---|---|---|---|---|
| | cuGraph | Ours | cuGraph | Ours | cuGraph | Ours | cuGraph | Ours |
| artnet-exp | 1.25 | 2.46 | 0.98 | 1.57 | 1.21 | 1.71 | 1.14 | 1.40 |
| avazu-ctr | 0.61 | 6.03 | 0.39 | 1.32 | 0.96 | 2.63 | 0.97 | 2.24 |
| citeseer | 1.54 | 4.37 | 0.68 | 1.89 | 1.04 | 1.07 | 1.03 | 1.07 |
| city-reviews | 0.66 | 5.06 | 1.12 | 1.46 | 1.17 | 1.61 | 1.12 | 1.38 |
| city-roads-L | 2.36 | 2.63 | 1.30 | 1.58 | 1.21 | 1.49 | 1.14 | 1.30 |
| city-roads-M | 0.23 | 2.31 | 1.00 | 1.54 | 1.26 | 1.65 | 1.16 | 1.36 |
| cora | 1.51 | 4.28 | 0.85 | 1.90 | 1.08 | 1.17 | 1.07 | 1.14 |
| hm-categories | 0.55 | 4.06 | 1.03 | 2.09 | 0.94 | 3.13 | 0.94 | 2.68 |
| ogbn-arxiv | 4.79 | 8.58 | 1.70 | 2.41 | 1.22 | 1.62 | 1.14 | 1.36 |
| ogbn-products | 1.07 | 2.70 | 1.08 | 1.41 | 1.10 | 2.00 | 1.08 | 1.61 |
| pokec-regions | 1.40 | 2.95 | 1.17 | 1.49 | 1.19 | 1.83 | 1.13 | 1.46 |
| pubmed | 1.29 | 2.86 | 0.88 | 1.69 | 1.15 | 1.35 | 1.11 | 1.24 |
| tolokers-2 | 0.75 | 3.45 | 0.66 | 1.75 | 1.06 | 2.39 | 1.04 | 1.85 |
| twitch-views | 0.68 | 5.55 | 1.09 | 1.39 | 1.06 | 2.34 | 1.05 | 1.80 |
| web-traffic | 0.13 | 12.87 | 2.54 | 1.42 | 1.09 | 1.20 | 1.07 | 1.16 |

*Table 38.* Forward/Backward latency speedup ($t_{\text{DGL}}/t_{\text{backend}}$) and Forward/Backward memory footprint improvement ratio (memory$_{\text{DGL}}$/memory$_{\text{backend}}$) on reduction-based convolutions (with min-aggregation as the operator) on all datasets (hidden dim $D$=128); **higher is better**.

| Dataset | Fwd time | | Bwd time | | Fwd mem | | Bwd mem | |
|---|---|---|---|---|---|---|---|---|
| | cuGraph | Ours | cuGraph | Ours | cuGraph | Ours | cuGraph | Ours |
| artnet-exp | 1.32 | 2.85 | 1.08 | 1.53 | 1.26 | 1.71 | 1.16 | 1.38 |
| avazu-ctr | 0.34 | 4.32 | 0.97 | 1.33 | 1.01 | 2.43 | 1.01 | 1.97 |
| citeseer | 1.48 | 4.29 | 0.88 | 2.03 | 1.07 | 1.13 | 1.06 | 1.11 |
| city-reviews | 0.41 | 4.34 | 1.20 | 1.45 | 1.22 | 1.66 | 1.14 | 1.38 |
| city-roads-L | 3.13 | 3.58 | 1.34 | 1.61 | 1.25 | 1.60 | 1.16 | 1.33 |
| city-roads-M | 2.16 | 3.04 | 1.00 | 1.60 | 1.29 | 1.70 | 1.17 | 1.37 |
| cora | 1.26 | 4.10 | 0.82 | 2.02 | 1.13 | 1.27 | 1.10 | 1.20 |
| hm-categories | 0.48 | 2.73 | 1.10 | 1.42 | 0.98 | 2.83 | 0.98 | 2.28 |
| ogbn-arxiv | 3.69 | 6.37 | 1.82 | 2.54 | 1.26 | 1.67 | 1.16 | 1.37 |
| ogbn-products | 1.07 | 2.48 | 1.09 | 1.41 | 1.17 | 1.90 | 1.11 | 1.50 |
| pokec-regions | 1.43 | 2.80 | 1.16 | 1.47 | 1.24 | 1.79 | 1.15 | 1.42 |
| pubmed | 1.12 | 2.97 | 0.93 | 1.48 | 1.20 | 1.46 | 1.14 | 1.28 |
| tolokers-2 | 0.57 | 2.57 | 0.87 | 1.75 | 1.13 | 2.10 | 1.09 | 1.62 |
| twitch-views | 0.48 | 3.62 | 1.09 | 1.38 | 1.14 | 2.10 | 1.09 | 1.61 |
| web-traffic | 0.09 | 10.93 | 2.33 | 1.35 | 1.14 | 1.31 | 1.10 | 1.22 |

*Table 39.* Forward/Backward latency speedup ($t_{\text{DGL}}/t_{\text{backend}}$) and Forward/Backward memory footprint improvement ratio (memory$_{\text{DGL}}$/memory$_{\text{backend}}$) on reduction-based convolutions (with min-aggregation as the operator) on all datasets (hidden dim $D$=256); **higher is better**.

| Dataset | Fwd time | | Bwd time | | Fwd mem | | Bwd mem | |
|---|---|---|---|---|---|---|---|---|
| | cuGraph | Ours | cuGraph | Ours | cuGraph | Ours | cuGraph | Ours |
| artnet-exp | 1.34 | 2.96 | 1.19 | 1.51 | 1.28 | 1.74 | 1.17 | 1.38 |
| avazu-ctr | 0.28 | 3.23 | 1.10 | 1.38 | 1.07 | 2.21 | 1.06 | 1.74 |
| citeseer | 1.17 | 3.94 | 0.98 | 1.77 | 1.12 | 1.22 | 1.09 | 1.16 |
| city-reviews | 0.43 | 3.50 | 1.18 | 1.49 | 1.26 | 1.70 | 1.16 | 1.38 |
| city-roads-L | 3.50 | 4.14 | 1.38 | 1.63 | 1.28 | 1.66 | 1.17 | 1.35 |
| city-roads-M | 2.78 | 3.73 | 1.00 | 1.63 | 1.30 | 1.72 | 1.17 | 1.37 |
| cora | 0.96 | 3.87 | 0.85 | 1.83 | 1.18 | 1.40 | 1.13 | 1.26 |
| hm-categories | 0.36 | 2.41 | 1.18 | 1.51 | 1.03 | 2.51 | 1.02 | 1.95 |
| ogbn-arxiv | 2.87 | 4.84 | 1.45 | 2.09 | 1.28 | 1.71 | 1.17 | 1.37 |
| ogbn-products | 1.05 | 2.48 | 1.09 | 1.39 | 1.23 | 1.83 | 1.14 | 1.44 |
| pokec-regions | 1.42 | 2.59 | 1.14 | 1.41 | 1.27 | 1.77 | 1.16 | 1.40 |
| pubmed | 1.19 | 2.93 | 1.07 | 1.50 | 1.24 | 1.58 | 1.15 | 1.33 |
| tolokers-2 | 0.44 | 2.54 | 0.87 | 1.32 | 1.19 | 1.97 | 1.12 | 1.51 |
| twitch-views | 0.35 | 2.69 | 1.10 | 1.32 | 1.20 | 1.95 | 1.13 | 1.50 |
| web-traffic | 0.06 | 8.31 | 2.17 | 1.52 | 1.19 | 1.43 | 1.13 | 1.28 |

*Table 40.* Forward/Backward latency speedup ($t_{\mathrm{DGL}}/t_{\mathrm{backend}}$) and Forward/Backward memory footprint improvement ratio (memory$_{\mathrm{DGL}}$/memory$_{\mathrm{backend}}$) on reduction-based convolutions (with min-aggregation as the operator) on all datasets (hidden dim $D$=512); **higher is better**.

| Dataset | Fwd time | | | | Bwd time | | | | Fwd mem | | | | Bwd mem | | | |
|---|---|---|---|---|---|---|---|---|---|---|---|---|---|---|---|---|
| | DGL | DF-GNN | Ours | Ours (WSB/TC) | DGL | DF-GNN | Ours | Ours (WSB/TC) | DGL | DF-GNN | Ours | Ours (WSB/TC) | DGL | DF-GNN | Ours | Ours (WSB/TC) |
| artnet-exp | 2.71 | 2.15 | 2.01 | 1.98 | 5.51 | 4.62 | 4.96 | 8.30 | 404.91 | 339.57 | 328.75 | 378.50 | 767.08 | 746.43 | 741.92 | 643.47 |
| avazu-ctr | 54.92 | N/A | 22.10 | 10.62 | 78.56 | N/A | 91.98 | 86.73 | 1644.74 | N/A | 710.89 | 1311.04 | 2614.23 | N/A | 1308.05 | 1684.20 |
| citeseer | 1.22 | 0.40 | 0.27 | 0.48 | 1.31 | 0.60 | 0.51 | 0.70 | 87.75 | 84.77 | 83.69 | 85.78 | 125.10 | 112.28 | 124.19 | 116.54 |
| city-reviews | 11.38 | N/A | 7.73 | 6.10 | 19.60 | N/A | 21.55 | 26.89 | 1263.54 | N/A | 1030.37 | 1183.51 | 2303.21 | N/A | 2196.43 | 1911.58 |
| city-roads-L | 5.18 | 4.97 | 4.58 | 4.40 | 10.90 | 9.47 | 8.93 | 11.04 | 1118.33 | 974.71 | 970.44 | 1052.42 | 2095.34 | 1917.25 | 2082.58 | 1747.78 |
| city-roads-M | 2.23 | 1.92 | 1.93 | 1.85 | 4.74 | 4.10 | 4.06 | 4.85 | 429.67 | 372.99 | 369.36 | 402.45 | 822.59 | 817.73 | 815.99 | 681.88 |
| cora | 1.22 | 0.42 | 0.27 | 0.47 | 1.29 | 0.61 | 0.52 | 0.69 | 50.99 | 48.03 | 47.86 | 49.58 | 81.95 | 70.39 | 81.36 | 75.14 |
| hm-categories | 53.47 | N/A | 15.08 | 8.00 | 69.66 | N/A | 73.45 | 76.12 | 1416.12 | N/A | 477.68 | 1060.52 | 2254.51 | N/A | 861.60 | 1306.97 |
| ogbn-arxiv | 14.66 | 6.01 | 9.60 | 7.32 | 20.43 | 17.15 | 13.01 | 33.27 | 1317.56 | 1131.22 | 1109.60 | 1244.17 | 2489.21 | 2236.22 | 2435.84 | 2072.42 |
| ogbn-products | 259.88 | N/A | 210.58 | 116.01 | 491.59 | N/A | 458.58 | 839.18 | 23420.77 | N/A | 16337.11 | 22245.57 | 41127.10 | N/A | 35472.38 | 34204.84 |
| pokec-regions | 107.05 | N/A | 78.40 | 62.12 | 205.48 | N/A | 178.55 | 349.89 | 12971.79 | N/A | 10227.77 | 12479.97 | 24381.13 | N/A | 22985.20 | 20453.79 |
| pubmed | 1.29 | 0.86 | 0.82 | 0.77 | 2.32 | 2.48 | 1.85 | 2.29 | 194.21 | 177.88 | 171.55 | 185.92 | 402.54 | 336.77 | 398.31 | 353.31 |
| tolokers-2 | 3.01 | N/A | 1.33 | 1.02 | 4.93 | N/A | 5.07 | 8.44 | 146.06 | N/A | 95.11 | 133.67 | 281.98 | N/A | 235.51 | 239.63 |
| twitch-views | 51.41 | N/A | 23.94 | 15.60 | 79.14 | N/A | 81.25 | 145.35 | 1809.94 | N/A | 1128.59 | 1810.49 | 3065.38 | N/A | 2442.73 | 2632.12 |
| web-traffic | 253.29 | N/A | 164.41 | 112.52 | 831.95 | N/A | 1195.89 | 538.99 | 33450.46 | N/A | 30169.03 | 32083.52 | 53330.01 | N/A | 52750.73 | 46197.45 |

*Table 41.* Latency (ms) and Memory footprint (MB) measurements on Graph Transformer on all datasets (heads $H$=2, head dim $D$=128); **lower is better**. N/A means DF-GNN encountered illegal memory access on this setup and failed to launch.

| Dataset | Fwd time | | | | Bwd time | | | | Fwd mem | | | | Bwd mem | | | |
|---|---|---|---|---|---|---|---|---|---|---|---|---|---|---|---|---|
| | DGL | DF-GNN | Ours | Ours (WSB/TC) | DGL | DF-GNN | Ours | Ours (WSB/TC) | DGL | DF-GNN | Ours | Ours (WSB/TC) | DGL | DF-GNN | Ours | Ours (WSB/TC) |
| artnet-exp | 6.91 | 6.00 | 5.97 | 5.35 | 14.87 | 12.78 | 13.71 | 23.83 | 759.77 | 640.63 | 631.56 | 708.06 | 1448.28 | 1343.33 | 1422.83 | 1203.96 |
| avazu-ctr | 80.17 | N/A | 39.38 | 17.35 | 134.29 | N/A | 188.49 | 198.07 | 2483.80 | N/A | 1159.48 | 1798.18 | 3361.54 | N/A | 2354.93 | 2546.00 |
| citeseer | 1.22 | 0.57 | 0.50 | 0.45 | 1.77 | 1.08 | 1.10 | 1.37 | 112.42 | 106.08 | 105.44 | 109.15 | 190.59 | 162.58 | 190.29 | 173.66 |
| city-reviews | 24.64 | N/A | 20.71 | 16.10 | 48.64 | N/A | 55.58 | 72.28 | 2317.01 | N/A | 1902.74 | 2130.45 | 4336.73 | N/A | 4230.88 | 3586.71 |
| city-roads-L | 15.66 | 13.93 | 13.57 | 13.72 | 34.30 | 29.84 | 30.38 | 34.57 | 2099.96 | 1811.42 | 1807.59 | 1957.42 | 4046.93 | 3591.43 | 4034.44 | 3350.27 |
| city-roads-M | 6.47 | 5.77 | 5.58 | 5.59 | 14.16 | 12.47 | 12.53 | 14.26 | 827.23 | 711.24 | 707.61 | 766.99 | 1611.63 | 1495.25 | 1604.50 | 1327.35 |
| cora | 1.21 | 0.58 | 0.48 | 0.44 | 1.67 | 1.02 | 1.03 | 1.39 | 71.83 | 66.27 | 65.98 | 69.14 | 136.56 | 112.76 | 135.98 | 123.31 |
| hm-categories | 77.68 | N/A | 29.31 | 12.91 | 122.26 | N/A | 73.68 | 182.23 | 1990.48 | N/A | 751.71 | 1358.87 | 2667.72 | N/A | 1482.26 | 1816.59 |
| ogbn-arxiv | 27.76 | 17.86 | 24.37 | 18.61 | 49.63 | 42.85 | 42.01 | 83.40 | 2494.61 | 2121.96 | 2102.29 | 2322.42 | 4803.95 | 4218.45 | 4751.80 | 3979.92 |
| ogbn-products | 554.70 | N/A | 536.49 | 308.39 | 1197.22 | N/A | 1232.62 | 2622.15 | 42072.62 | N/A | 30690.47 | 37795.69 | 74618.41 | N/A | 68962.74 | 61715.95 |
| pokec-regions | 258.07 | N/A | 227.46 | 182.06 | 570.44 | N/A | 544.81 | 1134.51 | 24620.00 | N/A | 19800.56 | 22847.51 | 46717.36 | N/A | 45320.64 | 38797.58 |
| pubmed | 2.53 | 2.26 | 2.32 | 2.07 | 5.39 | 4.90 | 4.84 | 6.07 | 332.99 | 291.19 | 289.81 | 313.80 | 639.33 | 605.24 | 635.35 | 543.34 |
| tolokers-2 | 4.79 | N/A | 2.74 | 1.99 | 8.86 | N/A | 10.12 | 19.44 | 244.62 | N/A | 166.26 | 212.10 | 465.53 | N/A | 418.35 | 395.01 |
| twitch-views | 79.76 | N/A | 50.84 | 31.07 | 142.10 | N/A | 172.83 | 386.92 | 3171.29 | N/A | 2117.02 | 2881.58 | 5369.96 | N/A | 4748.28 | 4527.29 |
| web-traffic | 475.79 | N/A | 402.89 | 313.91 | OOM | N/A | OOM | OOM | 53433.19 | N/A | 47109.51 | 50432.36 | OOM | N/A | OOM | OOM |

*Table 42.* Latency (ms) and Memory footprint (MB) measurements on Graph Transformer on all datasets (heads $H$=2, head dim $D$=256); **lower is better**. OOM indicates that the we couldn't measure the metric for the layer. N/A means DF-GNN encountered illegal memory access on this setup and failed to launch.

| Dataset | Fwd time | | | | Bwd time | | | | Fwd mem | | | | Bwd mem | | | |
|---|---|---|---|---|---|---|---|---|---|---|---|---|---|---|---|---|
| | DGL | DF-GNN | Ours | Ours (WSB/TC) | DGL | DF-GNN | Ours | Ours (WSB/TC) | DGL | DF-GNN | Ours | Ours (WSB/TC) | DGL | DF-GNN | Ours | Ours (WSB/TC) |
| artnet-exp | 2.94 | 2.37 | 2.39 | 2.03 | 5.72 | 4.64 | 4.83 | 7.49 | 422.41 | 351.98 | 334.80 | 385.90 | 771.36 | 750.70 | 742.31 | 643.86 |
| avazu-ctr | 64.95 | N/A | 24.92 | 12.22 | 88.56 | N/A | 100.11 | 73.38 | 2486.44 | N/A | 711.49 | 1311.63 | 3455.62 | N/A | 1308.65 | 1684.79 |
| citeseer | 1.22 | 0.32 | 0.27 | 0.45 | 1.30 | 0.60 | 0.54 | 0.63 | 88.06 | 84.84 | 84.22 | 85.81 | 125.17 | 112.35 | 124.72 | 116.56 |
| city-reviews | 13.22 | N/A | 8.68 | 6.20 | 20.87 | N/A | 22.65 | 23.72 | 1335.81 | N/A | 1031.50 | 1185.17 | 2320.99 | N/A | 2197.57 | 1913.23 |
| city-roads-L | 5.33 | 5.09 | 4.83 | 4.15 | 11.01 | 9.67 | 9.17 | 12.55 | 1128.42 | 976.83 | 971.92 | 1053.36 | 2097.38 | 1919.38 | 2084.06 | 1748.72 |
| city-roads-M | 2.29 | 2.01 | 2.17 | 1.84 | 4.78 | 4.19 | 3.95 | 5.19 | 436.02 | 373.83 | 371.11 | 402.88 | 826.74 | 818.57 | 819.60 | 682.31 |
| cora | 1.21 | 0.32 | 0.27 | 0.45 | 1.28 | 0.59 | 0.53 | 0.63 | 51.34 | 48.11 | 47.88 | 49.60 | 82.03 | 70.47 | 81.38 | 75.16 |
| hm-categories | 61.81 | N/A | 18.82 | 9.72 | 77.68 | N/A | 76.56 | 65.83 | 2236.40 | N/A | 478.04 | 1060.88 | 3074.32 | N/A | 861.96 | 1307.33 |
| ogbn-arxiv | 16.04 | 6.16 | 9.96 | 7.22 | 21.15 | 17.84 | 13.93 | 27.21 | 1353.16 | 1140.32 | 1110.30 | 1245.85 | 2496.23 | 2245.32 | 2436.55 | 2074.10 |
| ogbn-products | 307.41 | N/A | 248.69 | 146.49 | 541.99 | N/A | 499.76 | 802.31 | 27215.25 | N/A | 16355.80 | 22264.26 | 42070.89 | N/A | 35491.07 | 34223.53 |
| pokec-regions | 119.39 | N/A | 95.09 | 75.04 | 217.43 | N/A | 195.40 | 338.89 | 13919.14 | N/A | 10240.23 | 12492.43 | 24614.76 | N/A | 22997.66 | 20466.25 |
| pubmed | 1.28 | 0.95 | 0.91 | 0.80 | 2.32 | 2.49 | 1.92 | 2.39 | 197.06 | 178.56 | 171.70 | 186.07 | 403.21 | 337.44 | 398.47 | 353.46 |
| tolokers-2 | 3.52 | N/A | 1.50 | 1.10 | 5.47 | N/A | 5.14 | 6.58 | 177.82 | N/A | 95.20 | 133.76 | 289.90 | N/A | 235.60 | 239.72 |
| twitch-views | 57.77 | N/A | 25.48 | 19.36 | 84.83 | N/A | 86.90 | 126.41 | 2227.41 | N/A | 1130.11 | 1811.72 | 3182.40 | N/A | 2444.25 | 2633.35 |
| web-traffic | 280.37 | N/A | 162.95 | 121.45 | 846.52 | N/A | 1284.58 | 458.24 | 33866.05 | N/A | 30191.08 | 32105.57 | 53428.39 | N/A | 52772.78 | 46219.50 |

*Table 43.* Latency (ms) and Memory footprint (MB) measurements on Graph Transformer on all datasets (heads $H$=4, head dim $D$=64); **lower is better**. N/A means DF-GNN encountered illegal memory access on this setup and failed to launch.

| Dataset | Fwd time | | | | Bwd time | | | | Fwd mem | | | | Bwd mem | | | |
|---|---|---|---|---|---|---|---|---|---|---|---|---|---|---|---|---|
| | DGL | DF-GNN | Ours | Ours (WSB/TC) | DGL | DF-GNN | Ours | Ours (WSB/TC) | DGL | DF-GNN | Ours | Ours (WSB/TC) | DGL | DF-GNN | Ours | Ours (WSB/TC) |
| artnet-exp | 7.15 | 6.03 | 6.10 | 5.55 | 15.11 | 12.60 | 13.42 | 19.67 | 768.01 | 644.91 | 631.95 | 708.44 | 1452.55 | 1347.60 | 1423.19 | 1204.34 |
| avazu-ctr | 90.45 | N/A | 38.59 | 18.86 | 146.17 | N/A | 162.22 | 152.20 | 2860.74 | N/A | 1160.06 | 1798.01 | 4129.89 | N/A | 2354.78 | 2546.55 |
| citeseer | 1.22 | 0.56 | 0.51 | 0.45 | 1.77 | 1.06 | 1.05 | 1.24 | 112.56 | 106.15 | 105.46 | 109.18 | 190.66 | 162.65 | 190.64 | 173.69 |
| city-reviews | 26.33 | N/A | 20.50 | 16.55 | 50.19 | N/A | 49.92 | 60.48 | 2352.57 | N/A | 1903.64 | 2132.08 | 4354.51 | N/A | 4232.51 | 3588.34 |
| city-roads-L | 15.80 | 14.09 | 13.82 | 13.81 | 34.43 | 29.90 | 30.35 | 34.07 | 2104.22 | 1813.55 | 1808.64 | 1958.30 | 4048.96 | 3593.56 | 4035.49 | 3351.15 |
| city-roads-M | 6.53 | 5.86 | 5.68 | 5.64 | 14.22 | 12.45 | 12.55 | 13.76 | 828.69 | 712.08 | 708.04 | 767.42 | 1612.07 | 1496.09 | 1604.93 | 1327.79 |
| cora | 1.21 | 0.55 | 0.47 | 0.45 | 1.67 | 0.97 | 0.94 | 1.15 | 71.99 | 66.35 | 66.00 | 69.04 | 136.53 | 112.84 | 136.00 | 123.17 |
| hm-categories | 85.61 | N/A | 29.48 | 14.56 | 131.14 | N/A | 133.80 | 138.19 | 2466.01 | N/A | 752.06 | 1359.23 | 3488.08 | N/A | 1482.61 | 1816.95 |
| ogbn-arxiv | 29.08 | 18.09 | 23.31 | 18.78 | 50.50 | 43.03 | 41.52 | 66.49 | 2512.41 | 2131.06 | 2103.34 | 2323.66 | 4812.84 | 4227.56 | 4752.73 | 3981.16 |
| ogbn-products | 605.63 | N/A | 526.40 | 332.24 | OOM | N/A | 1185.58 | 1944.58 | 43960.62 | N/A | 30709.16 | 37813.77 | OOM | N/A | 68981.42 | 61734.03 |
| pokec-regions | 271.27 | N/A | 229.13 | 191.23 | 584.70 | N/A | 532.38 | 874.32 | 25087.28 | N/A | 19812.88 | 22859.51 | 46951.02 | N/A | 45333.09 | 38809.58 |
| pubmed | 2.57 | 2.26 | 2.28 | 2.09 | 5.42 | 4.82 | 4.72 | 5.49 | 334.48 | 292.01 | 289.96 | 313.20 | 640.35 | 606.05 | 635.50 | 543.49 |
| tolokers-2 | 5.29 | N/A | 2.69 | 2.03 | 9.44 | N/A | 8.92 | 13.76 | 260.46 | N/A | 166.35 | 211.17 | 473.45 | N/A | 418.44 | 394.83 |
| twitch-views | 86.12 | N/A | 46.54 | 33.98 | 148.22 | N/A | 145.96 | 274.12 | 3379.29 | N/A | 2118.36 | 2882.50 | 5473.68 | N/A | 4749.63 | 4528.95 |
| web-traffic | 502.98 | N/A | 385.20 | 314.21 | OOM | N/A | OOM | OOM | 53629.95 | N/A | 47131.57 | 50454.41 | OOM | N/A | OOM | OOM |

*Table 44.* Latency (ms) and Memory footprint (MB) measurements on Graph Transformer on all datasets (heads $H$=4, head dim $D$=128); **lower is better**. OOM indicates that the we couldn't measure the metric for the layer. N/A means DF-GNN encountered illegal memory access on this setup and failed to launch.

| Dataset | Fwd time | | | | Bwd time | | | | Fwd mem | | | | Bwd mem | | | |
|---|---|---|---|---|---|---|---|---|---|---|---|---|---|---|---|---|
| | DGL | DF-GNN | Ours | Ours (WSB/TC) | DGL | DF-GNN | Ours | Ours (WSB/TC) | DGL | DF-GNN | Ours | Ours (WSB/TC) | DGL | DF-GNN | Ours | Ours (WSB/TC) |
| artnet-exp | 3.57 | 2.91 | 2.95 | 2.39 | 6.37 | 5.45 | 6.08 | 7.22 | 440.41 | 361.42 | 335.57 | 386.71 | 779.91 | 760.15 | 743.08 | 644.67 |
| avazu-ctr | 88.42 | N/A | 26.39 | 16.44 | 112.47 | N/A | 126.51 | 65.32 | 3491.66 | N/A | 712.63 | 1312.79 | 5128.84 | N/A | 1309.79 | 1685.95 |
| citeseer | 1.22 | 0.32 | 0.27 | 0.45 | 1.30 | 0.59 | 0.57 | 0.61 | 88.33 | 84.98 | 84.27 | 85.86 | 125.30 | 112.49 | 124.77 | 116.61 |
| city-reviews | 16.08 | N/A | 9.90 | 7.20 | 23.87 | N/A | 27.21 | 22.14 | 1406.94 | N/A | 1033.77 | 1186.86 | 2356.56 | N/A | 2199.84 | 1914.93 |
| city-roads-L | 5.67 | 5.59 | 6.12 | 4.38 | 11.34 | 10.63 | 10.56 | 13.26 | 1136.94 | 981.09 | 973.55 | 1055.53 | 2101.63 | 1923.63 | 2085.69 | 1750.89 |
| city-roads-M | 2.43 | 2.23 | 2.61 | 1.85 | 4.92 | 4.58 | 4.52 | 5.34 | 440.06 | 376.09 | 371.99 | 403.75 | 828.77 | 820.83 | 820.47 | 683.18 |
| cora | 1.22 | 0.32 | 0.27 | 0.45 | 1.28 | 0.59 | 0.56 | 0.58 | 51.66 | 48.27 | 47.92 | 49.64 | 82.19 | 70.64 | 81.42 | 75.20 |
| hm-categories | 82.56 | N/A | 22.82 | 13.75 | 100.13 | N/A | 104.89 | 59.36 | 3218.72 | N/A | 478.91 | 1061.75 | 4710.04 | N/A | 862.83 | 1308.20 |
| ogbn-arxiv | 19.17 | 7.28 | 9.77 | 8.03 | 23.61 | 19.40 | 16.55 | 24.47 | 1388.75 | 1158.32 | 1112.89 | 1248.02 | 2514.03 | 2263.32 | 2439.13 | 2076.27 |
| ogbn-products | 442.34 | 6.28 | 291.97 | 202.94 | 679.89 | 12.88 | 733.21 | 772.10 | 30991.46 | N/A | 16393.17 | 22301.63 | 47178.58 | N/A | 35528.44 | 34260.90 |
| pokec-regions | 152.91 | N/A | 117.54 | 97.00 | 251.40 | N/A | 269.70 | 328.94 | 14853.67 | N/A | 10265.28 | 12517.51 | 25082.02 | N/A | 23022.71 | 20491.34 |
| pubmed | 1.28 | 1.05 | 1.11 | 0.86 | 2.34 | 2.65 | 2.06 | 2.24 | 199.77 | 179.91 | 172.00 | 186.37 | 404.56 | 338.80 | 398.77 | 353.76 |
| tolokers-2 | 4.89 | N/A | 1.65 | 1.34 | 6.88 | N/A | 6.35 | 5.53 | 218.93 | N/A | 95.38 | 133.94 | 328.61 | N/A | 235.78 | 239.90 |
| twitch-views | 73.45 | N/A | 25.26 | 27.01 | 100.11 | N/A | 114.84 | 113.57 | 2737.99 | N/A | 1132.24 | 1814.29 | 4220.09 | N/A | 2446.39 | 2635.91 |
| web-traffic | 307.16 | N/A | 163.25 | 129.85 | 873.10 | N/A | 1404.93 | 398.55 | 34259.59 | N/A | 30235.18 | 32149.67 | 53625.16 | N/A | 52816.88 | 46263.60 |

*Table 45.* Latency (ms) and Memory footprint (MB) measurements on Graph Transformer on all datasets (heads $H$=8, head dim $D$=32); **lower is better**. N/A means DF-GNN encountered illegal memory access on this setup and failed to launch.

| Dataset | Fwd time | | | | Bwd time | | | | Fwd mem | | | | Bwd mem | | | |
|---|---|---|---|---|---|---|---|---|---|---|---|---|---|---|---|---|
| | DGL | DF-GNN | Ours | Ours (WSB/TC) | DGL | DF-GNN | Ours | Ours (WSB/TC) | DGL | DF-GNN | Ours | Ours (WSB/TC) | DGL | DF-GNN | Ours | Ours (WSB/TC) |
| artnet-exp | 7.58 | 6.28 | 6.85 | 5.96 | 15.59 | 12.88 | 14.03 | 19.09 | 785.43 | 654.36 | 632.76 | 709.22 | 1461.11 | 1357.05 | 1424.00 | 1205.12 |
| avazu-ctr | 108.26 | N/A | 46.41 | 23.54 | 166.21 | N/A | 179.20 | 135.22 | 3866.44 | N/A | 1161.22 | 1799.92 | 5803.13 | N/A | 2355.93 | 2547.74 |
| citeseer | 1.22 | 0.56 | 0.53 | 0.45 | 1.82 | 1.06 | 1.07 | 1.21 | 112.83 | 106.29 | 105.51 | 109.23 | 190.80 | 162.79 | 190.36 | 173.74 |
| city-reviews | 28.70 | N/A | 22.65 | 17.66 | 52.53 | N/A | 52.19 | 56.99 | 2423.70 | N/A | 1905.91 | 2133.77 | 4390.08 | N/A | 4234.05 | 3590.03 |
| city-roads-L | 16.03 | 14.50 | 15.13 | 14.02 | 34.67 | 30.40 | 31.28 | 38.28 | 2112.83 | 1817.81 | 1810.27 | 1960.67 | 4053.31 | 3597.82 | 4037.82 | 3353.52 |
| city-roads-M | 6.63 | 6.01 | 6.27 | 5.73 | 14.33 | 12.62 | 12.96 | 15.20 | 832.73 | 714.34 | 708.92 | 768.29 | 1614.10 | 1498.35 | 1605.80 | 1328.65 |
| cora | 1.22 | 0.55 | 0.49 | 0.45 | 1.67 | 0.97 | 0.98 | 1.06 | 72.43 | 66.51 | 66.04 | 69.56 | 136.81 | 113.00 | 136.04 | 123.69 |
| hm-categories | 100.31 | N/A | 37.78 | 18.90 | 148.35 | N/A | 143.95 | 123.07 | 3448.49 | N/A | 753.05 | 1359.94 | 5123.63 | N/A | 1483.60 | 1817.66 |
| ogbn-arxiv | 31.72 | 18.80 | 24.20 | 19.76 | 52.67 | 44.24 | 43.97 | 61.25 | 2548.00 | 2149.06 | 2105.50 | 2325.83 | 4830.64 | 4245.56 | 4755.24 | 3983.33 |
| ogbn-products | 701.30 | N/A | 598.61 | 391.81 | OOM | N/A | 1278.95 | 1869.35 | 47736.62 | N/A | 30746.53 | 37851.74 | OOM | N/A | 69018.79 | 61772.00 |
| pokec-regions | 294.59 | N/A | 259.24 | 215.95 | 609.00 | N/A | 565.86 | 852.05 | 26021.79 | N/A | 19837.80 | 22885.05 | 47418.26 | N/A | 45357.87 | 38835.13 |
| pubmed | 2.64 | 2.31 | 2.55 | 2.15 | 5.51 | 4.88 | 4.85 | 5.60 | 337.04 | 293.22 | 290.26 | 314.25 | 641.36 | 607.27 | 635.80 | 543.80 |
| tolokers-2 | 6.39 | N/A | 3.13 | 2.29 | 10.61 | N/A | 9.44 | 11.93 | 292.13 | N/A | 166.52 | 211.35 | 489.29 | N/A | 418.62 | 395.00 |
| twitch-views | 96.18 | N/A | 52.26 | 42.01 | 158.91 | N/A | 158.47 | 248.87 | 3793.62 | N/A | 2120.50 | 2884.63 | 5703.12 | N/A | 4751.76 | 4531.08 |
| web-traffic | 523.32 | N/A | 394.86 | 326.18 | OOM | N/A | OOM | OOM | 54023.49 | N/A | 47175.67 | 50498.51 | OOM | N/A | OOM | OOM |

*Table 46.* Latency (ms) and Memory footprint (MB) measurements on Graph Transformer on all datasets (heads $H$=8, head dim $D$=64); **lower is better**. OOM indicates that the we couldn't measure the metric for the layer. N/A means DF-GNN encountered illegal memory access on this setup and failed to launch.

| Dataset | Fwd time | | | Bwd time | | | Fwd mem | | | Bwd mem | | |
|---|---|---|---|---|---|---|---|---|---|---|---|---|
| | DGL | Ours | PyG | DGL | Ours | PyG | DGL | Ours | PyG | DGL | Ours | PyG |
| artnet-exp | 2.95 | 0.45 | 4.08 | 2.62 | 1.45 | 4.07 | 494.43 | 78.87 | 827.04 | 702.33 | 148.55 | 1005.33 |
| avazu-ctr | 77.78 | 15.83 | 115.17 | 101.34 | 71.11 | 119.33 | 17071.02 | 337.77 | 27890.18 | 23026.90 | 431.66 | 33786.08 |
| citeseer | 1.60 | 0.26 | 1.33 | 1.34 | 0.55 | 0.84 | 72.38 | 66.69 | 80.94 | 78.17 | 70.78 | 84.69 |
| city-reviews | 9.33 | 2.90 | 13.47 | 13.43 | 12.06 | 14.33 | 2025.84 | 300.48 | 3347.58 | 2768.71 | 482.15 | 4028.52 |
| city-roads-L | 1.65 | 0.50 | 2.76 | 1.85 | 1.44 | 3.16 | 434.45 | 276.44 | 746.35 | 625.13 | 451.29 | 876.25 |
| city-roads-M | 1.62 | 0.30 | 3.55 | 1.42 | 0.81 | 1.57 | 170.71 | 89.71 | 304.76 | 257.59 | 159.67 | 363.09 |
| cora | 2.24 | 0.26 | 1.30 | 1.35 | 0.60 | 0.80 | 41.48 | 33.89 | 49.42 | 48.30 | 37.22 | 53.31 |
| hm-categories | 73.45 | 9.89 | 111.49 | 83.03 | 46.30 | 118.05 | 16607.94 | 250.14 | 27148.81 | 22409.38 | 308.27 | 32898.17 |
| ogbn-arxiv | 10.68 | 3.96 | 7.77 | 10.71 | 10.56 | 8.09 | 1105.40 | 278.94 | 1884.15 | 1548.53 | 487.65 | 2260.83 |
| ogbn-products | OOM | 42.42 | OOM | OOM | 188.69 | OOM | OOM | 4358.62 | OOM | OOM | 7348.55 | OOM |
| pokec-regions | 75.96 | 13.00 | 171.16 | 89.67 | 52.26 | 184.86 | 24989.80 | 2244.59 | 41959.50 | 34234.03 | 4239.20 | 50771.96 |
| pubmed | 1.61 | 0.27 | 1.29 | 1.41 | 0.78 | 1.00 | 136.05 | 75.70 | 203.23 | 184.03 | 99.79 | 234.60 |
| tolokers-2 | 3.69 | 0.77 | 6.16 | 5.49 | 3.32 | 6.03 | 826.11 | 37.96 | 1346.89 | 1139.32 | 52.73 | 1628.36 |
| twitch-views | 53.93 | 11.59 | 70.43 | 70.45 | 54.64 | 75.66 | 10619.01 | 307.29 | 17462.54 | 14405.97 | 515.40 | 21157.31 |
| web-traffic | 207.15 | 85.42 | 87.71 | 766.44 | 866.76 | 116.61 | 24780.82 | 16038.75 | 34650.23 | 30059.66 | 19568.42 | 39218.03 |

*Table 47.* Latency (ms) and Memory footprint (MB) measurements on Gatv2 results on all datasets (heads $H$=2, head dim $D$=32); **lower is better**. OOM indicates that the we couldn't measure the metric for the layer.

| Dataset | Fwd time | | | Bwd time | | | Fwd mem | | | Bwd mem | | |
|---|---|---|---|---|---|---|---|---|---|---|---|---|
| | DGL | Ours | PyG | DGL | Ours | PyG | DGL | Ours | PyG | DGL | Ours | PyG |
| artnet-exp | 2.62 | 0.48 | 6.68 | 3.63 | 1.78 | 7.06 | 945.54 | 137.06 | 1607.31 | 1393.44 | 260.99 | 1932.09 |
| avazu-ctr | 103.87 | 17.69 | 241.28 | 131.92 | 89.70 | 260.53 | 33541.34 | 412.32 | 54848.32 | 44696.43 | 600.90 | 66134.84 |
| citeseer | 1.62 | 0.45 | 1.29 | 1.35 | 0.70 | 0.86 | 81.31 | 70.86 | 98.21 | 99.86 | 79.07 | 113.38 |
| city-reviews | 12.94 | 3.40 | 23.81 | 18.02 | 15.17 | 26.88 | 3858.27 | 446.55 | 6462.25 | 5338.21 | 810.62 | 7766.19 |
| city-roads-L | 7.44 | 0.77 | 4.83 | 3.88 | 2.27 | 6.30 | 731.71 | 414.16 | 1349.38 | 1151.85 | 763.18 | 1599.32 |
| city-roads-M | 1.71 | 0.37 | 2.75 | 1.90 | 1.10 | 2.93 | 305.56 | 145.85 | 571.11 | 548.81 | 285.88 | 726.17 |
| cora | 1.60 | 0.26 | 1.29 | 1.34 | 0.65 | 0.81 | 51.55 | 36.60 | 67.90 | 70.93 | 44.17 | 84.39 |
| hm-categories | 96.11 | 11.10 | 236.02 | 112.45 | 57.52 | 254.35 | 32683.60 | 296.33 | 53427.59 | 43548.03 | 411.27 | 64433.57 |
| ogbn-arxiv | 13.15 | 4.66 | 13.79 | 14.50 | 14.78 | 15.21 | 2082.45 | 445.67 | 3617.28 | 2976.16 | 860.48 | 4341.43 |
| ogbn-products | OOM | 55.55 | OOM | OOM | 261.55 | OOM | OOM | 6750.79 | OOM | OOM | 12730.79 | OOM |
| pokec-regions | 119.57 | 17.89 | OOM | 153.81 | 75.99 | OOM | 48893.76 | 3837.83 | OOM | 66306.77 | 7826.46 | OOM |
| pubmed | 1.61 | 0.28 | 1.59 | 1.43 | 0.84 | 1.64 | 216.57 | 95.77 | 347.76 | 372.78 | 145.49 | 470.78 |
| tolokers-2 | 4.87 | 0.87 | 10.74 | 7.65 | 4.05 | 10.93 | 1608.21 | 49.34 | 2634.98 | 2245.41 | 78.66 | 3174.29 |
| twitch-views | 72.97 | 13.72 | 128.49 | 94.65 | 72.53 | 141.85 | 20883.22 | 472.84 | 34365.08 | 28038.00 | 884.32 | 41441.56 |
| web-traffic | 234.50 | 95.98 | 158.02 | 810.91 | 979.53 | 203.71 | 36208.68 | 18861.76 | 55685.47 | 46328.78 | 25920.80 | 64459.82 |

*Table 48.* Latency (ms) and Memory footprint (MB) measurements on Gatv2 results on all datasets (heads $H$=2, head dim $D$=64); **lower is better**. OOM indicates that the we couldn't measure the metric for the layer.

| Dataset | Fwd time | | | Bwd time | | | Fwd mem | | | Bwd mem | | |
|---|---|---|---|---|---|---|---|---|---|---|---|---|
| | DGL | Ours | PyG | DGL | Ours | PyG | DGL | Ours | PyG | DGL | Ours | PyG |
| artnet-exp | 4.18 | 0.96 | 11.67 | 7.06 | 3.74 | 13.80 | 1821.96 | 227.66 | 3162.62 | 2614.62 | 484.61 | 3797.60 |
| avazu-ctr | OOM | 22.20 | OOM | OOM | 144.44 | OOM | OOM | 562.33 | OOM | OOM | 935.87 | OOM |
| citeseer | 1.67 | 0.27 | 1.27 | 1.30 | 0.68 | 1.54 | 98.83 | 77.29 | 132.90 | 135.05 | 93.91 | 163.52 |
| city-reviews | 18.24 | 5.13 | 44.67 | 28.37 | 26.87 | 52.81 | 7418.60 | 736.96 | 12701.76 | 10255.98 | 1465.96 | 15252.86 |
| city-roads-L | 4.01 | 1.94 | 9.76 | 7.87 | 5.45 | 12.88 | 1309.50 | 693.33 | 2553.32 | 2045.62 | 1389.02 | 3044.80 |
| city-roads-M | 6.44 | 0.86 | 4.79 | 3.86 | 2.41 | 9.09 | 567.71 | 258.13 | 1108.67 | 948.89 | 538.37 | 1327.57 |
| cora | 2.88 | 0.26 | 1.27 | 2.16 | 0.78 | 0.80 | 69.42 | 42.17 | 103.30 | 108.44 | 55.77 | 133.95 |
| hm-categories | OOM | 14.02 | OOM | OOM | 100.72 | OOM | OOM | 386.42 | OOM | OOM | 615.74 | OOM |
| ogbn-arxiv | 17.21 | 7.23 | 26.50 | 21.40 | 27.59 | 31.17 | 3978.51 | 775.38 | 7085.58 | 5701.61 | 1604.51 | 8502.87 |
| ogbn-products | OOM | 101.31 | OOM | OOM | 542.65 | OOM | OOM | 11535.16 | OOM | OOM | 23495.35 | OOM |
| pokec-regions | OOM | 36.32 | OOM | OOM | 159.81 | OOM | OOM | 7028.51 | OOM | OOM | 15003.22 | OOM |
| pubmed | 1.69 | 0.41 | 2.56 | 2.01 | 1.44 | 3.01 | 367.71 | 134.40 | 638.30 | 613.72 | 234.77 | 777.29 |
| tolokers-2 | 7.21 | 1.08 | 18.08 | 10.83 | 6.92 | 20.82 | 3128.51 | 72.79 | 5216.85 | 4322.19 | 131.61 | 6273.00 |
| twitch-views | 102.76 | 18.69 | 296.09 | 142.99 | 127.17 | OOM | 40795.53 | 799.38 | 68171.25 | 55002.73 | 1620.63 | OOM |
| web-traffic | 283.36 | 132.16 | OOM | OOM | 1449.60 | OOM | 58411.46 | 24507.98 | OOM | OOM | 38622.83 | OOM |

*Table 49.* Latency (ms) and Memory footprint (MB) measurements on Gatv2 results on all datasets (heads $H$=2, head dim $D$=128); **lower is better**. OOM indicates that the we couldn't measure the metric for the layer.

| Dataset | Fwd time | | | Bwd time | | | Fwd mem | | | Bwd mem | | |
|---|---|---|---|---|---|---|---|---|---|---|---|---|
| | DGL | Ours | PyG | DGL | Ours | PyG | DGL | Ours | PyG | DGL | Ours | PyG |
| artnet-exp | 8.87 | 2.75 | 24.17 | 14.36 | 8.95 | 28.46 | 3588.85 | 426.46 | 6272.14 | 5082.69 | 925.47 | 7530.85 |
| avazu-ctr | OOM | 41.79 | OOM | OOM | 234.31 | OOM | OOM | 859.11 | OOM | OOM | 1605.43 | OOM |
| citeseer | 1.63 | 0.30 | 1.30 | 1.38 | 0.89 | 1.12 | 134.82 | 90.98 | 202.92 | 206.33 | 124.97 | 263.21 |
| city-reviews | 35.98 | 12.62 | 93.96 | 56.32 | 51.48 | 110.81 | 14609.28 | 1318.24 | 25174.00 | 20086.98 | 2773.62 | 30221.57 |
| city-roads-L | 10.58 | 6.62 | 21.44 | 20.71 | 16.91 | 31.05 | 2476.18 | 1248.39 | 4955.52 | 3834.03 | 2639.43 | 5928.32 |
| city-roads-M | 4.66 | 2.73 | 9.58 | 8.86 | 6.82 | 13.47 | 1094.10 | 481.14 | 2170.76 | 1744.68 | 1040.00 | 2603.48 |
| cora | 1.63 | 0.32 | 1.84 | 1.36 | 1.02 | 1.92 | 108.08 | 53.88 | 176.48 | 184.94 | 81.82 | 238.84 |
| hm-categories | OOM | 26.72 | OOM | OOM | 177.44 | OOM | OOM | 567.96 | OOM | OOM | 1024.18 | OOM |
| ogbn-arxiv | 28.65 | 17.16 | 54.80 | 42.40 | 54.71 | 69.34 | 7808.88 | 1437.34 | 14020.28 | 11085.74 | 3093.08 | 16826.30 |
| ogbn-products | OOM | 258.53 | OOM | OOM | 936.45 | OOM | OOM | 21102.80 | OOM | OOM | 45023.20 | OOM |
| pokec-regions | OOM | 106.76 | OOM | OOM | 365.68 | OOM | OOM | 13405.87 | OOM | OOM | 29355.51 | OOM |
| pubmed | 2.34 | 1.11 | 5.21 | 4.12 | 3.27 | 6.61 | 674.80 | 210.66 | 1214.63 | 1055.90 | 404.71 | 1449.87 |
| tolokers-2 | 12.41 | 2.15 | 35.60 | 18.09 | 10.50 | 40.69 | 6198.51 | 118.73 | 10368.49 | 8563.28 | 235.06 | 12455.81 |
| twitch-views | OOM | 38.47 | OOM | OOM | 218.97 | OOM | OOM | 1455.23 | OOM | OOM | 3098.47 | OOM |
| web-traffic | OOM | 302.61 | OOM | OOM | 2352.66 | OOM | OOM | 35794.49 | OOM | OOM | 64024.34 | OOM |

*Table 50.* Latency (ms) and Memory footprint (MB) measurements on Gatv2 results on all datasets (heads $H$=2, head dim $D$=256); **lower is better**. OOM indicates that the we couldn't measure the metric for the layer.

| Dataset | Fwd time | | | Bwd time | | | Fwd mem | | | Bwd mem | | |
|---|---|---|---|---|---|---|---|---|---|---|---|---|
| | DGL | Ours | PyG | DGL | Ours | PyG | DGL | Ours | PyG | DGL | Ours | PyG |
| artnet-exp | 3.20 | 0.67 | 6.96 | 3.93 | 2.21 | 7.50 | 943.62 | 125.09 | 1605.78 | 1377.29 | 236.67 | 1926.04 |
| avazu-ctr | 117.14 | 20.89 | 256.48 | 144.25 | 95.72 | 276.19 | 33699.59 | 394.59 | 55007.15 | 44994.28 | 564.50 | 66769.49 |
| citeseer | 1.62 | 0.26 | 1.28 | 1.50 | 0.56 | 0.81 | 80.93 | 70.02 | 97.85 | 98.28 | 77.38 | 112.04 |
| city-reviews | 14.22 | 3.73 | 25.00 | 18.89 | 15.70 | 27.88 | 3858.61 | 411.17 | 6463.12 | 5301.40 | 738.86 | 7769.11 |
| city-roads-L | 1.99 | 0.82 | 4.87 | 3.11 | 3.03 | 5.79 | 716.93 | 381.56 | 1335.82 | 1087.05 | 696.43 | 1543.43 |
| city-roads-M | 1.61 | 0.42 | 2.51 | 1.77 | 1.16 | 3.01 | 299.17 | 132.17 | 566.43 | 523.19 | 258.15 | 703.93 |
| cora | 1.59 | 0.26 | 1.30 | 1.50 | 0.61 | 0.84 | 51.25 | 35.91 | 67.62 | 69.68 | 42.77 | 83.38 |
| hm-categories | 106.17 | 14.76 | 249.49 | 123.27 | 67.96 | 270.14 | 32841.14 | 284.64 | 53585.64 | 43853.19 | 388.03 | 65067.32 |
| ogbn-arxiv | 15.57 | 4.73 | 14.21 | 14.46 | 13.24 | 15.71 | 2070.63 | 404.23 | 3606.09 | 2911.18 | 777.00 | 4298.75 |
| ogbn-products | OOM | 83.03 | OOM | OOM | 372.46 | OOM | OOM | 6171.42 | OOM | OOM | 11553.38 | OOM |
| pokec-regions | 137.18 | 24.56 | OOM | 166.83 | 99.58 | OOM | 48928.03 | 3452.97 | OOM | 65976.67 | 7043.60 | OOM |
| pubmed | 1.61 | 0.30 | 1.75 | 1.40 | 0.95 | 1.70 | 214.78 | 91.06 | 346.45 | 363.50 | 135.92 | 465.27 |
| tolokers-2 | 6.01 | 1.00 | 10.64 | 7.89 | 4.28 | 11.90 | 1614.65 | 47.08 | 2641.51 | 2256.90 | 72.96 | 3200.36 |
| twitch-views | 80.46 | 15.72 | 136.13 | 100.63 | 74.20 | 151.46 | 20966.38 | 430.57 | 34448.56 | 28163.26 | 800.01 | 41778.44 |
| web-traffic | 245.84 | 94.28 | 165.64 | 821.84 | 890.48 | 206.15 | 35954.19 | 18177.41 | 55452.69 | 45114.16 | 24530.40 | 63529.11 |

*Table 51.* Latency (ms) and Memory footprint (MB) measurements on Gatv2 results on all datasets (heads $H$=4, head dim $D$=32); **lower is better**. OOM indicates that the we couldn't measure the metric for the layer.

| Dataset | Fwd time | | | Bwd time | | | Fwd mem | | | Bwd mem | | |
|---|---|---|---|---|---|---|---|---|---|---|---|---|
| | DGL | Ours | PyG | DGL | Ours | PyG | DGL | Ours | PyG | DGL | Ours | PyG |
| artnet-exp | 4.44 | 0.86 | 11.84 | 6.03 | 3.16 | 13.26 | 1821.91 | 211.37 | 3154.79 | 2572.80 | 435.40 | 3765.88 |
| avazu-ctr | 162.27 | 24.38 | OOM | OOM | 126.03 | OOM | 65965.63 | 524.18 | OOM | OOM | 859.53 | OOM |
| citeseer | 1.61 | 0.26 | 1.39 | 1.37 | 0.63 | 0.85 | 98.06 | 75.00 | 132.00 | 131.06 | 89.81 | 159.57 |
| city-reviews | 20.39 | 4.67 | 46.27 | 27.41 | 20.69 | 52.43 | 7435.77 | 665.61 | 12684.33 | 10144.40 | 1321.77 | 15182.16 |
| city-roads-L | 3.48 | 1.46 | 9.26 | 6.00 | 4.47 | 11.18 | 1280.56 | 623.16 | 2520.54 | 1908.64 | 1248.66 | 2918.00 |
| city-roads-M | 1.90 | 0.65 | 4.25 | 2.86 | 1.94 | 5.11 | 558.21 | 230.38 | 1096.61 | 896.56 | 482.50 | 1277.32 |
| cora | 1.61 | 0.26 | 1.30 | 1.36 | 0.68 | 0.81 | 69.27 | 41.11 | 103.14 | 105.61 | 54.07 | 131.33 |
| hm-categories | 152.93 | 17.08 | OOM | OOM | 87.87 | OOM | 64329.83 | 364.99 | OOM | OOM | 572.02 | OOM |
| ogbn-arxiv | 17.83 | 6.01 | 25.98 | 19.88 | 20.50 | 29.69 | 3964.80 | 695.09 | 7054.90 | 5552.85 | 1442.00 | 8377.91 |
| ogbn-products | OOM | 108.13 | OOM | OOM | 515.36 | OOM | OOM | 10357.56 | OOM | OOM | 21121.66 | OOM |
| pokec-regions | OOM | 34.64 | OOM | OOM | 146.38 | OOM | OOM | 6242.38 | OOM | OOM | 13418.91 | OOM |
| pubmed | 2.24 | 0.36 | 2.55 | 2.36 | 1.17 | 3.63 | 365.87 | 124.90 | 634.12 | 594.94 | 214.71 | 761.42 |
| tolokers-2 | 8.47 | 1.18 | 18.71 | 11.13 | 5.44 | 21.42 | 3149.38 | 66.35 | 5222.20 | 4326.67 | 120.68 | 6292.79 |
| twitch-views | 115.15 | 19.01 | 303.02 | 145.82 | 98.16 | OOM | 41065.93 | 719.93 | 68235.99 | 55044.85 | 1459.86 | OOM |
| web-traffic | 291.61 | 111.57 | OOM | OOM | 1032.47 | OOM | 58021.34 | 23117.91 | OOM | OOM | 35820.92 | OOM |

*Table 52.* Latency (ms) and Memory footprint (MB) measurements on Gatv2 results on all datasets (heads $H$=4, head dim $D$=64); **lower is better**. OOM indicates that the we couldn't measure the metric for the layer.

| Dataset | Fwd time | | | Bwd time | | | Fwd mem | | | Bwd mem | | |
|---|---|---|---|---|---|---|---|---|---|---|---|---|
| | DGL | Ours | PyG | DGL | Ours | PyG | DGL | Ours | PyG | DGL | Ours | PyG |
| artnet-exp | 8.14 | 1.77 | 22.65 | 12.33 | 7.03 | 26.34 | 3576.45 | 383.44 | 6250.66 | 4991.04 | 827.21 | 7448.78 |
| avazu-ctr | OOM | 30.88 | OOM | OOM | 208.08 | OOM | OOM | 786.80 | OOM | OOM | 1458.72 | OOM |
| citeseer | 1.61 | 0.27 | 2.31 | 1.28 | 0.74 | 3.02 | 133.05 | 87.44 | 200.57 | 200.03 | 117.43 | 256.31 |
| city-reviews | 34.58 | 7.73 | 92.45 | 51.31 | 40.24 | 105.30 | 14591.98 | 1174.61 | 25120.47 | 19831.80 | 2483.92 | 30006.08 |
| city-roads-L | 7.73 | 3.78 | 18.37 | 14.42 | 11.09 | 25.32 | 2415.84 | 1111.34 | 4888.96 | 3561.08 | 2363.55 | 5662.83 |
| city-roads-M | 3.54 | 1.63 | 8.54 | 6.44 | 4.62 | 10.86 | 1070.20 | 425.88 | 2144.20 | 1634.45 | 929.97 | 2495.10 |
| cora | 2.86 | 0.26 | 1.28 | 1.39 | 0.83 | 1.07 | 106.27 | 50.60 | 174.30 | 179.18 | 75.15 | 231.63 |
| hm-categories | OOM | 21.73 | OOM | OOM | 158.69 | OOM | OOM | 523.15 | OOM | OOM | 933.26 | OOM |
| ogbn-arxiv | 26.76 | 10.06 | 51.05 | 36.15 | 37.97 | 60.82 | 7753.66 | 1273.03 | 13947.06 | 10771.52 | 2762.65 | 16534.83 |
| ogbn-products | OOM | 201.19 | OOM | OOM | 1068.14 | OOM | OOM | 18728.74 | OOM | OOM | 40256.57 | OOM |
| pokec-regions | OOM | 71.22 | OOM | OOM | 315.17 | OOM | OOM | 11824.94 | OOM | OOM | 26180.03 | OOM |
| pubmed | 1.98 | 0.67 | 4.68 | 3.31 | 2.37 | 5.83 | 669.71 | 192.92 | 1207.26 | 1021.95 | 368.69 | 1415.35 |
| tolokers-2 | 13.34 | 1.55 | 35.93 | 19.39 | 10.18 | 41.37 | 6217.18 | 107.26 | 10369.23 | 8555.47 | 211.42 | 12462.60 |
| twitch-views | OOM | 25.95 | OOM | OOM | 181.90 | OOM | OOM | 1293.19 | OOM | OOM | 2771.51 | OOM |
| web-traffic | OOM | 171.66 | OOM | OOM | 1645.54 | OOM | OOM | 32998.17 | OOM | OOM | 58405.40 | OOM |

*Table 53.* Latency (ms) and Memory footprint (MB) measurements on Gatv2 results on all datasets (heads $H$=4, head dim $D$=128); **lower is better**. OOM indicates that the we couldn't measure the metric for the layer.

| Dataset | Fwd time | | | Bwd time | | | Fwd mem | | | Bwd mem | | |
|---|---|---|---|---|---|---|---|---|---|---|---|---|
| | DGL | Ours | PyG | DGL | Ours | PyG | DGL | Ours | PyG | DGL | Ours | PyG |
| artnet-exp | 17.83 | 5.43 | 47.44 | 27.70 | 17.87 | 56.35 | 7083.12 | 729.59 | 12445.20 | 9766.47 | 1619.32 | 14814.95 |
| avazu-ctr | OOM | 58.52 | OOM | OOM | 344.10 | OOM | OOM | 1308.08 | OOM | OOM | 2652.55 | OOM |
| citeseer | 1.70 | 0.45 | 1.63 | 1.65 | 1.31 | 1.92 | 202.89 | 111.93 | 338.91 | 335.85 | 173.42 | 449.74 |
| city-reviews | 70.30 | 20.95 | 222.12 | 107.03 | 81.34 | 256.46 | 28904.31 | 2193.10 | 49985.97 | 39207.38 | 4813.51 | 59643.22 |
| city-roads-L | 20.93 | 13.19 | 42.64 | 40.74 | 33.60 | 60.69 | 4673.70 | 2084.51 | 9631.70 | 6879.97 | 4589.36 | 11161.16 |
| city-roads-M | 9.16 | 5.44 | 18.83 | 16.71 | 14.01 | 25.76 | 2098.80 | 817.49 | 4248.80 | 3120.14 | 1823.73 | 4945.10 |
| cora | 1.67 | 0.45 | 2.43 | 1.65 | 1.46 | 1.88 | 182.28 | 71.27 | 318.19 | 327.69 | 121.87 | 434.36 |
| hm-categories | OOM | 41.89 | OOM | OOM | 286.32 | OOM | OOM | 842.65 | OOM | OOM | 1664.14 | OOM |
| ogbn-arxiv | 50.26 | 25.75 | 109.58 | 78.36 | 81.08 | 136.63 | 15333.87 | 2432.88 | 27741.65 | 21210.79 | 5414.15 | 32857.93 |
| ogbn-products | OOM | 513.45 | OOM | OOM | OOM | OOM | OOM | 35472.99 | OOM | OOM | OOM | OOM |
| pokec-regions | OOM | 211.68 | OOM | OOM | 740.04 | OOM | OOM | 22986.11 | OOM | OOM | 51690.72 | OOM |
| pubmed | 4.33 | 2.05 | 9.81 | 7.53 | 5.96 | 12.65 | 1274.92 | 327.84 | 2354.64 | 1868.35 | 677.43 | 2767.32 |
| tolokers-2 | 24.14 | 3.36 | 70.59 | 33.97 | 16.96 | 81.62 | 12351.17 | 189.22 | 20674.93 | 16758.58 | 398.91 | 24815.12 |
| twitch-views | OOM | 56.79 | OOM | OOM | 322.57 | OOM | OOM | 2443.06 | OOM | OOM | 5402.26 | OOM |
| web-traffic | OOM | 435.45 | OOM | OOM | OOM | OOM | OOM | 52752.75 | OOM | OOM | OOM | OOM |

*Table 54.* Latency (ms) and Memory footprint (MB) measurements on Gatv2 results on all datasets (heads $H$=4, head dim $D$=256); **lower is better**. OOM indicates that the we couldn't measure the metric for the layer.

| Dataset | Fwd time | | | Bwd time | | | Fwd mem | | | Bwd mem | | |
|---|---|---|---|---|---|---|---|---|---|---|---|---|
| | DGL | Ours | PyG | DGL | Ours | PyG | DGL | Ours | PyG | DGL | Ours | PyG |
| artnet-exp | 5.11 | 1.07 | 12.26 | 6.46 | 3.95 | 14.36 | 1824.22 | 199.76 | 3157.87 | 2565.11 | 412.16 | 3778.38 |
| avazu-ctr | 189.89 | 30.91 | OOM | OOM | 141.89 | OOM | 66291.41 | 506.65 | OOM | OOM | 823.29 | OOM |
| citeseer | 1.61 | 0.26 | 1.29 | 1.35 | 0.61 | 0.80 | 97.70 | 74.15 | 131.69 | 129.53 | 88.05 | 158.52 |
| city-reviews | 23.23 | 5.51 | 49.43 | 29.53 | 22.76 | 55.06 | 7453.07 | 631.46 | 12704.07 | 10142.68 | 1251.20 | 15260.66 |
| city-roads-L | 3.47 | 1.56 | 9.43 | 8.39 | 4.82 | 10.99 | 1267.86 | 591.78 | 2510.86 | 1848.68 | 1183.08 | 2874.70 |
| city-roads-M | 1.88 | 0.70 | 4.70 | 2.68 | 2.15 | 5.53 | 553.17 | 217.09 | 1091.42 | 871.69 | 455.12 | 1259.52 |
| cora | 1.61 | 0.27 | 1.30 | 1.35 | 0.71 | 0.81 | 69.00 | 40.40 | 102.92 | 104.42 | 52.60 | 130.63 |
| hm-categories | 177.98 | 24.24 | OOM | OOM | 111.89 | OOM | 64651.07 | 353.14 | OOM | OOM | 548.58 | OOM |
| ogbn-arxiv | 20.32 | 6.17 | 27.56 | 21.13 | 19.00 | 30.51 | 3961.83 | 654.93 | 7053.85 | 5505.56 | 1359.74 | 8376.55 |
| ogbn-products | OOM | 164.66 | OOM | OOM | 743.15 | OOM | OOM | 9796.84 | OOM | OOM | 19962.84 | OOM |
| pokec-regions | OOM | 48.32 | OOM | OOM | 196.41 | OOM | OOM | 5870.11 | OOM | OOM | 12648.58 | OOM |
| pubmed | 1.62 | 0.42 | 2.67 | 1.78 | 1.21 | 2.91 | 364.35 | 120.29 | 633.27 | 587.75 | 205.20 | 758.97 |
| tolokers-2 | 9.68 | 1.50 | 20.08 | 12.38 | 6.39 | 22.85 | 3163.69 | 64.53 | 5236.69 | 4352.01 | 114.46 | 6350.94 |
| twitch-views | 131.48 | 23.58 | 321.40 | 160.16 | 109.79 | OOM | 41252.76 | 680.04 | 68424.42 | 55377.47 | 1377.87 | OOM |
| web-traffic | 292.95 | 113.95 | OOM | OOM | 970.25 | OOM | 57865.19 | 22455.56 | OOM | OOM | 34452.48 | OOM |

*Table 55.* Latency (ms) and Memory footprint (MB) measurements on Gatv2 results on all datasets (heads $H$=8, head dim $D$=32); **lower is better**. OOM indicates that the we couldn't measure the metric for the layer.

| Dataset | Fwd time | | | Bwd time | | | Fwd mem | | | Bwd mem | | |
|---|---|---|---|---|---|---|---|---|---|---|---|---|
| | DGL | Ours | PyG | DGL | Ours | PyG | DGL | Ours | PyG | DGL | Ours | PyG |
| artnet-exp | 8.38 | 1.60 | 22.92 | 11.70 | 5.91 | 26.14 | 3572.33 | 359.24 | 6247.31 | 4958.18 | 778.02 | 7436.10 |
| avazu-ctr | OOM | 37.30 | OOM | OOM | 188.17 | OOM | OOM | 748.83 | OOM | OOM | 1382.37 | OOM |
| citeseer | 1.61 | 0.26 | 1.29 | 1.37 | 0.69 | 0.93 | 132.00 | 84.99 | 199.57 | 195.59 | 113.49 | 252.28 |
| city-reviews | 37.38 | 7.31 | 94.37 | 49.96 | 31.38 | 105.60 | 14590.75 | 1104.07 | 25121.77 | 19756.78 | 2340.36 | 30011.20 |
| city-roads-L | 6.79 | 2.83 | 17.57 | 11.55 | 8.75 | 21.83 | 2383.60 | 1042.60 | 4860.46 | 3428.53 | 2224.43 | 5548.33 |
| city-roads-M | 3.11 | 1.23 | 8.02 | 5.19 | 3.66 | 9.79 | 1059.20 | 398.38 | 2132.72 | 1583.29 | 874.16 | 2450.13 |
| cora | 1.61 | 0.26 | 1.35 | 1.37 | 0.74 | 0.96 | 106.70 | 49.28 | 174.05 | 176.42 | 73.00 | 229.04 |
| hm-categories | OOM | 29.04 | OOM | OOM | 147.24 | OOM | OOM | 501.89 | OOM | OOM | 889.52 | OOM |
| ogbn-arxiv | 28.46 | 8.39 | 51.52 | 33.97 | 29.13 | 59.52 | 7729.55 | 1193.86 | 13926.38 | 10640.80 | 2601.08 | 16451.54 |
| ogbn-products | OOM | 214.50 | OOM | OOM | 1025.04 | OOM | OOM | 17569.64 | OOM | OOM | 37901.19 | OOM |
| pokec-regions | OOM | 68.45 | OOM | OOM | 287.38 | OOM | OOM | 11051.26 | OOM | OOM | 24607.97 | OOM |
| pubmed | 2.98 | 0.55 | 4.91 | 3.17 | 1.89 | 5.92 | 666.64 | 183.39 | 1202.98 | 1005.42 | 348.41 | 1401.66 |
| tolokers-2 | 14.81 | 1.80 | 36.74 | 20.13 | 8.50 | 41.92 | 6229.78 | 100.72 | 10382.00 | 8575.32 | 199.16 | 12514.58 |
| twitch-views | OOM | 28.60 | OOM | OOM | 144.97 | OOM | OOM | 1215.80 | OOM | OOM | 2612.61 | OOM |
| web-traffic | OOM | 142.72 | OOM | OOM | 1141.86 | OOM | OOM | 31629.96 | OOM | OOM | 55625.17 | OOM |

*Table 56.* Latency (ms) and Memory footprint (MB) measurements on Gatv2 results on all datasets (heads $H$=8, head dim $D$=64); **lower is better**. OOM indicates that the we couldn't measure the metric for the layer.

| Dataset | Fwd time | | | Bwd time | | | Fwd mem | | | Bwd mem | | |
|---|---|---|---|---|---|---|---|---|---|---|---|---|
| | DGL | Ours | PyG | DGL | Ours | PyG | DGL | Ours | PyG | DGL | Ours | PyG |
| artnet-exp | 16.11 | 3.40 | 45.15 | 24.80 | 13.55 | 52.84 | 7065.31 | 680.52 | 12427.64 | 9681.88 | 1519.03 | 14750.03 |
| avazu-ctr | OOM | 47.26 | OOM | OOM | 335.34 | OOM | OOM | 1235.75 | OOM | OOM | 2504.50 | OOM |
| citeseer | 1.61 | 0.30 | 1.66 | 1.37 | 0.99 | 1.65 | 201.15 | 107.74 | 335.90 | 329.31 | 164.48 | 441.93 |
| city-reviews | 66.22 | 13.35 | 216.13 | 97.07 | 65.50 | 244.66 | 28866.76 | 2049.17 | 49952.27 | 38985.92 | 4523.24 | 59501.74 |
| city-roads-L | 15.36 | 7.64 | 36.56 | 28.75 | 21.39 | 49.65 | 4609.10 | 1948.26 | 9567.71 | 6609.78 | 4313.22 | 10905.86 |
| city-roads-M | 6.90 | 3.15 | 16.24 | 12.50 | 9.03 | 21.34 | 2072.08 | 761.51 | 4221.71 | 3010.18 | 1710.76 | 4842.03 |
| cora | 1.69 | 0.30 | 1.53 | 1.44 | 1.05 | 1.64 | 181.91 | 68.11 | 315.83 | 322.89 | 115.41 | 426.62 |
| hm-categories | OOM | 36.99 | OOM | OOM | 277.49 | OOM | OOM | 797.67 | OOM | OOM | 1572.64 | OOM |
| ogbn-arxiv | 47.25 | 15.18 | 103.82 | 66.33 | 56.96 | 122.79 | 15268.71 | 2268.61 | 27677.86 | 20913.37 | 5082.98 | 32605.20 |
| ogbn-products | OOM | 400.81 | OOM | OOM | OOM | OOM | OOM | 33116.86 | OOM | OOM | OOM | OOM |
| pokec-regions | OOM | 141.55 | OOM | OOM | 629.71 | OOM | OOM | 21417.05 | OOM | OOM | 48526.37 | OOM |
| pubmed | 3.64 | 1.25 | 9.03 | 6.00 | 4.07 | 10.88 | 1267.50 | 309.51 | 2347.90 | 1830.92 | 643.01 | 2733.23 |
| tolokers-2 | 24.93 | 2.55 | 70.89 | 36.33 | 16.65 | 81.80 | 12359.77 | 177.22 | 20683.61 | 16764.29 | 374.21 | 24852.99 |
| twitch-views | OOM | 40.61 | OOM | OOM | 287.10 | OOM | OOM | 2282.16 | OOM | OOM | 5075.23 | OOM |
| web-traffic | OOM | 249.49 | OOM | OOM | OOM | OOM | OOM | 49977.74 | OOM | OOM | OOM | OOM |

*Table 57.* Latency (ms) and Memory footprint (MB) measurements on Gatv2 results on all datasets (heads $H$=8, head dim $D$=128); **lower is better**. OOM indicates that the we couldn't measure the metric for the layer.

| Dataset | Fwd time | | | Bwd time | | | Fwd mem | | | Bwd mem | | |
|---|---|---|---|---|---|---|---|---|---|---|---|---|
| | DGL | Ours | PyG | DGL | Ours | PyG | DGL | Ours | PyG | DGL | Ours | PyG |
| artnet-exp | 35.27 | 10.50 | 93.34 | 57.13 | 35.04 | 115.55 | 14059.40 | 1324.30 | 24782.87 | 19141.43 | 3004.76 | 29372.63 |
| avazu-ctr | OOM | 91.15 | OOM | OOM | 563.65 | OOM | OOM | 2206.03 | OOM | OOM | 4744.99 | OOM |
| citeseer | 1.79 | 0.77 | 2.78 | 2.63 | 2.12 | 3.61 | 338.72 | 153.99 | 610.89 | 560.31 | 270.52 | 736.47 |
| city-reviews | 132.97 | 37.81 | OOM | OOM | 138.49 | OOM | 57420.90 | 3942.78 | OOM | OOM | 8890.42 | OOM |
| city-roads-L | 41.53 | 26.21 | 84.57 | 83.45 | 66.64 | 123.51 | 9063.55 | 3756.76 | 18984.14 | 12923.70 | 8488.57 | 21625.22 |
| city-roads-M | 17.82 | 10.62 | 36.57 | 35.36 | 27.73 | 53.06 | 4103.65 | 1490.19 | 8405.04 | 5893.14 | 3391.54 | 9632.29 |
| cora | 1.76 | 0.71 | 2.77 | 2.50 | 2.13 | 3.58 | 330.30 | 106.04 | 601.62 | 556.20 | 201.97 | 725.71 |
| hm-categories | OOM | 71.21 | OOM | OOM | 502.60 | OOM | OOM | 1392.02 | OOM | OOM | 2944.92 | OOM |
| ogbn-arxiv | 94.29 | 43.14 | 257.79 | 154.65 | 131.01 | 315.25 | 30345.66 | 4422.47 | 55168.37 | 41461.39 | 10051.20 | 64904.61 |
| ogbn-products | OOM | 1025.55 | OOM | OOM | OOM | OOM | OOM | 64212.92 | OOM | OOM | OOM | OOM |
| pokec-regions | OOM | 421.01 | OOM | OOM | OOM | OOM | OOM | 42148.86 | OOM | OOM | OOM | OOM |
| pubmed | 8.33 | 3.93 | 18.84 | 14.93 | 11.09 | 24.91 | 2472.16 | 562.21 | 4627.63 | 3493.25 | 1222.88 | 5393.89 |
| tolokers-2 | 48.21 | 5.67 | 172.27 | 66.68 | 29.48 | 196.14 | 24627.81 | 330.19 | 41292.72 | 33155.81 | 726.60 | 49539.98 |
| twitch-views | OOM | 93.15 | OOM | OOM | 526.96 | OOM | OOM | 4418.71 | OOM | OOM | 10009.07 | OOM |

*Table 58.* Latency (ms) and Memory footprint (MB) measurements on Gatv2 results on all datasets (heads $H$=8, head dim $D$=256); **lower is better**. OOM indicates that the we couldn't measure the metric for the layer.

| Dataset | Fwd time | | | | | | | | Bwd time | | | | | | | |
|---|---|---|---|---|---|---|---|---|---|---|---|---|---|---|---|---|
| | DGL | cuSPARSE | cuSPARSE+ $A^\top$ | FuseGNN | TC-GNN | Ours (WSB/TC) | PyG | Pytorch CSR-SpMM | DGL | cuSPARSE | cuSPARSE+ $A^\top$ | FuseGNN | TC-GNN | Ours (WSB/TC) | PyG | Pytorch CSR-SpMM |
| artnet-exp | 1.21 | 0.17 | 0.17 | 4.00 | 0.50 | 0.24 | 1.62 | 0.19 | 0.38 | 0.23 | 0.25 | 0.29 | 0.67 | 0.27 | 1.09 | 0.87 |
| avazu-ctr | 3.26 | 2.24 | 2.23 | 18.23 | 343.32 | 2.07 | 34.19 | 2.63 | 2.73 | 3.68 | 2.27 | 10.88 | 338.45 | 2.71 | 31.81 | 16.03 |
| citeseer | 1.15 | 0.12 | 0.10 | 2.10 | 0.09 | 0.21 | 0.79 | 0.08 | 0.38 | 0.20 | 0.19 | 0.19 | 0.45 | 0.21 | 0.19 | 0.72 |
| city-reviews | 1.30 | 0.51 | 0.49 | 5.44 | 5.09 | 0.57 | 4.29 | 0.55 | 0.78 | 0.60 | 0.60 | 1.71 | 4.42 | 1.61 | 3.84 | 2.55 |
| city-roads-L | 1.17 | 0.20 | 0.19 | 2.82 | 0.22 | 0.26 | 1.31 | 0.19 | 0.46 | 0.33 | 0.30 | 0.34 | 0.47 | 0.33 | 0.85 | 0.75 |
| city-roads-M | 1.15 | 0.14 | 0.13 | 4.21 | 0.11 | 0.25 | 0.91 | 0.13 | 0.38 | 0.22 | 0.18 | 0.19 | 0.39 | 0.23 | 0.42 | 0.62 |
| cora | 1.15 | 0.11 | 0.10 | 2.08 | 0.10 | 0.25 | 0.79 | 0.09 | 0.38 | 0.23 | 0.20 | 0.19 | 0.45 | 0.21 | 0.20 | 0.75 |
| hm-categories | 3.35 | 2.35 | 2.34 | 13.07 | 843.59 | 1.66 | 34.13 | 2.72 | 2.79 | 3.21 | 2.26 | 6.04 | 838.68 | 2.85 | 31.64 | 19.59 |
| ogbn-arxiv | 1.16 | 0.36 | 0.35 | 5.20 | 0.68 | 0.72 | 2.71 | 0.38 | 0.63 | 0.50 | 0.44 | 0.53 | OOM | 0.48 | 2.21 | 2.11 |
| ogbn-products | 20.17 | 23.99 | 23.88 | 86.71 | 85.01 | 13.09 | 274.38 | 26.30 | 20.45 | 40.21 | 25.02 | 33.10 | 59.82 | 27.51 | OOM | 103.97 |
| pokec-regions | 7.95 | 6.18 | 6.17 | 21.09 | 16.40 | 3.45 | 55.66 | 6.97 | 7.75 | 9.46 | 6.85 | 8.83 | 11.51 | 7.52 | 50.28 | 25.49 |
| pubmed | 1.18 | 0.10 | 0.10 | 2.94 | 0.16 | 0.21 | 0.80 | 0.09 | 0.35 | 0.22 | 0.19 | 0.15 | 0.43 | 0.20 | 0.26 | 0.60 |
| tolokers-2 | 1.15 | 0.17 | 0.16 | 3.06 | 1.23 | 0.64 | 2.12 | 0.18 | 0.36 | 0.41 | 0.21 | 0.41 | 1.33 | 0.31 | 1.62 | 1.25 |
| twitch-views | 3.06 | 2.17 | 2.13 | 13.96 | 139.53 | 2.04 | 22.23 | 2.40 | 2.55 | 2.43 | 2.25 | 8.09 | 138.14 | 2.53 | 20.56 | 11.25 |
| web-traffic | 5.51 | 2.40 | 2.28 | 69.50 | 84477.44 | 17.04 | 28.09 | 3.06 | 7.03 | 11.08 | 13.98 | 234.44 | 37505.82 | 17.47 | 34.93 | 44.00 |

*Table 59.* Latency (ms) measurements on SpMM (with GCN as the operator) on all datasets (hidden dim $D$=64); **lower is better**. OOM indicates that the we couldn't measure the metric for the layer.

| Dataset | Fwd time | | | | | | | | Bwd time | | | | | | | |
|---|---|---|---|---|---|---|---|---|---|---|---|---|---|---|---|---|
| | DGL | cuSPARSE | cuSPARSE+ $A^\top$ | FuseGNN | TC-GNN | Ours (WSB/TC) | PyG | Pytorch CSR-SpMM | DGL | cuSPARSE | cuSPARSE+ $A^\top$ | FuseGNN | TC-GNN | Ours (WSB/TC) | PyG | Pytorch CSR-SpMM |
| artnet-exp | 1.13 | 0.19 | 0.19 | 4.20 | 0.67 | 0.25 | 2.53 | 0.20 | 0.45 | 0.35 | 0.28 | 0.40 | 0.76 | 0.32 | 2.04 | 0.92 |
| avazu-ctr | 3.70 | 2.68 | 2.72 | 27.65 | 349.08 | 2.33 | 77.91 | 2.90 | 3.10 | 6.38 | 2.68 | 19.93 | 342.22 | 3.14 | 75.63 | 16.40 |
| citeseer | 1.15 | 0.08 | 0.08 | 2.06 | 0.08 | 0.20 | 0.78 | 0.07 | 0.38 | 0.16 | 0.16 | 0.14 | 0.37 | 0.22 | 0.18 | 0.57 |
| city-reviews | 1.38 | 0.58 | 0.57 | 5.81 | 6.16 | 0.70 | 7.98 | 0.65 | 0.95 | 1.12 | 0.74 | 3.21 | 4.51 | 0.86 | 7.51 | 2.69 |
| city-roads-L | 1.19 | 0.23 | 0.22 | 2.84 | 0.29 | 0.30 | 1.96 | 0.24 | 0.62 | 0.49 | 0.40 | 0.45 | 0.59 | 0.43 | 1.55 | 0.86 |
| city-roads-M | 1.14 | 0.15 | 0.13 | 4.00 | 0.14 | 0.21 | 1.21 | 0.14 | 0.41 | 0.32 | 0.23 | 0.23 | 0.41 | 0.25 | 0.75 | 0.64 |
| cora | 1.13 | 0.11 | 0.08 | 2.07 | 0.10 | 0.19 | 0.78 | 0.09 | 0.37 | 0.30 | 0.17 | 0.17 | 0.37 | 0.24 | 0.19 | 0.73 |
| hm-categories | 3.81 | 2.86 | 2.70 | 17.47 | 852.74 | 1.99 | 77.03 | 3.00 | 3.29 | 5.92 | 2.78 | 10.39 | 839.54 | 3.34 | 74.09 | 19.78 |
| ogbn-arxiv | 1.22 | 0.43 | 0.42 | 7.45 | 0.95 | 0.91 | 4.64 | 0.47 | 0.78 | 0.89 | 0.55 | 0.85 | OOM | 0.61 | 4.22 | 2.23 |
| ogbn-products | 24.78 | 37.48 | 37.45 | 113.76 | 121.71 | 23.21 | OOM | 40.59 | 25.94 | 84.75 | 39.62 | 61.20 | 78.57 | 41.04 | OOM | 117.39 |
| pokec-regions | 9.96 | 8.99 | 9.03 | 27.29 | 22.16 | 6.25 | 123.78 | 9.84 | 10.74 | 20.07 | 10.28 | 15.74 | 16.03 | 11.09 | 119.47 | 29.09 |
| pubmed | 1.13 | 0.10 | 0.08 | 2.23 | 0.17 | 0.20 | 0.94 | 0.09 | 0.36 | 0.28 | 0.16 | 0.18 | 0.38 | 0.20 | 0.43 | 0.59 |
| tolokers-2 | 1.12 | 0.19 | 0.19 | 3.40 | 1.50 | 0.36 | 3.50 | 0.21 | 0.38 | 0.43 | 0.23 | 0.83 | 1.54 | 0.60 | 3.06 | 1.25 |
| twitch-views | 3.49 | 2.65 | 2.64 | 20.99 | 144.38 | 2.53 | 42.17 | 2.81 | 2.90 | 5.45 | 2.77 | 15.82 | 137.86 | 3.23 | 41.02 | 11.53 |
| web-traffic | 8.85 | 3.55 | 3.52 | 125.18 | 84730.25 | 20.99 | 51.86 | 4.45 | 11.38 | 16.00 | 16.43 | 447.99 | 37535.07 | 20.55 | 58.81 | 46.60 |

*Table 60.* Latency (ms) measurements on SpMM (with GCN as the operator) on all datasets (hidden dim $D$=128); **lower is better**. OOM indicates that the we couldn't measure the metric for the layer.

| Dataset | Fwd time | | | | | | | | Bwd time | | | | | | | |
|---|---|---|---|---|---|---|---|---|---|---|---|---|---|---|---|---|
| | DGL | cuSPARSE | cuSPARSE+ $A^\top$ | FuseGNN | TC-GNN | Ours (WSB/TC) | PyG | Pytorch CSR-SpMM | DGL | cuSPARSE | cuSPARSE+ $A^\top$ | FuseGNN | TC-GNN | Ours (WSB/TC) | PyG | Pytorch CSR-SpMM |
| artnet-exp | 1.16 | 0.30 | 0.30 | 4.20 | 0.70 | 0.29 | 4.34 | 0.33 | 0.63 | 0.58 | 0.44 | 0.46 | 0.83 | 0.48 | 4.08 | 1.10 |
| avazu-ctr | 6.09 | 5.15 | 4.94 | 23.32 | 351.03 | 2.75 | 157.89 | 5.59 | 5.76 | 11.70 | 5.10 | 16.21 | 342.22 | 5.73 | 158.46 | 19.14 |
| citeseer | 1.15 | 0.08 | 0.07 | 2.92 | 0.08 | 0.18 | 0.79 | 0.07 | 0.38 | 0.16 | 0.15 | 0.19 | 0.36 | 0.19 | 0.18 | 0.56 |
| city-reviews | 1.96 | 1.07 | 1.08 | 5.70 | 5.77 | 1.01 | 15.17 | 1.20 | 1.71 | 2.08 | 1.36 | 3.27 | 4.74 | 2.83 | 14.98 | 3.27 |
| city-roads-L | 1.34 | 0.36 | 0.36 | 3.07 | 0.34 | 0.48 | 3.31 | 0.43 | 1.11 | 0.77 | 0.67 | 0.69 | 0.76 | 0.75 | 3.00 | 1.16 |
| city-roads-M | 1.17 | 0.19 | 0.19 | 4.00 | 0.17 | 0.23 | 1.79 | 0.20 | 0.56 | 0.39 | 0.34 | 0.34 | 0.49 | 0.37 | 1.36 | 0.78 |
| cora | 1.12 | 0.09 | 0.07 | 2.11 | 0.10 | 0.18 | 0.78 | 0.09 | 0.38 | 0.17 | 0.17 | 0.17 | 0.36 | 0.18 | 0.19 | 0.73 |
| hm-categories | 6.46 | 5.39 | 5.12 | 17.90 | 854.18 | 2.17 | 155.22 | 5.66 | 6.05 | 10.66 | 5.26 | 10.72 | 841.23 | 6.05 | 155.89 | 22.45 |
| ogbn-arxiv | 1.69 | 0.77 | 0.76 | 7.20 | 1.04 | 1.32 | 8.66 | 0.87 | 1.43 | 1.61 | 0.97 | 1.11 | OOM | 1.11 | 8.48 | 2.68 |
| ogbn-products | 48.00 | 74.82 | 74.70 | 135.49 | 125.35 | 43.99 | OOM | 78.32 | 51.23 | 168.93 | 79.11 | 84.51 | 84.96 | 81.81 | OOM | 158.20 |
| pokec-regions | 19.23 | 17.90 | 17.66 | 33.58 | 23.28 | 12.10 | 267.84 | 19.55 | 21.29 | 39.54 | 20.43 | 21.22 | 18.78 | 21.86 | OOM | 39.91 |
| pubmed | 1.13 | 0.13 | 0.12 | 2.26 | 0.20 | 0.20 | 1.27 | 0.12 | 0.36 | 0.23 | 0.19 | 0.20 | 0.43 | 0.21 | 0.77 | 0.59 |
| tolokers-2 | 1.13 | 0.33 | 0.33 | 3.32 | 1.60 | 0.40 | 6.32 | 0.37 | 0.55 | 0.75 | 0.38 | 0.81 | 1.61 | 0.63 | 6.02 | 1.41 |
| twitch-views | 5.81 | 5.02 | 4.92 | 20.60 | 144.10 | 3.68 | 98.90 | 5.51 | 5.60 | 10.29 | 5.31 | 15.80 | 140.11 | 5.78 | 98.86 | 14.38 |
| web-traffic | 16.91 | 6.95 | 6.93 | 93.93 | 83131.37 | 27.56 | 121.47 | 8.82 | 22.63 | 32.88 | 22.53 | 326.42 | 37547.82 | 27.17 | 131.90 | 53.53 |

*Table 61.* Latency (ms) measurements on SpMM (with GCN as the operator) on all datasets (hidden dim $D$=256); **lower is better**. OOM indicates that the we couldn't measure the metric for the layer.

| Dataset | Fwd time | | | | | | | | Bwd time | | | | | | | |
|---|---|---|---|---|---|---|---|---|---|---|---|---|---|---|---|---|
| | DGL | cuSPARSE | cuSPARSE+ $A^\top$ | FuseGNN | TC-GNN | Ours (WSB/TC) | PyG | Pytorch CSR-SpMM | DGL | cuSPARSE | cuSPARSE+ $A^\top$ | FuseGNN | TC-GNN | Ours (WSB/TC) | PyG | Pytorch CSR-SpMM |
| artnet-exp | 1.42 | 0.53 | 0.53 | 5.05 | 0.83 | 0.45 | 7.96 | 0.60 | 1.08 | 1.04 | 0.75 | 0.83 | 0.83 | 0.84 | 7.50 | 1.45 |
| avazu-ctr | 11.24 | 9.79 | 9.75 | 40.29 | 348.73 | 3.72 | OOM | 10.96 | 11.07 | 22.29 | 10.08 | 32.90 | 341.79 | 11.28 | OOM | 24.67 |
| citeseer | 1.15 | 0.09 | 0.07 | 2.08 | 0.08 | 0.19 | 0.78 | 0.08 | 0.38 | 0.16 | 0.15 | 0.15 | 0.36 | 0.21 | 0.25 | 0.57 |
| city-reviews | 3.30 | 2.06 | 2.07 | 8.82 | 6.63 | 1.69 | 31.29 | 2.28 | 3.30 | 4.12 | 2.51 | 6.62 | 5.59 | 3.30 | 30.12 | 4.59 |
| city-roads-L | 2.11 | 0.66 | 0.65 | 4.01 | 0.43 | 0.91 | 5.92 | 0.80 | 2.10 | 1.41 | 1.23 | 1.19 | 1.10 | 1.38 | 5.91 | 1.78 |
| city-roads-M | 1.24 | 0.31 | 0.30 | 4.42 | 0.21 | 0.39 | 2.99 | 0.35 | 0.93 | 0.66 | 0.56 | 0.55 | 0.63 | 0.63 | 2.63 | 1.03 |
| cora | 1.15 | 0.08 | 0.07 | 2.11 | 0.10 | 0.18 | 0.78 | 0.09 | 0.38 | 0.17 | 0.17 | 0.15 | 0.36 | 0.19 | 0.26 | 0.74 |
| hm-categories | 11.98 | 10.10 | 10.04 | 30.88 | 854.65 | 2.83 | OOM | 11.12 | 11.75 | 21.18 | 10.27 | 23.74 | 841.04 | 11.35 | OOM | 28.07 |
| ogbn-arxiv | 2.74 | 1.43 | 1.44 | 12.22 | 1.25 | 2.11 | 16.68 | 1.62 | 2.74 | 3.07 | 1.82 | 2.07 | OOM | 2.10 | 16.55 | 3.51 |
| ogbn-products | 94.89 | 149.59 | 149.51 | 219.93 | 142.66 | 84.44 | OOM | 155.02 | 101.84 | 337.85 | 157.88 | 172.14 | 103.64 | 163.52 | OOM | 239.99 |
| pokec-regions | 37.70 | 35.78 | 35.57 | 51.82 | 27.16 | 24.04 | OOM | 38.88 | 42.34 | 78.86 | 40.73 | 43.67 | 25.54 | 43.70 | OOM | 61.64 |
| pubmed | 1.14 | 0.18 | 0.17 | 2.33 | 0.23 | 0.22 | 1.96 | 0.18 | 0.48 | 0.36 | 0.29 | 0.32 | 0.50 | 0.31 | 1.48 | 0.70 |
| tolokers-2 | 1.38 | 0.60 | 0.59 | 3.99 | 1.97 | 0.54 | 12.36 | 0.67 | 0.86 | 1.36 | 0.67 | 1.53 | 1.77 | 1.13 | 12.26 | 1.76 |
| twitch-views | 10.91 | 9.75 | 9.75 | 38.19 | 148.91 | 6.83 | 216.31 | 10.82 | 10.99 | 20.41 | 10.46 | 33.72 | 143.70 | 11.53 | OOM | 19.97 |
| web-traffic | 33.05 | 13.82 | 13.78 | 180.37 | 82664.58 | 38.77 | OOM | 17.54 | 44.95 | 60.83 | 35.80 | 662.34 | 37566.82 | 42.20 | OOM | 68.64 |

*Table 62.* Latency (ms) measurements on SpMM (with GCN as the operator) on all datasets (hidden dim $D$=512); **lower is better**. OOM indicates that the we couldn't measure the metric for the layer.

| Dataset | Fwd mem | | | | | | | | Bwd mem | | | | | | | |
|---|---|---|---|---|---|---|---|---|---|---|---|---|---|---|---|---|
| | DGL | cuSPARSE | cuSPARSE+$\hat{A}^\top$ | FuseGNN | TC-GNN | Ours (WSB/TC) | PyG | Pytorch CSR-SpMM | DGL | cuSPARSE | cuSPARSE+$\hat{A}^\top$ | FuseGNN | TC-GNN | Ours (WSB/TC) | PyG | Pytorch CSR-SpMM |
| artnet-exp | 91.19 | 44.42 | 46.75 | 2398.58 | 46.19 | 78.17 | 2688.81 | 63.88 | 118.34 | 83.86 | 86.19 | 2427.34 | 85.44 | 121.42 | 2713.42 | 111.88 |
| avazu-ctr | 1232.80 | 2148.34 | 5452.49 | 3942.12 | 366.35 | 957.50 | 13964.00 | 559.68 | 1288.66 | 2288.59 | 5592.94 | 3255.39 | 506.50 | 1029.65 | 14001.24 | 1919.91 |
| citeseer | 50.79 | 2378.31 | 5974.95 | 2379.71 | 48.78 | 49.67 | 2385.75 | 49.67 | 52.44 | 2380.81 | 5977.45 | 2382.08 | 51.26 | 52.51 | 2387.37 | 52.31 |
| city-reviews | 375.17 | 2054.77 | 5265.95 | 2647.12 | 217.50 | 348.83 | 3814.95 | 273.32 | 456.89 | 2173.79 | 5384.96 | 2710.85 | 335.94 | 476.72 | 3887.60 | 454.53 |
| city-roads-L | 270.09 | 192.88 | 202.82 | 2528.55 | 187.64 | 217.82 | 2732.23 | 225.10 | 340.37 | 299.43 | 309.36 | 2630.84 | 293.65 | 338.83 | 2801.69 | 335.88 |
| city-roads-M | 79.14 | 1888.28 | 5086.88 | 2380.49 | 45.15 | 57.24 | 2472.56 | 60.41 | 107.48 | 1931.02 | 5129.62 | 2421.11 | 87.88 | 106.24 | 2500.43 | 105.38 |
| cora | 18.04 | 2345.69 | 5942.27 | 2347.26 | 16.29 | 17.14 | 2353.69 | 17.04 | 19.39 | 2347.73 | 5944.32 | 2349.17 | 18.32 | 19.46 | 2355.01 | 19.25 |
| hm-categories | 1130.73 | 2242.86 | 5799.52 | 3844.90 | 292.68 | 775.09 | 13616.91 | 472.96 | 1164.83 | 2359.19 | 5915.85 | 3146.70 | 410.86 | 821.70 | 13639.65 | 1794.56 |
| ogbn-arxiv | 310.60 | 2381.67 | 5861.29 | 2548.97 | 182.52 | 263.20 | 3193.35 | 234.80 | 398.77 | 2511.45 | 5992.37 | 2664.10 | OOM | 410.90 | 3276.03 | 385.25 |
| ogbn-products | 9080.94 | 3989.71 | 6688.19 | 12444.09 | 3581.69 | 9823.04 | 70387.41 | 5171.71 | 10778.69 | 6274.39 | 8972.87 | 10547.88 | 5857.19 | 11955.41 | OOM | 13332.66 |
| pokec-regions | 3408.04 | 1876.55 | 4001.39 | 5222.43 | 1520.27 | 3161.69 | 20327.99 | 2169.11 | 4325.63 | 3201.72 | 5327.88 | 5755.51 | 2839.22 | 4565.96 | 21125.26 | 4450.10 |
| pubmed | 62.80 | 2378.01 | 5975.80 | 2382.04 | 48.93 | 55.91 | 2435.27 | 54.52 | 72.72 | 2393.01 | 5990.73 | 2395.88 | 64.53 | 72.78 | 2444.97 | 70.80 |
| tolokers-2 | 61.66 | 2336.09 | 5933.54 | 2406.19 | 18.65 | 63.46 | 2885.80 | 29.97 | 70.66 | 2348.75 | 5947.17 | 2381.82 | 31.26 | 73.84 | 2891.54 | 96.23 |
| twitch-views | 825.61 | 2420.07 | 5959.21 | 3328.11 | 258.45 | 909.20 | 9620.32 | 405.75 | 948.74 | 2596.34 | 6137.54 | 3021.20 | 437.09 | 1061.00 | 9702.41 | 1278.68 |
| web-traffic | 16532.88 | 14738.82 | 15692.57 | 17346.26 | 14614.41 | 15458.88 | 24988.95 | 15430.80 | 17987.15 | 16928.07 | 17882.62 | 19365.88 | 16794.41 | 17934.15 | 26400.25 | 17816.82 |

*Table 63.* Memory footprint (MB) measurements on SpMM (with GCN as the operator) on all datasets (hidden dim $D$=64); **lower is better**. OOM indicates that the we couldn't measure the metric for the layer.

| Dataset | Fwd mem | | | | | | | | Bwd mem | | | | | | | |
|---|---|---|---|---|---|---|---|---|---|---|---|---|---|---|---|---|
| | DGL | cuSPARSE | cuSPARSE+$\hat{A}^\top$ | FuseGNN | TC-GNN | Ours (WSB/TC) | PyG | Pytorch CSR-SpMM | DGL | cuSPARSE | cuSPARSE+$\hat{A}^\top$ | FuseGNN | TC-GNN | Ours (WSB/TC) | PyG | Pytorch CSR-SpMM |
| artnet-exp | 140.41 | 71.55 | 73.89 | 2419.92 | 70.80 | 108.93 | 3011.81 | 100.79 | 192.18 | 145.39 | 147.72 | 2489.02 | 146.96 | 195.25 | 3061.03 | 186.46 |
| avazu-ctr | 1288.66 | 2269.07 | 5574.31 | 3960.73 | 405.11 | 1004.81 | 24765.24 | 615.54 | 1400.38 | 2380.79 | 5688.31 | 3348.09 | 603.40 | 1145.17 | 24839.72 | 1994.79 |
| citeseer | 54.04 | 2380.82 | 5976.70 | 2382.17 | 50.40 | 51.70 | 2394.27 | 52.11 | 57.31 | 2386.63 | 5982.86 | 2387.90 | 55.32 | 57.39 | 2398.45 | 57.18 |
| city-reviews | 520.48 | 2137.46 | 5348.63 | 2693.65 | 290.15 | 439.65 | 5097.74 | 382.30 | 674.86 | 2355.43 | 5566.60 | 2893.84 | 517.58 | 694.69 | 5243.05 | 646.07 |
| city-roads-L | 411.17 | 265.77 | 275.71 | 2598.01 | 258.18 | 305.09 | 3007.41 | 330.90 | 551.78 | 475.77 | 485.71 | 2804.49 | 469.79 | 550.44 | 3146.33 | 548.13 |
| city-roads-M | 135.28 | 1917.09 | 5115.69 | 2408.36 | 73.15 | 92.27 | 2593.03 | 102.41 | 191.61 | 2000.69 | 5199.69 | 2490.78 | 157.88 | 190.38 | 2648.76 | 189.38 |
| cora | 20.68 | 2347.30 | 5944.14 | 2348.82 | 17.61 | 18.79 | 2361.72 | 19.03 | 23.36 | 2352.06 | 5949.03 | 2353.49 | 21.63 | 23.42 | 2364.82 | 23.22 |
| hm-categories | 1164.20 | 2347.82 | 5904.48 | 3856.26 | 314.15 | 803.51 | 24140.09 | 507.06 | 1232.54 | 2416.03 | 5972.69 | 3203.54 | 464.53 | 889.91 | 24185.56 | 1839.77 |
| ogbn-arxiv | 473.35 | 2470.10 | 5949.71 | 2630.85 | 365.89 | 365.90 | 2870.82 | OOM | 642.90 | 2718.16 | 6197.77 | 2870.42 | OOM | 655.02 | 4093.55 | 628.48 |
| ogbn-products | 11473.22 | 5676.39 | 8374.97 | 13042.09 | 4777.69 | 11318.08 | OOM | 6965.99 | 14367.15 | 9264.39 | 11962.97 | 13537.69 | 8847.19 | 15543.41 | OOM | 15725.35 |
| pokec-regions | 5005.51 | 2804.56 | 4930.72 | 5621.43 | 2319.00 | 4159.06 | 36877.03 | 3367.21 | 6721.83 | 5198.56 | 7324.72 | 7752.34 | 4836.05 | 6962.16 | 38471.56 | 6357.74 |
| pubmed | 82.80 | 2388.20 | 5985.14 | 2391.74 | 59.30 | 67.94 | 2497.88 | 69.71 | 103.09 | 2417.08 | 6014.02 | 2419.94 | 89.35 | 103.16 | 2517.14 | 101.18 |
| tolokers-2 | 72.37 | 2346.00 | 5943.70 | 2409.67 | 24.39 | 70.64 | 3404.24 | 38.98 | 88.90 | 2363.22 | 5961.89 | 2396.17 | 45.61 | 90.70 | 3415.92 | 108.27 |
| twitch-views | 974.33 | 2555.30 | 6094.58 | 3369.15 | 338.62 | 1010.85 | 16422.73 | 528.88 | 1194.18 | 2801.56 | 6340.85 | 3226.42 | 637.52 | 1302.48 | 16586.91 | 1443.20 |
| web-traffic | 19356.60 | 16222.07 | 17176.62 | 18758.61 | 16026.41 | 17224.06 | 34109.31 | 17548.80 | 22223.15 | 20458.07 | 21412.62 | 22896.23 | 20324.41 | 22170.15 | 36931.89 | 22052.82 |

*Table 64.* Memory footprint (MB) measurements on SpMM (with GCN as the operator) on all datasets (hidden dim $D$=128); **lower is better**. OOM indicates that the we couldn't measure the metric for the layer.

| Dataset | Fwd mem | | | | | | | | Bwd mem | | | | | | | |
|---|---|---|---|---|---|---|---|---|---|---|---|---|---|---|---|---|
| | DGL | cuSPARSE | cuSPARSE+$\hat{A}^\top$ | FuseGNN | TC-GNN | Ours (WSB/TC) | PyG | Pytorch CSR-SpMM | DGL | cuSPARSE | cuSPARSE+$\hat{A}^\top$ | FuseGNN | TC-GNN | Ours (WSB/TC) | PyG | Pytorch CSR-SpMM |
| artnet-exp | 241.96 | 122.33 | 124.66 | 2469.14 | 121.57 | 172.01 | 3657.81 | 176.96 | 344.09 | 272.33 | 274.66 | 2612.08 | 273.90 | 347.58 | 3756.26 | 338.79 |
| avazu-ctr | 1400.38 | 2343.55 | 5648.80 | 3997.54 | 478.07 | 1097.15 | 46367.72 | 727.26 | 1624.02 | 2566.99 | 5872.24 | 3534.30 | 785.81 | 1364.82 | 46516.69 | 2143.75 |
| citeseer | 60.54 | 2383.24 | 5979.89 | 2384.58 | 53.65 | 55.76 | 2408.83 | 56.98 | 67.57 | 2392.99 | 5989.84 | 2394.26 | 63.44 | 67.64 | 2415.33 | 67.44 |
| city-reviews | 813.86 | 2282.77 | 5495.32 | 2838.43 | 436.84 | 622.67 | 7665.54 | 602.33 | 1114.92 | 2718.71 | 5933.32 | 3256.59 | 884.30 | 1134.75 | 7956.16 | 1085.92 |
| city-roads-L | 686.86 | 403.62 | 413.56 | 2736.93 | 396.02 | 478.20 | 3557.78 | 537.67 | 965.33 | 820.39 | 830.32 | 3151.80 | 814.41 | 963.98 | 3836.18 | 960.86 |
| city-roads-M | 247.28 | 1972.82 | 5171.95 | 2464.09 | 129.15 | 162.33 | 2833.96 | 186.41 | 359.61 | 2140.03 | 5339.95 | 2630.12 | 297.88 | 358.38 | 2945.43 | 357.38 |
| cora | 25.97 | 2349.94 | 5946.50 | 2351.46 | 20.26 | 22.10 | 2377.64 | 22.99 | 31.29 | 2358.45 | 5954.73 | 2359.88 | 28.24 | 31.36 | 2383.51 | 31.15 |
| hm-categories | 1232.41 | 2393.29 | 5950.48 | 3879.00 | 360.68 | 860.88 | 45190.01 | 575.27 | 1368.95 | 2529.71 | 6088.48 | 3317.21 | 580.86 | 1028.97 | 45280.95 | 1930.97 |
| ogbn-arxiv | 806.60 | 2635.48 | 6116.34 | 2797.03 | 430.52 | 573.21 | 5397.83 | 606.80 | 1142.77 | 3131.60 | 6614.34 | 3284.25 | OOM | 1154.90 | 5728.58 | 1129.25 |
| ogbn-products | 16257.25 | 8068.58 | 10766.97 | 14237.94 | 7169.69 | 14308.08 | OOM | 10553.99 | 21543.15 | 15244.58 | 17942.97 | 19518.20 | 14827.19 | 22719.41 | OOM | 20509.35 |
| pokec-regions | 8191.64 | 4397.62 | 6523.78 | 7185.04 | 3912.07 | 6151.49 | 69970.10 | 5756.81 | 11501.03 | 9181.23 | 11307.39 | 11735.01 | 8818.72 | 11741.36 | OOM | 11136.95 |
| pubmed | 122.80 | 2407.38 | 6004.39 | 2410.92 | 79.30 | 93.50 | 2622.89 | 100.08 | 163.09 | 2465.15 | 6062.16 | 2468.01 | 139.72 | 163.90 | 2661.40 | 161.92 |
| tolokers-2 | 96.97 | 2357.36 | 5953.85 | 2415.29 | 36.90 | 85.66 | 4441.60 | 57.23 | 125.41 | 2391.81 | 5988.30 | 2425.64 | 76.91 | 127.73 | 4464.56 | 132.87 |
| twitch-views | 1302.67 | 2719.47 | 6258.76 | 3451.24 | 502.79 | 1216.07 | 30027.56 | 775.14 | 1686.56 | 3211.99 | 6751.28 | 3636.85 | 1047.96 | 1795.00 | 30355.91 | 1771.46 |
| web-traffic | 24999.24 | 19043.25 | 19997.80 | 21579.79 | 18847.59 | 20751.23 | 52346.48 | 21780.57 | 30686.68 | 27511.01 | 28465.57 | 29949.17 | 27377.35 | 30633.69 | 57991.66 | 30516.35 |

*Table 65.* Memory footprint (MB) measurements on SpMM (with GCN as the operator) on all datasets (hidden dim $D$=256); **lower is better**. OOM indicates that the we couldn't measure the metric for the layer.

| Dataset | Fwd mem | | | | | | | | Bwd mem | | | | | | | |
|---|---|---|---|---|---|---|---|---|---|---|---|---|---|---|---|---|
| | DGL | cuSPARSE | cuSPARSE+$\hat{A}^\top$ | FuseGNN | TC-GNN | Ours (WSB/TC) | PyG | Pytorch CSR-SpMM | DGL | cuSPARSE | cuSPARSE+$\hat{A}^\top$ | FuseGNN | TC-GNN | Ours (WSB/TC) | PyG | Pytorch CSR-SpMM |
| artnet-exp | 435.75 | 219.23 | 221.56 | 2567.59 | 218.47 | 294.30 | 4950.70 | 322.30 | 644.77 | 514.57 | 516.90 | 2858.20 | 516.14 | 638.26 | 5147.59 | 629.47 |
| avazu-ctr | 1682.29 | 2492.51 | 5797.76 | 4072.06 | 627.04 | 1283.35 | OOM | 950.70 | 2070.91 | 2939.40 | 6244.65 | 3906.70 | 1158.22 | 1811.71 | OOM | 2441.68 |
| citeseer | 74.04 | 2389.74 | 5986.80 | 2391.08 | 60.15 | 63.88 | 2439.61 | 67.24 | 87.57 | 2409.23 | 6006.29 | 2410.50 | 80.20 | 87.64 | 2452.60 | 87.44 |
| city-reviews | 1392.36 | 2573.40 | 5784.57 | 3129.06 | 726.09 | 985.26 | 12800.16 | 1036.21 | 1982.68 | 3445.28 | 6656.45 | 3983.15 | 1607.41 | 2002.51 | 13381.42 | 1953.92 |
| city-roads-L | 1243.17 | 681.77 | 691.71 | 3014.78 | 674.18 | 825.28 | 4658.66 | 954.90 | 1799.78 | 1515.77 | 1525.71 | 3846.57 | 1509.79 | 1798.44 | 5214.35 | 1796.13 |
| city-roads-M | 471.28 | 2084.29 | 5283.95 | 2575.57 | 241.15 | 302.33 | 3315.83 | 354.41 | 695.61 | 2418.70 | 5619.95 | 2908.79 | 577.88 | 694.38 | 3538.77 | 693.38 |
| cora | 36.55 | 2355.58 | 5952.71 | 2357.09 | 25.54 | 28.71 | 2409.19 | 30.93 | 47.16 | 2371.44 | 5968.58 | 2372.88 | 41.46 | 47.22 | 2419.43 | 47.02 |
| hm-categories | 1371.25 | 2484.24 | 6040.89 | 3924.41 | 450.56 | 974.03 | OOM | 744.26 | 1641.78 | 2757.07 | 6313.72 | 3544.36 | 805.57 | 1299.16 | OOM | 2112.86 |
| ogbn-arxiv | 1465.59 | 2966.22 | 6445.84 | 3126.97 | 760.02 | 986.02 | 8337.15 | 1101.04 | 2131.26 | 3958.47 | 7438.08 | 4111.12 | OOM | 2143.38 | 8998.64 | 2117.74 |
| ogbn-products | 25825.22 | 12852.58 | 15550.97 | 18069.63 | 11953.69 | 20288.00 | OOM | 17729.99 | 35895.15 | 27204.58 | 29902.97 | 31478.20 | 26787.19 | 37071.41 | OOM | 34460.41 |
| pokec-regions | 14573.51 | 7588.56 | 9714.72 | 10375.98 | 7103.00 | 10138.96 | OOM | 10543.21 | 21073.83 | 17158.56 | 19284.72 | 19712.34 | 16796.05 | 21314.16 | OOM | 20709.74 |
| pubmed | 196.84 | 2445.89 | 6042.90 | 2449.43 | 116.32 | 140.89 | 2873.05 | 155.61 | 274.15 | 2561.42 | 6158.43 | 2564.28 | 232.27 | 274.96 | 2950.07 | 273.12 |
| tolokers-2 | 140.67 | 2380.33 | 5976.82 | 2426.29 | 58.83 | 113.85 | 6514.03 | 90.25 | 191.04 | 2449.22 | 6045.71 | 2482.65 | 131.73 | 194.04 | 6559.96 | 179.03 |
| twitch-views | 1959.37 | 3047.82 | 6587.11 | 3615.41 | 831.14 | 1626.50 | 57238.60 | 1267.67 | 2671.60 | 4032.86 | 7572.15 | 4457.72 | 1868.83 | 2780.04 | OOM | 2513.90 |
| web-traffic | 36292.88 | 24690.07 | 25644.62 | 27226.61 | 24494.41 | 27808.64 | OOM | 30250.80 | 47627.15 | 41628.07 | 42582.62 | 44066.23 | 41494.41 | 47574.15 | OOM | 47456.82 |

*Table 66.* Memory footprint (MB) measurements on SpMM (with GCN as the operator) on all datasets (hidden dim $D$=512); **lower is better**. OOM indicates that the we couldn't measure the metric for the layer.

| Dataset | Fwd time | | | Bwd time | | | Fwd mem | | | Bwd mem | | |
|---|---|---|---|---|---|---|---|---|---|---|---|---|
| | DGL | cuGraph | Ours | DGL | cuGraph | Ours | DGL | cuGraph | Ours | DGL | cuGraph | Ours |
| artnet-exp | 0.42 | 0.45 | 0.17 | 0.26 | 0.33 | 0.19 | 97.05 | 83.81 | 58.03 | 133.97 | 120.73 | 94.94 |
| avazu-ctr | 10.89 | 22.69 | 1.66 | 0.44 | 0.58 | 0.39 | 847.90 | 907.31 | 301.54 | 903.76 | 963.17 | 357.40 |
| citeseer | 0.33 | 4.75 | 0.08 | 0.24 | 0.31 | 0.15 | 51.60 | 50.62 | 49.58 | 54.04 | 53.06 | 52.02 |
| city-reviews | 1.95 | 3.47 | 0.46 | 0.45 | 0.41 | 0.33 | 381.74 | 344.90 | 246.03 | 490.73 | 453.89 | 355.02 |
| city-roads-L | 0.51 | 0.36 | 0.25 | 0.41 | 0.35 | 0.31 | 307.07 | 264.54 | 222.36 | 411.26 | 368.73 | 326.55 |
| city-roads-M | 0.33 | 0.28 | 0.15 | 0.27 | 0.39 | 0.20 | 93.58 | 76.39 | 59.07 | 135.52 | 118.39 | 101.00 |
| cora | 0.32 | 0.23 | 0.08 | 0.24 | 0.43 | 0.16 | 18.65 | 17.86 | 16.93 | 20.63 | 19.84 | 18.91 |
| hm-categories | 5.80 | 11.40 | 1.57 | 0.31 | 0.32 | 0.23 | 739.79 | 805.50 | 223.02 | 773.90 | 839.61 | 258.39 |
| ogbn-arxiv | 3.16 | 0.73 | 0.40 | 0.84 | 0.51 | 0.34 | 342.07 | 293.82 | 221.32 | 468.07 | 419.82 | 347.32 |
| ogbn-products | 27.82 | 55.25 | 10.25 | 6.59 | 14.26 | 4.68 | 7893.99 | 7618.08 | 3725.95 | 9687.72 | 9411.80 | 5519.76 |
| pokec-regions | 10.79 | 13.76 | 3.91 | 5.10 | 6.36 | 3.07 | 3407.92 | 3025.80 | 1814.16 | 4603.82 | 4221.70 | 3010.06 |
| pubmed | 0.33 | 0.22 | 0.09 | 0.28 | 0.48 | 0.17 | 67.08 | 60.66 | 53.55 | 81.53 | 75.85 | 68.74 |
| tolokers-2 | 0.59 | 0.99 | 0.19 | 0.26 | 0.34 | 0.17 | 47.81 | 48.35 | 17.64 | 56.43 | 57.36 | 26.85 |
| twitch-views | 8.76 | 15.99 | 1.41 | 0.61 | 0.61 | 0.47 | 652.44 | 650.90 | 249.31 | 775.57 | 774.03 | 375.31 |
| web-traffic | 60.85 | 495.45 | 4.29 | 12.44 | 4.59 | 10.52 | 17162.69 | 16329.47 | 15290.86 | 19280.69 | 18447.47 | 17408.86 |

*Table 67.* Latency (ms) and Memory footprint (MB) measurements on reduction-based convolutions (with min-aggregation as the operator) on all datasets (hidden dim $D$=64); **lower is better**.

| Dataset | Fwd time | | | Bwd time | | | Fwd mem | | | Bwd mem | | |
|---|---|---|---|---|---|---|---|---|---|---|---|---|
| | DGL | cuGraph | Ours | DGL | cuGraph | Ours | DGL | cuGraph | Ours | DGL | cuGraph | Ours |
| artnet-exp | 0.56 | 0.45 | 0.23 | 0.32 | 0.32 | 0.23 | 162.44 | 133.81 | 94.94 | 236.27 | 207.65 | 168.78 |
| avazu-ctr | 11.00 | 21.25 | 2.51 | 0.57 | 1.67 | 0.49 | 947.17 | 981.79 | 359.67 | 1061.17 | 1094.27 | 473.67 |
| citeseer | 0.32 | 0.22 | 0.08 | 0.25 | 0.42 | 0.15 | 55.87 | 53.87 | 52.01 | 60.74 | 58.74 | 56.89 |
| city-reviews | 2.45 | 3.74 | 0.65 | 0.78 | 0.81 | 0.55 | 573.15 | 490.90 | 355.02 | 791.12 | 708.87 | 572.99 |
| city-roads-L | 0.88 | 0.37 | 0.35 | 0.69 | 0.54 | 0.45 | 490.48 | 404.00 | 328.17 | 700.48 | 614.00 | 538.17 |
| city-roads-M | 0.43 | 1.90 | 0.19 | 0.34 | 0.36 | 0.24 | 167.13 | 132.39 | 101.07 | 251.13 | 216.39 | 185.07 |
| cora | 0.33 | 0.22 | 0.08 | 0.24 | 0.33 | 0.15 | 22.12 | 20.51 | 18.91 | 26.08 | 24.47 | 22.88 |
| hm-categories | 6.38 | 11.61 | 2.44 | 0.44 | 0.42 | 0.33 | 798.74 | 850.97 | 255.23 | 866.94 | 919.18 | 323.44 |
| ogbn-arxiv | 3.80 | 0.79 | 0.54 | 1.17 | 0.77 | 0.50 | 557.01 | 458.51 | 343.38 | 805.07 | 706.57 | 591.44 |
| ogbn-products | 45.11 | 59.89 | 17.19 | 12.78 | 31.43 | 9.06 | 11033.56 | 10010.45 | 5519.95 | 14621.56 | 13598.45 | 9107.85 |
| pokec-regions | 17.96 | 14.22 | 6.77 | 9.40 | 12.53 | 6.32 | 5502.90 | 4621.06 | 3012.25 | 7896.90 | 7015.06 | 5406.25 |
| pubmed | 0.32 | 0.26 | 0.12 | 0.27 | 0.32 | 0.16 | 93.10 | 80.66 | 68.74 | 122.73 | 111.03 | 99.11 |
| tolokers-2 | 0.75 | 1.05 | 0.24 | 0.26 | 0.43 | 0.16 | 64.15 | 60.74 | 26.85 | 81.37 | 77.97 | 44.08 |
| twitch-views | 11.04 | 16.97 | 2.29 | 1.09 | 1.09 | 0.84 | 866.00 | 815.08 | 370.53 | 1112.26 | 1061.34 | 616.79 |
| web-traffic | 66.11 | 492.70 | 6.34 | 17.48 | 7.83 | 13.29 | 20867.69 | 19152.06 | 17408.86 | 25103.69 | 23388.06 | 21644.86 |

*Table 68.* Latency (ms) and Memory footprint (MB) measurements on reduction-based convolutions (with min-aggregation as the operator) on all datasets (hidden dim $D$=128); **lower is better**.

| Dataset | Fwd time | | | Bwd time | | | Fwd mem | | | Bwd mem | | |
|---|---|---|---|---|---|---|---|---|---|---|---|---|
| | DGL | cuGraph | Ours | DGL | cuGraph | Ours | DGL | cuGraph | Ours | DGL | cuGraph | Ours |
| artnet-exp | 0.91 | 0.72 | 0.33 | 0.55 | 0.51 | 0.37 | 293.20 | 233.03 | 171.11 | 443.20 | 383.03 | 321.11 |
| avazu-ctr | 12.98 | 41.36 | 4.65 | 1.06 | 1.10 | 0.80 | 1140.41 | 1130.76 | 469.12 | 1363.85 | 1354.20 | 692.56 |
| citeseer | 0.32 | 0.23 | 0.08 | 0.25 | 0.32 | 0.14 | 64.40 | 60.37 | 56.89 | 74.65 | 70.62 | 67.14 |
| city-reviews | 3.45 | 8.49 | 1.06 | 1.45 | 1.55 | 1.01 | 955.97 | 782.22 | 575.05 | 1393.97 | 1220.22 | 1013.05 |
| city-roads-L | 1.63 | 0.53 | 0.48 | 1.26 | 0.94 | 0.79 | 853.69 | 680.92 | 534.93 | 1270.46 | 1097.69 | 951.70 |
| city-roads-M | 0.73 | 0.35 | 0.24 | 0.58 | 0.60 | 0.37 | 314.10 | 244.39 | 185.07 | 482.10 | 412.39 | 353.07 |
| cora | 0.32 | 0.27 | 0.08 | 0.25 | 0.34 | 0.13 | 29.06 | 25.79 | 22.88 | 36.99 | 33.73 | 30.81 |
| hm-categories | 9.24 | 19.21 | 4.36 | 0.66 | 0.65 | 0.48 | 919.79 | 941.92 | 325.02 | 1057.79 | 1078.86 | 463.02 |
| ogbn-arxiv | 4.97 | 1.35 | 0.85 | 2.26 | 1.39 | 0.91 | 993.03 | 789.89 | 593.32 | 1491.03 | 1287.89 | 1091.32 |
| ogbn-products | 82.42 | 112.56 | 33.40 | 26.48 | 64.93 | 19.19 | 17312.52 | 14794.45 | 9107.95 | 24488.52 | 21970.45 | 16283.85 |
| pokec-regions | 33.70 | 26.59 | 12.70 | 19.60 | 25.95 | 13.36 | 9686.60 | 7809.60 | 5401.86 | 14470.20 | 12593.20 | 10185.46 |
| pubmed | 0.38 | 0.34 | 0.16 | 0.27 | 0.29 | 0.20 | 144.39 | 119.91 | 99.11 | 204.39 | 179.91 | 159.11 |
| tolokers-2 | 0.91 | 1.70 | 0.38 | 0.26 | 0.35 | 0.19 | 94.58 | 83.71 | 45.11 | 130.58 | 119.71 | 80.59 |
| twitch-views | 15.40 | 34.21 | 4.71 | 2.27 | 2.63 | 1.68 | 1297.92 | 1143.43 | 616.79 | 1790.44 | 1635.95 | 1109.31 |
| web-traffic | 83.04 | 904.21 | 9.19 | 35.69 | 16.64 | 26.77 | 28276.05 | 24796.65 | 21640.63 | 36743.81 | 33264.42 | 30108.39 |

*Table 69.* Latency (ms) and Memory footprint (MB) measurements on reduction-based convolutions (with min-aggregation as the operator) on all datasets (hidden dim $D$=256); **lower is better**.

| Dataset | Fwd time | | | Bwd time | | | Fwd mem | | | Bwd mem | | |
|---|---|---|---|---|---|---|---|---|---|---|---|---|
| | DGL | cuGraph | Ours | DGL | cuGraph | Ours | DGL | cuGraph | Ours | DGL | cuGraph | Ours |
| artnet-exp | 1.70 | 1.33 | 0.60 | 1.01 | 0.88 | 0.68 | 549.30 | 428.38 | 316.45 | 844.64 | 723.72 | 611.79 |
| avazu-ctr | 21.97 | 85.48 | 9.13 | 1.83 | 1.66 | 1.35 | 1532.27 | 1428.76 | 692.56 | 1979.16 | 1875.64 | 1139.45 |
| citeseer | 0.32 | 0.29 | 0.08 | 0.25 | 0.29 | 0.16 | 81.96 | 73.36 | 67.14 | 101.45 | 93.36 | 87.14 |
| city-reviews | 6.20 | 15.05 | 1.95 | 2.83 | 3.33 | 1.92 | 1717.55 | 1362.84 | 1008.93 | 2589.44 | 2234.72 | 1880.81 |
| city-roads-L | 3.20 | 0.92 | 0.78 | 2.39 | 1.73 | 1.49 | 1584.19 | 1237.07 | 952.17 | 2418.19 | 2071.07 | 1786.17 |
| city-roads-M | 1.36 | 0.50 | 0.37 | 1.04 | 1.09 | 0.66 | 607.11 | 467.33 | 353.07 | 943.11 | 803.33 | 689.07 |
| cora | 0.32 | 0.37 | 0.08 | 0.25 | 0.33 | 0.14 | 43.04 | 36.37 | 30.81 | 58.91 | 52.24 | 46.68 |
| hm-categories | 18.36 | 50.94 | 9.18 | 1.12 | 1.01 | 0.80 | 1156.41 | 1123.92 | 459.85 | 1429.24 | 1396.75 | 732.68 |
| ogbn-arxiv | 7.33 | 2.58 | 1.62 | 3.59 | 2.88 | 1.75 | 1859.21 | 1450.63 | 1087.57 | 2851.46 | 2442.88 | 2079.81 |
| ogbn-products | 164.52 | 226.48 | 66.50 | 53.65 | 133.67 | 38.85 | 29869.13 | 24360.97 | 16283.95 | 44221.13 | 38712.97 | 30635.85 |
| pokec-regions | 67.55 | 53.65 | 26.32 | 40.12 | 53.29 | 28.55 | 18060.50 | 14188.67 | 10188.25 | 27630.50 | 23758.67 | 19758.25 |
| pubmed | 0.60 | 0.51 | 0.21 | 0.44 | 0.42 | 0.30 | 244.22 | 196.42 | 154.64 | 359.75 | 311.95 | 270.17 |
| tolokers-2 | 1.39 | 3.38 | 0.60 | 0.34 | 0.39 | 0.26 | 153.38 | 128.79 | 77.93 | 222.27 | 197.69 | 146.82 |
| twitch-views | 24.54 | 75.88 | 9.82 | 4.90 | 6.13 | 3.73 | 2158.87 | 1800.12 | 1109.31 | 3143.92 | 2785.16 | 2094.35 |
| web-traffic | 116.39 | 1807.62 | 16.34 | 56.23 | 28.66 | 37.85 | 43096.64 | 36087.83 | 30110.86 | 60034.64 | 53025.83 | 47048.86 |

*Table 70.* Latency (ms) and Memory footprint (MB) measurements on reduction-based convolutions (with min-aggregation as the operator) on all datasets (hidden dim $D$=512); **lower is better**.

