# OpenReview forum: "On Efficient Scaling of GNNs via IO-Aware Layers Implementations"
_ICML.cc/2026/Conference — ICML 2026 spotlight_

### Official Review · Reviewer_MKDQ · 2026-03-04

**Soundness:** 3
**Presentation:** 3
**Significance:** 3
**Originality:** 3
**Overall Recommendation:** 4
**Confidence:** 3

**Summary:**

This paper analyzes the hardware performance bottlenecks of mainstream GNN layers from a unified perspective of I/O and computational intensity. Specifically, it proposes practical and implementable GPU kernel optimization schemes for three types of operator families, including SpMM-based, reduction-based, and attention-based operators. This paper bridges the gap between algorithmic abstraction and hardware reality, providing significant performance acceleration (up to several times to several tens of times in some scenarios) and memory savings. The experimental results are comprehensive, and it has been fully verified on a variety of real-world graph datasets. It has also been compared with multiple mainstream and cutting-edge baselines, and the conclusions are reliable.

**Compliance With Llm Reviewing Policy:**

Affirmed.

**Key Questions For Authors:**

1. The verification is conducted only in a single GPU (A100 80GB), without involving the adaptation for multi-GPU distributed training (such as model parallelism, data parallelism). Besides, it has not been tested on models larger than 100B or large-scale graphs, and thus cannot verify the algorithm's scalability under extreme scale conditions.
2. The experiments mainly focused on tasks such as node classification and general pre-training, but were not validated on complex tasks such as graph classification, link prediction, and temporal GNN.

**Limitations:**

See Questions.

**Strengths And Weaknesses:**

## Strengths

1. It captures that the core bottleneck of GNN stems from sparse and irregular memory access and limited data locality, and this problem becomes more severe as the gap between GPU computing throughput and HBM bandwidth widens. The research problem has a strong practical relevance.
2. It is the first to divide the mainstream GNN layers into three major kernel families, and designs differentiated optimization schemes for the differences in hardware behaviors of each family. The SpMM layer reuses cuSPARSE and optimizes cache and transposition strategies, the reduction layer adopts degree-aware partitioning and atomic reduction, and the attention layer draws inspiration from FlashAttention to design a fusion kernel and support Tensor Core.
3. The experimental results are comprehensive, including single-core performance, end-to-end training, GPU memory usage, graph reordering impact, etc. These results are remarkable. The forward speed of GATv2 reaches up to 8.5 $\times$, the GPU memory usage is reduced by 76 $\times$, and the Graph Transformer Tensor Core variant achieves an acceleration of 7.3 $\times$ on local dense graphs, fully verifying the effectiveness and robustness of the method.

## Weaknesses

---

> ### Author Rebuttal · Authors · 2026-03-30
>
> Thank you for your feedback, positive assessment and for recognizing the practical relevance! We address your concerns below.
>
> > Single GPU only, no multi-GPU training.
>
> Our paper focuses on kernel-level optimizations - these compute primitives execute on a single GPU even in distributed scenarios. This is the standard setting for GNN kernel papers. Our kernels **apply** to multi-GPU training **without modification**: each GPU executes the same kernels on its local graph partition, with only boundary gradient exchange as additional communication - orthogonal to our kernel-level scope. Our **memory savings are especially beneficial** here: reduced per-GPU footprint means larger partitions per GPU (fewer GPUs needed) or more memory for communication buffers. For example, our $76\times$ memory reduction on GATv2 could enable single-GPU execution of graphs that otherwise wouldn't be possible with DGL.
>
> > Not tested on models larger than 100B or large-scale graphs.
>
> GNN models are typically millions of parameters (not billions) - the bottleneck is sparse graph traversal, not model size. Could the reviewer clarify what “100B” refers to in the GNN context?
>
> Regarding graph scale, as discussed in our response to Reviewer w2R3, our suite includes `ogbn-products` (2.4M nodes, 123M edges) and `web-traffic` (2.9M nodes, 12.9M edges), which are among the largest for full-graph single-GPU evaluation. We also note that graph diversity (degree distribution, density, structure) is at least as important as raw size for evaluating kernel-level optimizations, which is why we selected graphs with heterogeneous structural properties.
>
> We additionally evaluate on three larger-scale graphs - `DGraph-Fraud` [1] (3.7M nodes, 8.0M edges), `T-Social` [2] (5.7M nodes, 145M edges), and `ogbn-papers100M` (111M nodes, 1.7B edges). We compare forward-pass kernel latency against DGL.
>
> On `ogbn-papers100M`, DGL runs out of memory while our backend completes the Graph Transformer (GT) forward pass in ~1.0s. The results on `DGraph-Fraud` and `T-Social` are displayed in Table 1.
>
> All this combined demonstrates that our kernels provide more efficient alternatives in terms of memory and latency, scaling better to even larger graphs.
>
> |Graph|Convolution|Ours (ms)|DGL (ms)|Speedup|
> |-|-|-|-|-|
> |`DGraph-Fraud`|GT|51.7|68.2|**1.32x**|
> |`DGraph-Fraud`|GATv2|20.4|82.2|**4.03x**|
> |`DGraph-Fraud`|SpMM (GCN)|9.8|22.5|**2.29x**|
> |`DGraph-Fraud` |Min Aggr|15.8|42.2|**2.66x**|
> |`T-Social`|GT|362.1|537.3|**1.48x** |
> |`T-Social`|GATv2|192.1|338.8|**1.76x**|
> |`T-Social`|SpMM (GCN)|99.4|74.6|0.75x|
> |`T-Social`|Min Aggr|104.0|131.0|**1.26x**|
>
>
> Table 1. Latency speedup measurements on additional large graph datasets. For Graph Transformer (GT) and GATv2 we use heads=4, head dim=128; for SpMM-based (GCN) and Reduction-based (Min Aggr) convolutions we use hidden dim=256.
>
> > No validation on complex tasks.
>
> We note that our paper is a systems/kernel optimization contribution, not an architectural paper - we do not propose new GNN architectures but accelerate existing ones. Our kernels are **task-agnostic by construction**, thus **optimized layers are identical regardless of downstream tasks**. In terms of that, benchmarking on other tasks was outside of the scope of the work, as we verified numerical correctness against DGL and made sure that our kernels are **mathematically equivalent** interchangeable drop-in replacements.
>
> However, as **we demonstrated end-to-end equivalence in our response to Reviewer DhTi**, we additionally validate the same architecture on two OGB link prediction benchmarks - `ogbl-citation2` (2.9M nodes, 30M edges) evaluated with MRR, and `ogbl-vessel` (3.5M nodes, 12M edges) evaluated with ROC-AUC - changing only the prediction head for link prediction task.
>
> DGL runs out of memory on GATv2 and GT for both graphs, while our implementations complete training successfully. When both backends finish (for SpMM-based and Reduction-based convolutions), metric difference is negligible (less than 1% diff in all cases), confirming that **our kernels preserve downstream quality**. It shows that our optimized convolution layers **generalize beyond node classification to other complex tasks** without modification.
>
> Regarding graph property prediction, standard benchmarks consist of small graphs where the bottleneck is batching/data loading, not convolution - our optimizations target the regime where graph size creates memory and compute pressure.
>
> Regarding temporal GNNs, whether using temporal features on a static graph or applying convolutions per snapshot of an evolving graph, the per-layer operation is unchanged - our kernels apply directly.
>
> [1] Huang et al. DGraph: A Large-Scale Financial Dataset for Graph Anomaly Detection. NeurIPS 2022
>
> [2] Tang et al. Rethinking Graph Neural Networks for Anomaly Detection. ICML 2022

---

### Official Review · Reviewer_DejS · 2026-03-09

**Soundness:** 3
**Presentation:** 3
**Significance:** 3
**Originality:** 2
**Overall Recommendation:** 5
**Confidence:** 3

**Summary:**

The paper takes an I/O- and arithmetic-intensity-centric view of GNN computation on GPUs. The authors observe that widely used GNN layers cluster into three kernel families with distinct hardware bottlenecks: (i) SpMM-based convolutions (e.g., GCN), (ii) reduction-based aggregations (e.g., min/max), and (iii) attention-based layers (e.g., GATv2, Graph Transformer). For each family, they develop GPU kernels that target reduced HBM traffic and improved locality. For SpMM-based layers, the main finding is that cuSPARSE with proper caching of descriptors and a precomputed transposed adjacency remains highly competitive. For reduction-based layers, they introduce degree-aware tiling that partitions nodes into light and heavy subsets to address load imbalance under heavy-tailed degree distributions. For attention-based layers, they adapt FlashAttention-style online softmax and recomputation to the sparse graph setting via fused CSR kernels, and optionally leverage Tensor Cores through a weighted block-sparse format. The paper also studies graph reordering and finds its benefit is kernel- and graph-dependent. Experiments on diverse graphs show speedups of up to 10x for reduction, 8.5x for GATv2, and 3.9x for Graph Transformer, with substantial memory savings for attention layers.

**Compliance With Llm Reviewing Policy:**

Affirmed.

**Final Justification:**

I'm satisfied with the reply and I have no further comments.

**Key Questions For Authors:**

1. For the attention-based fused CSR kernels, how does performance degrade (if at all) as the variance in node degree increases within a single graph? The paper shows results on graphs with different average degrees, but the intra-graph degree skew effect on warp utilization and load balance of the fused kernel is not explicitly characterized. Understanding this would clarify robustness.

2. The backward pass for attention-based layers shows more modest gains (and sometimes slowdowns) compared to the forward pass, attributed to atomic-heavy gradient accumulation. Have you explored alternatives such as deterministic reduction via pre-allocated buffers or two-pass backward schemes that could avoid atomics? This seems like a key bottleneck limiting end-to-end training speedups.

3. The block-sparse Tensor Core (WSB) format requires preprocessing to pack neighborhoods into 16x16 tiles. What is the wall-clock cost of this preprocessing relative to the per-iteration savings, and after how many training iterations does it amortize? This is important for understanding practical applicability, especially for dynamic graphs or inductive settings where the graph changes between batches.

**Limitations:**

Yes. The authors discuss limitations throughout the paper, including the backward-pass inefficiency of block-sparse formats for directed graphs, the non-universality of graph reordering, and the lack of an adaptive backend selector. The "no one size fits all" aspect is explicitly acknowledged as future work. There are no obvious negative societal impact concerns for a systems-level GPU kernel paper.

**Strengths And Weaknesses:**

**Strengths:**

- *Soundness:* The experimental methodology is rigorous and thorough. The authors evaluate across a diverse set of graphs with varying density and degree distributions (Graph-Land, OGB, citation networks), sweep multiple hyperparameter configurations (feature dimensions, head counts), and report both latency and peak memory for forward and backward passes separately. They are honest about limitations: e.g., the backward pass inefficiency of the block-sparse Tensor Core path, cuSPARSE being hard to beat for SpMM, and the non-universality of graph reordering. The roofline-based reasoning about why GNN operators are memory-bound is sound and well-grounded in hardware reality.

- *Presentation:* The paper is clearly written and well structured. The three-family taxonomy provides a clean organizational framework. The background section (Sections 2.1–2.5) is comprehensive and positions the work well against prior systems (DGL, PyG, GE-SpMM, DF-GNN, TC-GNN, FuseGNN, Fused3S, GNNAdvisor, MaxK-GNN). The paper is easy to follow for a reader with GPU systems background. Additional graphs (beyond the tables) would facilitate visual comparison of speedups across graphs and configurations, though this is a minor and somewhat subjective point.

- *Significance:* The work addresses a practical and important problem — GNN training/inference is a real bottleneck in many applications, and the gap between peak GPU throughput and achieved GNN performance is substantial. The drop-in replacement philosophy with minimal dependencies (CUDA, PyTorch, cuSPARSE) is valuable for adoption. The finding that no single approach dominates all settings is itself a useful practical insight.


**Weaknesses:**

- *Originality:* This is the main dimension where the paper is weakest, though not fatally so. The individual techniques draw heavily from existing ideas:
  - **SpMM-based layers:** The contribution is essentially using cuSPARSE with cached descriptors and a precomputed transpose — standard engineering practice rather than a novel algorithmic contribution. The authors themselves frame this as a practical finding ("cuSPARSE is still hard to beat").
  - **Reduction-based layers:** Degree-aware partitioning into light/heavy node sets and tiling for high-degree nodes are well-established GPU load-balancing strategies. The use of atomics on packed 64-bit (value, index) pairs for partial reductions is a known technique. The combination is sensible but not surprising.
  - **Attention-based layers:** The fused CSR attention kernels are a direct adaptation of FlashAttention/FlashAttention-2 (online softmax, recomputation-based backward storing compact per-node statistics) to the sparse graph setting. The block-sparse Tensor Core variant extends the format from Fused3S (Li & Chandramowlishwaran, 2025) to support weighted adjacency. The adaptation is non-trivial — irregular neighborhoods with varying sizes are fundamentally different from the regular sequence lengths of standard transformers — but the conceptual blueprint is borrowed.
  - **Graph reordering:** The study uses existing METIS-based reordering and the novelty lies in the empirical analysis of when it helps, not in the reordering method itself.

  That said, the paper offers a coherent and unified systems perspective that combines these techniques under a principled IO-centric framework. The novelty is more in the synthesis, the careful adaptation to irregular graph structures, and the systematic empirical evaluation than in any single technique.

- *Limited novelty in the "what" vs. strong execution in the "how":* The paper reads more as a high-quality systems/engineering contribution than a methodological advance. Readers looking for new algorithmic ideas may find the contribution incremental. However, the practical impact is real and the engineering is solid.

- *No "one size fits all" solution:* The authors correctly acknowledge this as a limitation and point to an adaptive runtime/autotuner as future work. As it stands, a practitioner must understand graph properties and kernel characteristics to select the right backend, which limits out-of-the-box usability.

---

> ### Author Rebuttal · Authors · 2026-03-30
>
> Thank you for the thorough review! We particularly appreciate the observation that “no single approach dominates all settings is itself a useful practical insight”.
>
> > Originality and individual techniques
>
> We appreciate the assessment of the originality and of the individual techniques. We'd also like to point out that known techniques don't imply trivial adaptation. **We address this in more detail in responses to reviewers DhTi and w2R3.**
>
> > How does fused CSR attention degrade with intra-graph degree variance?
>
> When intra-graph degree variance is high, thread blocks assigned to high-degree nodes run longer than those for low-degree nodes, creating tail latency and SM underutilization.
>
> To further address this challenge, we implemented a **degree-aware extension** for our kernels: nodes are partitioned into light/heavy buckets, each launching a separate kernel with independently configured warp counts. This directly addresses the load imbalance caused by degree skew. We report speedups over the original degree-agnostic kernels for Graph Transformer (GT) and GATv2 in Table 1.
>
> |Graph|Degree skewness|GT Fwd Speedup|GT Bwd Speedup|GATv2 Fwd Speedup|GATv2 Bwd Speedup|
> |-|-:|-:|-:|-:|-:|
> |`avazu-ctr`|9.4|2.48x|1.37x|2.30x|2.04x|
> |`city-reviews`|35.0|2.07x|1.40x|2.24x|1.78x|
> |`hm-categories`|7.3|1.79x|1.26x|1.62x|1.65x|
> |`ogbn-arxiv`|110.8|3.72x|1.09x|3.91x|2.49x|
> |`ogbn-products`|17.5|1.07x|1.02x|1.19x|1.11x|
> |`pokec-regions`|71.8|1.07x|1.02x|1.13x|1.10x|
> |`twitch-views`|42.4|2.10x|1.45x|2.23x|2.08x|
> |`web-traffic`|738.1|5.83x|6.34x|5.45x|7.34x|
>
> Table 1. Obtained speedups for our degree-aware warp allocation vs. our original kernel; head dim=512, heads=4.
>
> Speedup correlates with degree skewness with extreme hubs (`web-traffic` skew=738, `ogbn-arxiv` skew=111) yielding the highest gains. GT backward benefits less than GATv2 backward due to atomicAdd on dK regardless of warp count. We address the atomic contention challenge below.
>
> We can additionally provide NCU profiling metrics if the reviewer would find this valuable.
>
> > Alternatives to atomics in the backward pass?
>
> We have considered several alternatives and, inspired by your question, implemented the most promising ones:
>
> **1.** For **undirected** graphs, the backward pass traverses the same CSR as forward, removing the need for atomic operations and enabling more efficient backward pass. As reported in our response to Reviewer DhTi, we **developed dedicated kernels for CUDA and WSB**. For WSB it achieves a median 7.4x speedup. The CSR-based GT gains 5–21% from eliminating atomics. We report results in Table 2.
>
> |Graph|Avg Deg|GT (CUDA)|GATv2 (CUDA)|GT(WSB)|
> |:-|-:|:-|:-|:-|
> |`avazu-ctr`|289|1.21x|1.12x|8.59x|
> |`city-roads-L`|4|1.10x|1.01x|2.75x|
> |`hm-categories`|461|1.19x|1.08x|8.16x|
> |`ogbn-arxiv`|14|1.13x|1.09x|7.73x|
> |`ogbn-products`|51|1.11x|1.14x|7.03x|
> |`pokec-regions`|28|1.11x|0.97x|6.76x|
> |`tolokers-2`|89|1.01x|1.05x|7.57x|
> |`twitch-views`|81|1.03x|1.01x|9.36x|
> |`web-traffic|9|1.06x|1.21x|10.74x|
>
> Table 2. Backward speedup of undirected kernel vs. directed baseline (head dim 128, 4 heads) using kernel fusion via exploiting the symmetry of the adjacency matrix. All backends produce numerically identical gradients.
>
>
> **2.** Additionally, for GATv2 on **undirected** graphs, gradients computation w.r.t. source and destination features can be fused into a single kernel since both traverse edges in the same direction. We implemented this technique during the rebuttal. This eliminates one full graph pass; its effect is reflected in the GATv2 (CUDA) column of Table 3; it shows 1–12% improvement.
>
>
> **3.** Pre-allocated buffers/two-pass backward would reintroduce edge messages materialization negating our key memory advantage (e.g., 44x reduction on twitch-views). We considered this at an early design stage and chose to preserve the low-memory property.
>
>
> > WSB preprocessing cost and amortization
>
> We measured WSB construction using an optimized C++ CPU implementation. Table 3 reports per-epoch training time and break-even points on representative graphs.
>
> |Graph|WSB cost(s)|DGL (ms/ep)|WSB (ms/ep)|Speedup|Break-even (epochs)|
> |-|-:|-:|-:|-:|-:|
> |`avazu-ctr`|14.9|825.5|731.9|1.13x|159|
> |`city-roads-L`|4.4|296.3|285.5|1.04x|404|
> |`city-roads-M`|4.3|145.5|139.4|1.04x|693|
> |`hm-prices`|16.1|731.2|484.4|1.51x|65|
>
> Table 3. WSB amortization (GT, 4 layers, head dim=128, heads=4). Break-even - number of epochs to amortize the WSB construction (measured from reading from disk to obtaining WSB).
>
> On larger/denser graphs (hm-prices, avazu-ctr), WSB amortizes within 65–159 epochs. On 9/16 graphs, the CSR-fused backend is faster per-epoch, requiring more than 1000 epochs to amortize. It’s consistent with our paper's message that WSB is an optional acceleration path for locally dense graphs, not a universal replacement.
>
> We note that WSB format needs to be constructed only once per graph structure and can be serialized to disk for reuse across training runs.

---

> > ### Author Rebuttal · Reviewer_DejS · 2026-04-03
> >
> > I'm satisfied with the reply and I have no further comments.

---

> > > ### Author Response · Authors · 2026-04-07
> > >
> > > Dear Reviewer,
> > >
> > > We sincerely appreciate your confirmation that your concerns have been addressed, and thank you for your time and effort in reviewing our paper and carefully reading our rebuttal. We are thrilled that our responses to your technical questions were satisfactory.
> > >
> > > We strongly agree with your insightful observation that _"no single approach dominates all settings"_ and believe this is one of the key practical takeaways of our work. In the final version, we will incorporate all additional results developed during the rebuttal, including the degree-aware warp allocation analysis, the undirected backward kernels, and the WSB amortization study.
> > >
> > > Thank you again for the exceptional depth and care of your review.

---

### Official Review · Reviewer_w2R3 · 2026-03-15

**Soundness:** 3
**Presentation:** 3
**Significance:** 2
**Originality:** 1
**Overall Recommendation:** 3
**Confidence:** 3

**Summary:**

This paper studies the performance bottleneck of GNNs executed on GPUs. The paper discussed the IO & arithmetic intensity tradeoff for various existing optimization techniques, such as kernel fusion, graph partitioning (e.g., degree-based), and graph re-ordering, and delivers an optimized implementation that achieves execution time speedup and memory reduction compared with standard baselines such as cuSPARSE, DGL and PyG.

**Compliance With Llm Reviewing Policy:**

Affirmed.

**Final Justification:**

The responses partially addressed my concerns, while the followings still remain:

- Novelty / originality / technical depth: I agree that carefully engineered systems could be highly valuable to the community. However, the individual components in the proposed system are still well-known in the community.

- Graph size and subgraph sampling: those two are related points. First, I appreciate that the authors provide results on graphs of much larger scale. As the authors pointed out, when the graph grows, the hardware will eventually OOM. Many realistic applications indeed have their graphs so large that the full memory cannot fit (additionally, when the GNN becomes larger and deeper, the memory consumption will also explode). In those cases, subgraph sampling-based computation is a must rather than an option. However, applying the proposed framework to subgraph-based models may not be straightforward. I believe many proposed optimization steps rely on a static graph structure (e.g., graph reordering, degree-aware aggregation, block-sparse layout) -- e.g., re-applying reordering on every sampled subgraph on the fly may be too expensive; degree of the same node appearing in different subgraph samples would be dramatically different. Having a clear discussion on how the proposed system optimization can / cannot be combined with those algorithm-level sampling-based methods would be valuable in my opinion.

**Key Questions For Authors:**

See "Weaknesses" above

**Strengths And Weaknesses:**

## Strengths

+ Addressing the memory bound of GNN execution on FLOPS-optimized GPUs is an important topic.
+ The paper gives a thorough overview of the various infra optimization techniques for accelerating GNN execution.
+ The experiments are comprehensive, and reveal some interesting observations (e.g., the benefits of reordering w.r.t. graph properties)


## Weaknesses

- The paper is mostly engineering- / implementation-focused, and reads like a technical report. Optimizing memory movement for graph workloads is a well-established domain. The mentioned techniques of blocking, reordering, attention caching strategies, etc. are also known in the literature. The proposed framework lacks novelty / originality.
- Many evaluated graphs are rather small. For a systems paper studying performance bottleneck, it would be more convincing to include larger benchmarks. In addition, for tables in Sec 4, it would be better to also include the raw wall-clock time / memory consumption (in addition to the current speedup numbers) to evaluate the practical significance of the speedup.
- In addition to the system-wise optimization discussed in the paper, another important research direction in scaling up GNN execution is regarding the subgraph sampling. i.e., instead of performing full-graph message passing (which may not be even feasible when the graph is too large), works like [1, 2] sample small subgraphs where GNN can efficiently operate on. Such sampling techniques are also important components in frameworks like PyG and DGL. It would be better to include some discussion on how the proposed framework relate to the sampling-based models.

-----
### References

[1] GraphSAINT: Graph Sampling Based Inductive Learning Method. In ICLR 2020.

[2] Decoupling the Depth and Scope of Graph Neural Networks. In NeurIPS 2021.

---

> ### Author Rebuttal · Authors · 2026-03-30
>
> Thank you for your thorough feedback! We address your concerns below.
>
> > Engineering/implementation-focused; techniques are known; lacks novelty.
>
> We believe our contribution extends beyond any single technique. First, adaptation to irregular sparse graphs is non-trivial: variable-size neighborhoods require fundamentally different parallelization than the uniform sequence lengths assumed by existing ideas; the backward pass for directed graphs introduces atomic-heavy scatter patterns not addressed by prior techniques; and degree-aware tiling with hardcoded thresholds is fragile and error-prone in existing methods (e.g., DF-GNN fails on most of benchmarked graphs due to excessive shared memory requirements). To our knowledge, prior GNN systems each target isolated operators and none of them delivers complete fused fwd+bwd attention for both GATv2 and Graph Transformer with demonstrated memory savings (up to 76x).
>
> Second, the community recognizes that principled adaptation yielding substantial gains constitutes a meaningful contribution - SAGEAttention [1] builds directly on FlashAttention for quantization-aware consumer-GPU attention; FlashAttention itself builds on well-known online softmax and tiling ideas.
>
> Finally, our unified IO-centric treatment of three operator families - identifying when each strategy applies and when it does not - enables practitioners to accelerate full GNN pipelines, not just individual layers.
>
> Thus, known techniques do not imply trivial adaptation. **We detail graph-specific challenges and improvements in our responses to Reviewers DhTi and DejS.**
>
> > Many evaluated graphs are rather small.
>
> Our suite includes `ogbn-products` (2.4M nodes, 123M edges) and `web-traffic` (2.9M nodes, 12.9M edges) - among the largest graphs used in the literature. We note that graph diversity (degree distribution, density, structure) is at least as important as raw size for evaluating kernel-level optimizations, which is why we selected graphs with heterogeneous structural properties.
>
> We additionally evaluate on three larger-scale graphs - `DGraph-Fraud` [1] (3.7M nodes, 8.0M edges), `T-Social` [2] (5.7M nodes, 145M edges), and `ogbn-papers100M` (111M nodes, 1.7B edges). We compare forward-pass kernel latency against DGL.
>
> On `ogbn-papers100M`, DGL runs OOM while our backend completes the Graph Transformer (GT) forward pass in ~1.0s. The results on `DGraph-Fraud` and `T-Social` are displayed in Table 1.
>
> All this combined demonstrates that our kernels provide more efficient alternatives in terms of memory and latency, scaling better to even larger graphs.
>
> |Graph|Convolution|Ours (ms)|DGL (ms)|Speedup|
> |-|-|-|-|-|
> |`DGraph-Fraud`|GT|51.7|68.2|**1.32x**|
> |`DGraph-Fraud`|GATv2|20.4|82.2|**4.03x**|
> |`DGraph-Fraud`|SpMM (GCN)|9.8|22.5|**2.29x**|
> |`DGraph-Fraud` |Min Aggr|15.8|42.2|**2.66x**|
> |`T-Social`|GT|362.1|537.3|**1.48x** |
> |`T-Social`|GATv2|192.1|338.8|**1.76x**|
> |`T-Social`|SpMM (GCN)|99.4|74.6|0.75x|
> |`T-Social`|Min Aggr|104.0|131.0|**1.26x**|
>
> Table 1. Latency speedup measurements on additional large graph datasets. For GT and GATv2 we use heads=4, head dim=128; for SpMM-based (GCN) and Reduction-based convolutions (Min Aggr) we use hidden dim=256.
>
> > Raw wall-clock time / memory consumption should be included.
>
> We provide this information in Appendix D (Tables 38–67), reporting raw latency (ms) and memory (MB) for all configurations. We will improve cross-references to that section.
>
> > Relationship to subgraph sampling.
>
> Our kernel-level optimizations are **orthogonal to and composable with** subgraph sampling. Mini-batch methods produce subgraphs on which the same GNN layers execute - **faster kernels mean faster per-batch iterations, regardless of how batches are formed**.
>
> We would like to point out that the choice between full-graph and mini-batch training is itself an active research question. Recent studies [2,3] show the comparison is nuanced: mini-batch can converge faster but depends heavily on hyperparameter tuning; larger batch sizes tend to improve final test quality with full-graph being the limiting case; and mini-batch introduces additional stochasticity from neighborhood sampling that loses full graph structure information.
>
> Full-graph training remains the dominant evaluation setting in the GNN literature, thus making it efficient is directly impactful. We will add this discussion in the revision.
>
> [1] Zhang et al. SageAttention: Accurate 8-Bit Attention for Plug-and-play Inference Acceleration. ICLR 2025
>
> [2] Bajaj et al. Graph Neural Network Training Systems: A Performance Comparison of Full-Graph and Mini-Batch. 2025
>
> [3] Liu et al. Full-Graph vs. Mini-Batch Training: Comprehensive Analysis from a Batch Size and Fan-Out Size Perspective. https://openreview.net/forum?id=ZSfgsh43vT. 2026
>
> [4] Huang et al. DGraph: A Large-Scale Financial Dataset for Graph Anomaly Detection. NeurIPS 2022
>
> [5] Tang et al. Rethinking Graph Neural Networks for Anomaly Detection. ICML 2022

---

> > ### Author Rebuttal · Reviewer_w2R3 · 2026-04-03
> >
> > I thank the authors for providing the detailed rebuttal and new experimental results. The responses partially addressed my concerns, while the followings still remain:
> >
> > - Novelty / originality / technical depth: I agree that carefully engineered systems could be highly valuable to the community. However, the individual components in the proposed system are still well-known in the community.
> >
> > - Graph size and subgraph sampling: those two are related points. First, I appreciate that the authors provide results on graphs of much larger scale. As the authors pointed out, when the graph grows, the hardware will eventually OOM. Many realistic applications indeed have their graphs so large that the full memory cannot fit (additionally, when the GNN becomes larger and deeper, the memory consumption will also explode). In those cases, subgraph sampling-based computation is a must rather than an option. However, applying the proposed framework to subgraph-based models may not be straightforward. I believe many proposed optimization steps rely on a static graph structure (e.g., graph reordering, degree-aware aggregation, block-sparse layout) -- e.g., re-applying reordering on every sampled subgraph on the fly may be too expensive; degree of the same node appearing in different subgraph samples would be dramatically different. Having a clear discussion on how the proposed system optimization can / cannot be combined with those algorithm-level sampling-based methods would be valuable in my opinion.

---

> > > ### Author Response · Authors · 2026-04-04
> > >
> > > We thank the reviewer for the thoughtful follow-up. We address the remaining points.
> > >
> > > > Novelty, originality, technical depth
> > >
> > > We agree that several building blocks we use are known in the systems literature. However, our contribution is not a single new primitive, but the combination of: (i) a unified IO-centric decomposition of mainstream GNN layers into kernel families with distinct hardware bottlenecks, (ii) careful adaptation of these techniques to irregular sparse graphs, where neighborhood sizes, memory access patterns, and backward-pass behavior differ substantially from dense settings, and (iii) a systematic empirical analysis of when each optimization path helps or fails in practice. We believe this combination is the paper’s main technical contribution and practical value. More broadly, GNN efficiency remains underexplored compared to dense deep learning, and our results show that large gains in memory and latency are achievable, enabling larger graphs, higher-dimensional models, and more experiments on the same hardware through easy-to-use drop-in replacements.
> > >
> > > We note that this framing is consistent with the assessment of Reviewers DejS, MKDQ, and DhTi. In particular, DejS identifies our novelty as _"the synthesis, the careful adaptation to irregular graph structures, and the systematic empirical evaluation."_ MKDQ calls this _"the first to divide the mainstream GNN layers into three major kernel families”_ with _“differentiated optimization schemes."_
> > >
> > > We would also like to additionally point out that engineering contributions synthesizing known techniques into practical, high-impact systems are regularly accepted at top ML venues. The PyTorch paper (NeurIPS 2019) was reviewed with "There is no one big idea here. Ideas emerging from previous works are clearly stated" yet accepted as it demonstrated "the successful synthesis of several key ideas [1]." Similarly, einops (ICLR 2022, Oral), FlashAttention (NeurIPS 2022), and SAGEAttention (ICLR 2025) all build on **existing techniques**; their **contribution is making them work in practice**.
> > >
> > > [1] [PyTorch Reviews](https://shorturl.at/DtyVv)
> > >
> > > > Subgraph sampling
> > >
> > > We argue that our optimizations are straightforward to use in standard mini-batch neighbor-sampling pipelines. The key reason is that the main fused kernels operate on the sampled subgraph exactly as they do on the full graph, and no kernel redesign is required. In practice, subgraph construction already runs on CPU workers in parallel with GPU computation via prefetching (like in standard PyG/DGL dataloaders), so the additional per-subgraph setup needed by our backends stays off the critical GPU path.
> > >
> > > This integration is lightweight for the main operator families. For CSR-based fused attention, the sampled subgraph is simply converted to CSR and consumed directly by the same kernel as in the full-graph case. For degree-aware aggregation, the common fixed-fanout setting is especially simple: sampled degrees are bounded and often nearly uniform within each layer, so the required partitioning is trivial; with variable fanout, it is still cheap to compute from the sampled subgraph CSR `indptr`. For graph reordering, no per-subgraph recomputation is needed in our setup: sampled subgraphs preserve the reordered global node IDs, so the locality induced by the full-graph permutation carries over automatically.
> > >
> > > For completeness, we note that the main path under sampling is the CSR-fused backend. The block-sparse path is optional in our framework and is generally less attractive for typical small-fanout sampled subgraphs as blocks are under-filled. This is consistent with our message that WSB is an optional acceleration path rather than universal backend.
> > >
> > > To verify that this is not just a conceptual claim, **we integrated our fused kernels into a standard neighbor-sampling** mini-batch training pipeline with **zero kernel changes**. Concretely, we combine them with the PyTorch Geometric neighbor sampler; the sampled mini-batches are converted to CSR/WSB and then consumed directly by our kernels. Subgraph construction is handled by parallel CPU workers and overlapped with GPU execution via prefetching.
> > >
> > > We evaluated this setup on three large-scale graphs: `ogbn-products`, `pokec-regions`, and `web-traffic`, covering all convolution families. We use a 5-layer GNN with fanouts **`[15, 10, 10, 5, 5]`**. The hidden dimension is 512 for Reduction-based and SpMM-based convolutions, and 4 heads with head dimension 128 for attention-based ones. The number of CPU workers is 12, and the batch size is 1024 nodes. Compared to the standard DGL neighbor-sampling pipeline, our implementation achieves an average 1.19$\times$**wall-clock speedup**.
> > >
> > > The key result is that our implementations plug into sampled-subgraph workflows **directly and with minimal engineering effort**, without redesigning the kernels and with only the standard training-loop changes needed to move from full-batch to mini-batch training.

---

### Official Review · Reviewer_DhTi · 2026-03-24

**Soundness:** 3
**Presentation:** 4
**Significance:** 3
**Originality:** 2
**Overall Recommendation:** 5
**Confidence:** 2

**Summary:**

This work studies the training and inference efficiency of GNNs on GPUs.
They study three common operator families in GNNs: SpMM-based convolutions, reduction-based aggregations, attention-based layers, analyzing widely used public frameworks and various bottlenecks in them. They find that on GPUs, a lot of these operations are memory-bound due to irregular access patterns and intermediate materialization of activations.
For each of the three operator families, they focus on IO/memory-aware methods while designing their own kernels and benchmarking those against publicly available ones.

Overall, this work aims at analyzing and improving the hardware efficiency of a broad class of operators used in GNNs.

**Compliance With Llm Reviewing Policy:**

Affirmed.

**Final Justification:**

My concerns have been largely addressed, I believe it will be useful to include the additional results presented in the rebuttal in the final version.

I believe this work has value for the community: it demonstrates strong gains on a subset of problems, and the analysis of different GNN workloads is useful. I am not fully convinced about the novelty of the techniques but acknowledge that their application and empirical validation in the domain of GNNs could be of value.

**Key Questions For Authors:**

See weakness above

**Limitations:**

yes

**Strengths And Weaknesses:**

### Strengths

1. This work studies a broad class of GNN operators and aims to provide a unified library covering multiple commonly used GNN layers.
2. The paper is generally well-presented. The proposed taxonomy of operators helps in understanding the broader picture and the performance characteristics of each class.
3. The empirical results demonstrate strong improvements in GATv2 and reduction-based convolution experimental settings.

### Weakness

1. The individual techniques used in the paper largely build on existing ideas (e.g. online softmax, degree-aware tiling). The combination of these ideas does lead to meaningful practical improvements in some areas, but the overall conceptual novelty seems incremental.
2. The WSB code path leads to degraded backward pass performance in the case of Graph Transformers. A more detailed analysis of this degradation would strengthen the paper as backward passes are very important for training workloads.
3. Some benchmark results comparing loss convergence per unit of wall-clock time would be helpful to understand the exact end-to-end impact of the proposed techniques.

---

> ### Author Rebuttal · Authors · 2026-03-30
>
> Thank you for the constructive feedback! We address your concerns below.
>
> > Individual techniques build on existing ideas
>
> We believe our contribution extends beyond any single technique. First, adaptation to irregular sparse graphs is non-trivial: variable-size neighborhoods require fundamentally different parallelization than the uniform sequence lengths assumed by existing ideas; the backward pass for directed graphs introduces atomic-heavy scatter patterns not addressed by prior techniques; and degree-aware tiling with hardcoded thresholds is fragile and error-prone in existing methods (e.g., DF-GNN fails on most of benchmarked graphs due to excessive shared memory requirements). To our knowledge, no prior work delivers complete fused fwd+bwd attention for both GATv2 and Graph Transformer with the memory savings we demonstrate (up to 76$\times$).
>
> Second, the community recognizes that principled adaptation yielding substantial gains constitutes a meaningful contribution - SAGEAttention [1] builds directly on FlashAttention for quantization-aware consumer-GPU attention; FlashAttention itself builds on well-known online softmax and tiling ideas.
>
> Finally, our unified IO-centric treatment of three operator families - identifying when each strategy applies and when it does not - enables practitioners to accelerate full GNN pipelines, not just individual layers.
>
> > WSB backward pass degradation needs more analysis.
>
> The root cause is the mismatch between the WSB format's row-oriented construction and the column-oriented access required by the backward pass. In the forward pass, destination nodes are grouped into $16\times16$ row windows and processed via Tensor Core tile pairs. However, computing gradients w.r.t. source features (dK, dV) requires accumulating from all destination windows referencing a given source - in directed graphs, this source appears at different column positions across windows, forcing atomic additions that **serialize** updates. Building a transposed block-sparse representation would add memory and preprocessing overhead.
>
> This is a fundamental tension in sparse block formats for directed graphs, not unique to our approach, which we believe has _remained underexplored_ earlier, as TC-GNN required an undirected graph for Tensor Core operations and Fused-3S provided only a forward-pass kernel. As we note in the paper, to mitigate this, one can use WSB/TC for forward (up to 7.3$\times$ speedup) and fall back to CSR-fused for backward.
>
> Additionally, For **undirected graphs** atomics can be eliminated entirely for WSB. We developed a dedicated **direction-aware** backward kernel for this case. We convert graphs to undirected versions and measure the speedup of the new kernel over the original one:
>
> |Graph|Bwd speedup over original kernel|
> |:-|-|
> |`avazu-ctr`|8.59$\times$|
> |`city-roads-L`|2.75$\times$|
> |`hm-categories`| 8.16$\times$|
> |`ogbn-arxiv`|7.73$\times$|
> |`ogbn-products`|7.03$\times$|
> |`pokec-regions`|6.76$\times$|
> |`twitch-views`|9.36$\times$|
> |`tolokers-2`|7.57$\times$|
> |`web-traffic`|10.74$\times$|
> Table 1. WSB-TC backward speedup for the representative graphs: dedicated undirected kernel vs. original directed kernel (heads=4, head dim=512).
>
> As can be seen, we obtain large speedups over our original directed kernel, indicating that we can neglect bottlenecks introduced by atomics in some specific cases. We also deal with the atomics problem for CSR-based kernels in response to Reviewer DejS.
>
> > Loss convergence per unit of wall-clock time.
>
> We ran end-to-end training with 3-layer GNNs (hidden 512, 2-layer MLP after each conv; for attention models: 4 heads, head dim 128) on node classification task. Our kernels are numerically equivalent drop-in replacements, so convergence trajectories and downstream metrics match (training losses diff for all setups are less than 0.01, test accuracy diff is less than 1%). We report the results for attention-based models on the representative subset of graphs in Table 2. Reduction/SpMM-based models show more modest speedup gains (1.10–1.54$\times$), consistent with aggregation kernels being a smaller fraction of total training time.
>
> |Graph|GATv2 DGL (s)|GATv2 Ours (s)|Speedup|$\Delta$Acc (%)|GT DGL (s)|GT Ours (s)|Speedup|$\Delta$Acc (%)|
> |-|-|-|-|-|-|-|-|-|
> |`ogbn-arxiv`|421.3|329.9|1.28$\times$|+0.14|411.0|345.5|1.19$\times$|−0.34|
> |`artnet-exp`|135.1| 86.8|1.56$\times$|−0.18|121.5|108.5|1.12$\times$|+0.27|
> |`city-reviews`|298.1|133.1|2.24$\times$|+0.07|183.7|141.2|1.30$\times$|+0.60|
> |`tolokers-2`|118.0|43.8|2.70$\times$|−0.20|65.6|51.6|1.27$\times$|+0.38|
>
> Table 2. End-to-end training wall-clock time and test accuracy difference (ours minus DGL).
>
> We are ready to report additional results upon your request. We also show the convergence on a link-prediction task in the response to Reviewer MKDQ.
>
> [1] Zhang et al. SageAttention: Accurate 8-Bit Attention for Plug-and-play Inference Acceleration. ICLR 2025

---

> > ### Author Rebuttal · Reviewer_DhTi · 2026-04-03
> >
> > Thanks to the authors for response!
> > My concerns have been largely addressed, I believe it will be useful to include these results in the final version.
> > I am updating my score.

---

> > > ### Author Response · Authors · 2026-04-07
> > >
> > > Dear Reviewer,
> > >
> > > Thank you for your positive feedback and for acknowledging that your concerns have been largely resolved!
> > >
> > > We completely agree with your suggestion to include the additional rebuttal results in the final version. We will incorporate the undirected backward kernel speedups and the end-to-end wall-clock training comparison, as well as further strengthen the analysis of the WSB backward pass degradation.
> > >
> > > Thank you again for your time, your constructive feedback, and your support for our work.

---

### Decision · Program_Chairs · 2026-04-30

**Decision:**

Accept (spotlight)

**Comment:**

The submission studies GNN computation on GPUs, looking at I/O and arithmetic intensity. They consider three widely used GNN layer families, each of which has different hardware bottlenecks. For each family, the authors develop GPU kernels that reduce HBM traffic and improve locality. The work also presents experiments on a diverse set of graphs and shows substantial speedups.

Reviewers noted that the paper is well-written, tackles a well-motivated problem, and has very rigorous and comprehensive experimental methodology.

The main weaknesses are around novelty (i.e., many existing ideas are used) and at least one reviewer noted that many of the improvements are engineering-focused.